# Operationalizing Data Minimization for Privacy-Preserving LLM Prompting

**Jijie Zhou**[1]    **Niloofar Mireshghallah**[2]    **Tianshi Li**[1]
[1]Northeastern University    [2]Carnegie Mellon University
j.zhou@northeastern.edu    niloofar@cmu.edu    tia.li@northeastern.edu

## Abstract

The rapid deployment of large language models (LLMs) in consumer applications has led to frequent exchanges of personal information. To obtain useful responses, users often share more than necessary, increasing privacy risks via memorization, context-based personalization, or security breaches. We present a framework to formally define and operationalize **data minimization**: for a given user prompt and response model, quantifying the least privacy-revealing disclosure that *maintains* utility, and propose a priority-queue tree search to locate this optimal point within a privacy-ordered transformation space. We evaluated the framework on four datasets spanning open-ended conversations (ShareGPT, Wild-Chat) and knowledge-intensive tasks with single-ground-truth answers (Case-HOLD, MedQA), quantifying achievable data minimization with nine LLMs as the response model. Our results demonstrate that larger frontier LLMs can tolerate stronger data minimization while maintaining task quality than smaller open-source models (**85.7% redaction** for GPT-5 vs. **19.3%** for Qwen2.5-0.5B). By comparing with our search-derived benchmarks, we find that LLMs struggle to predict optimal data minimization directly, showing a bias toward abstraction that leads to oversharing. This suggests not just a privacy gap, but a capability gap: *models may lack awareness of what information they actually need to solve a task.*

## 1 Introduction

Users increasingly reveal sensitive personal information to large language model (LLM) applications (Mireshghallah et al., 2024a; Zhang et al., 2024), exposing themselves to privacy leaks via memorization, context-based personalization, or security breaches. Many share details believing it boosts task performance (Zhang et al., 2024), but this benefit is often illusory: people routinely overshare beyond what utility requires (Zhou et al., 2025). We ask a fundamental question: *What is the minimal information needed to maintain utility while preserving privacy?* This question is essential to quantify oversharing—that is, to compare actual disclosure against the true minimum.

Data minimization, defined as limiting the collection of personal information to what is necessary to accomplish a specified purpose, is a well-established privacy design pattern (Cavoukian et al., 2009) and is explicitly cited in numerous privacy regulations (e.g., GDPR (Parliament & Council, 2016)). Although considerable work has sought to mitigate the oversharing of sensitive information in LLM applications, few studies explicitly *formalize or quantify* this challenge from the perspective of data minimization. Existing approaches typically focus on detecting personal or sensitive disclosures and then apply redaction (e.g., "New York" → "[GEOLOCATION]") or abstraction (e.g., "New York" → "a city in the U.S.") (Dou et al., 2024; Zeng et al., 2025); related efforts develop heuristics to flag information types that are sensitive yet have low semantic relevance to the task (e.g., SSNs (Chowdhury et al., 2025)) or employ LLM-as-a-Judge to assess the relevance or importance of information to guide sanitization (Ma et al., 2025; Ngong et al., 2025). In this work, we introduce a framework that formally operationalizes data minimization for privacy-preserving LLM prompting, and present an algorithm that searches for the minimum privacy disclosure while preserving utility, thereby providing an oracle of data minimization for any prompt and target response generation model.

Figure 1 illustrates our framework with a running example. Our method can be viewed as a specialized tree search for data minimization. Starting from a root node that represents the most heav-

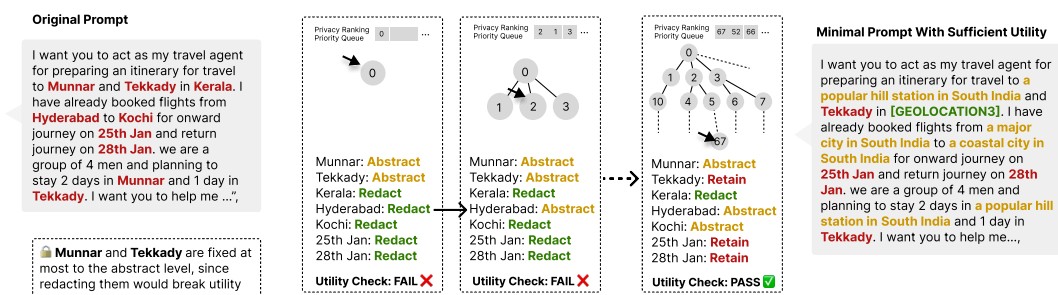

Figure 1: Framework Overview. We present a running example to demonstrate how we perform a tree search ranked by privacy variants, and a transformation that achieves data minimization.

ily sanitized prompt—capturing the globally most privacy-preserving formulation—we iteratively expand the tree. Unlike classical depth-first or breadth-first search, we maintain a priority queue ordered by privacy sensitivity. At each step, we dequeue the least sensitive node, generate slightly more informative (and thus more privacy-revealing) variants as its children, and enqueue them. This process systematically explores the space of possible prompts in order of increasing privacy disclosure, enabling the identification of a minimally sufficient prompt that satisfies the target utility.

Our experimental results show that even under this utility-first constraint, there remains significant room for preserving privacy with data minimization—far exceeding the level of protection typically achieved in current practice. We observe that more powerful frontier models offer greater potential for data minimization than smaller, less capable ones. On open-ended real-world LLM prompts, gpt-5 shows the strongest removal with 85.7% REDACT and 8.6% ABSTRACT (only 5.7% RETAIN), while the smallest model (qwen2.5-0.5b) lags with 19.3% REDACT, 11.0% ABSTRACT, and 69.7% RETAIN.

By comparing with our oracles, we show that LLMs from small edge models to frontier reasoning models are poor predictors of data minimization, which bias towards ABSTRACT actions, leading to prevalent oversharing predictions. Together, these results demonstrate data minimization as a promising paradigm for addressing input privacy in LLM systems, while also revealing gaps in the popular LLM-as-a-Judge method for privacy-utility assessment tasks (Ma et al., 2025; Ngong et al., 2025). **This suggests not just a privacy gap, but a capability gap: models may lack awareness of what information they actually need to solve tasks**. We call for research to investigate the underlying causes of the varied levels of information "redundancy" across models, with the goal of developing robust prediction methods for effective on-device data minimization.

## 2 BACKGROUND & RELATED WORK

**Theoretical & regulatory foundation.** LLMs can expose memorized training data and personally identifiable information (PII) under adversarial prompting, motivating a shift toward minimizing user-side disclosure before inference rather than relying solely on post-hoc filtering. This imperative embodies the data minimization principle, a cornerstone of privacy laws and design guidelines. For example, data minimization is a pillar of the privacy by design framework (Cavoukian et al., 2009), a foundational and widely recognized regulatory framework central to modern data protection regimes such as GDPR Art. 5(1)(c), which limits processing to data necessary for a specified purpose (Parliament & Council, 2016).

**User-led minimization for prompts.** User-assisted tools help them manually sanitize inputs prior to submission (Zhou et al., 2025; Kan et al., 2023). However, these workflows hinge on subjective judgments of what "feels safe," offer *no guarantees of utility preservation*, and rarely include *attacker-based verification* of residual leakage. User studies on implicit inference further show people systematically *underestimate* what models can infer and often choose ineffective rewriting strategies (e.g., paraphrasing) (Wang et al., 2025). In contrast, we automate selection among {RETAIN, ABSTRACT, REDACT} in accordance with the data minimization principle by expanding a tree in increasing order of privacy disclosure, with a priority queue guiding the exploration based on

this privacy order. We employed attacker LLMs tasked with type-wise and span-wise recovery of the redacted and abstracted information in the minimized prompts produced by our method, further verifying the limited recoverable signal and the efficacy of the minimization.

**Utility-preserving minimization and prompt sanitizers.** Prior input sanitization methods either do not consider utility (Dou et al., 2024), seek a balance between privacy and utility (Li et al., 2025b), or aim to maximize utility under a privacy constraint (e.g., a differential privacy budget Chowdhury et al. (2025)). Data minimization, representing a class of methods that optimize privacy under strict utility constraints, has received limited attention. A related line of work relies either on heuristics (e.g., detecting tokens whose format alone indicates sensitive content, such as SSNs Chowdhury et al. (2025)) or on LLM-as-a-Judge to assess how essential or relevant a piece of information is to the task, and then transforms the less essential and sensitive information to maintain utility (Ma et al., 2025; Ngong et al., 2025). However, we caution that it remains unclear to what extent LLM assessments align with the actual importance or necessity of the information, as this alignment depends not only on the semantic meaning of the information and the task but also on the capability of the target model. Our results further show that LLMs are poor predictors of data minimization, highlighting this gap.

**Training-stage defenses (orthogonal).** Differentially private (DP) training/fine-tuning (Abadi et al., 2016) and machine unlearning (Bărbulescu & Triantafillou, 2024) offer training-side protection against the downstream harms of oversharing caused by memorization during training. These approaches require access to model parameters and incur utility and compute costs, and they do not address other threat models to which oversharing is also vulnerable, including inference-stage leakage (Shao et al., 2025), data breaches (Theori Research, 2025; Meta Security Team, 2025; Gadget Review, 2025), or uninformed consents (Zhang et al., 2024; Fast Company, 2025). Our method is *black-box and pre-inference*: it operates solely on the *user input* and uses output-level utility checks, complementing these methods by remaining compatible with closed and rapidly evolving models, when fundamentally mitigating multiple threats through protection of the initial disclosure.

## 3 DATA MINIMIZATION FOR PRIVACY-PRESERVING LLM PROMPTING

### 3.1 PROBLEM FORMULATION

Let $x$ be a user message and let $D = \{e_1, \ldots, e_n\}$ be a set of detected sensitive spans. Each span $e_i$ can be transformed by an action $a_i$ chosen from some finite action space $A$, forming an action vector $a = (a_1, \ldots, a_n)$. Applying $a$ to $x$ yields a transformed message $\tau(x; a)$. Given a target large language model $\mathcal{F}$, we seek a transformation that maximizes privacy while preserving downstream utility. Because placeholders or abstractions may later be replaced with their recovered context, the utility is evaluated *after* a context-recovery step $\mathcal{R}$ that reconstructs a usable output from $\mathcal{F}$:

$$\max_{a \in A^n} \text{Priv}\big(\tau(x; a)\big) \quad \text{subject to} \quad \text{Util}\big(\mathcal{R}(\mathcal{F}(\tau(x; a)); a)\big) \geq \gamma, \quad (1)$$

where

- Priv is any privacy metric (e.g., risk of sensitive-entity disclosure),
- Util is any downstream utility metric evaluated on the recovered output $\mathcal{R}(\mathcal{F}(\cdot))$,
- $\mathcal{R}$ is the context-recovery operator that replaces placeholders or abstractions with the appropriate recovered content, and
- $\gamma$ is a minimum acceptable utility level.

This formulation is agnostic to the choice of action space, privacy/utility metrics, and search strategy.

### 3.2 SPECIFIC INSTANTIATION

In this instantiation, we ground the generic formulation by defining a span-level action space $A = \{\text{RETAIN}, \text{ABSTRACT}, \text{REDACT}\}$, which we arrange as an ordinal hierarchy reflecting increasing privacy strength. Each detected sensitive span $e_i$ is assigned one of these actions, inducing a space of possible variants guided by human preferences for privacy sensitivity. The algorithm searches this preference-ordered space to identify the most privacy-preserving variant while ensuring that the

utility predicate yields an acceptable judgment. This construction provides the foundation for the formal definitions that follow.

**Action Space.** The action space is $A = \{\text{RETAIN, ABSTRACT, REDACT}\}$. These actions form an *ordinal lattice*, $\text{RETAIN} \prec \text{ABSTRACT} \prec \text{REDACT}$, encoding increasing privacy strength. The lattice is used to define one-step relaxations for the search procedure, and identify spans that cannot be modified without violating the utility constraint (Stage 1 of our search algorithm).

**Utility Predicate.** Let $y = \mathcal{F}(x)$ and $\tilde{y} = \mathcal{F}(\tau(x; \mathbf{a}))$. For open-ended tasks, placeholders/abstractions in $\tilde{y}$ are deterministically restored to $\tilde{y}^{\text{rb}}$ using the transformation map. A judge model then evaluates the pair $(y, \tilde{y}^{\text{rb}})$ under a fixed rubric to verify that the transformation does not degrade task performance, returning `pass` or `fail`. For tasks with fixed ground truths, utility is `pass` iff $\mathcal{F}(\tau(x; \mathbf{a}))$ is correct under the task's scoring rule (e.g., exact match or multiple-choice accuracy). The only criterion for accepting a candidate is the utility predicate UTIL returns `pass`.

We examined how sensitive the utility predicate is to small relaxations of the threshold $\gamma$. To test whether users can perceive such utility reductions, we conducted a user study (see Appendix F) comparing outputs produced under different $\gamma$ settings. The results show that even minor relaxations of $\gamma$ lead to noticeable quality degradation from a user's perspective. This supports our choice of a strict pass-fail utility predicate that requires preserving the original utility without degradation.

**Privacy Comparator.** To define a structured search space over privacy transformations, we introduce a pairwise privacy comparator $\mathcal{C} : (x, \tau_A, \tau_B) \longmapsto \{\tau_A, \tau_B, \text{SAME}\}$. Given two variants of the *same* source message, it returns which is more privacy-preserving (or SAME).

Unlike a partial order, this relation is not assumed to be transitive or total, reflecting the empirical reality that human privacy preferences may exhibit intransitivities or context-dependent judgments. Our algorithm leverages this relation as an ordering signal, treating it as an oracle for guided search without requiring formal lattice properties.

# 4 ALGORITHM AND IMPLEMENTATION

This section presents both the algorithmic procedure and the practical implementation of our framework. The algorithm specifies a two-stage search over the privacy-ordered action space, and the implementation focuses on instantiating the privacy comparator to align with human preferences. Together, they define the end-to-end system used in our experiments.

## 4.1 ALGORITHM: FREEZE-THEN-SEARCH

**Stage 1: Freeze Inflexible Entities.** For each $e \in D$, probe REDACT$(e)$ and ABSTRACT$(e)$ in isolation while keeping all other entities RETAIN. If both probes cause utility to fail, mark $e$ as *frozen* (forced RETAIN thereafter). Let $D' \subseteq D$ be the non-frozen entities with $n' = |D'|$; only $D'$ participates in Stage 2. This step both preserves utility invariants and reduces the branching factor.

**Stage 2: Privacy-Comparator Priority-Queue Tree Search.** The tree search begins at a root node obtained by applying to each $e \in D'$ the most privacy-preserving transformation allowed by Stage 1. Each node encodes a transformation action vector $a$ and its corresponding transformed message $\tau(x; a)$. For any nodes, child nodes are generated by relaxing exactly one action (i.e., REDACT $\rightarrow$ ABSTRACT; ABSTRACT $\rightarrow$ RETAIN). The tree is traversed in order of decreasing privacy, guided by a priority queue that uses $\mathcal{C}$ as the comparator. Ties (SAME) are broken by stable insertion order. The complete search procedure is given in Algorithm 1.

The procedure returns the *first* action profile $\mathbf{a}$ that satisfies the utility predicate. We record (i) the transformed input $\tau(x; \mathbf{a})$; (ii) the Stage 1 freeze set $D'$ (entities forced to RETAIN); (iii) the per-entity action map. If no candidate passes, we return $\text{RETAIN}^{|D|}$.

**Complexity.** Stage 2 explores at most $|\mathcal{M}| = 3^{n'}$ action profiles on the non-frozen coordinates. If $T$ candidates are expanded, a binary-heap implementation requires at most $C \leq cT \log T$ pairwise comparisons (many avoided by caching). With average per-call latencies $t_{\mathcal{C}}$ and $t_{\text{UTIL}}$ for comparator and utility respectively, Time $\lesssim cT \log T \cdot t_{\mathcal{C}} + T \cdot t_{\text{util}}$.

---

**Algorithm 1:** Privacy-Comparator Priority Queue Tree Search (Stage 2)

---

**Input:** message $x$; non-frozen entities $D'$; utility predicate $U$; comparator $\mathcal{C}$
**Output:** first passing action profile $\mathbf{a}$

1 Initialize $\mathbf{a}_0$: for $e \in D'$, set REDACT unless it failed in Stage 1 (then ABSTRACT); for $e \notin D'$,
   set RETAIN;
2 $Q \leftarrow$ comparator-based priority queue seeded with $\mathbf{a}_0$ (ties: stable order);
3 $V \leftarrow \emptyset$; // visited
4 **while** $Q$ *not empty* **do**
5    $\mathbf{a} \leftarrow Q.\text{pop}()$; **if** $\mathbf{a} \in V$ **then**
6       | continue
7    $V \leftarrow V \cup \{\mathbf{a}\}$;
8    **if** $U\big(\mathcal{F}(x), \mathcal{R}(\mathcal{F}(\tau(x;\mathbf{a}));\mathbf{a})\big) = pass$ **then**
9       | **return** $\mathbf{a}$
10    **foreach** $e \in D'$ *with* $a_e \in \{$REDACT, ABSTRACT$\}$ **do**
11       $\mathbf{a}' \leftarrow$ degrade $a_e$ by one step (REDACT$\rightarrow$ABSTRACT or ABSTRACT$\rightarrow$RETAIN);
12       **if** $\mathbf{a}' \notin V$ **then**
13          | push $\mathbf{a}'$ into $Q$
14 **return** RETAIN$^{|D|}$ // fallback

---

## 4.2 IMPLEMENTATION

**Privacy Transformations and Utility Check.** For each prompt we fix detected PII spans $D$ and a per-entity variants map (e.g., New York City and NYC) detected and clustered by GPT-4o; identical REDACT/ABSTRACT mappings and GPT-4o-generated abstractions are used across all models (App. D). We implement the span-level privacy transformation actions with a deterministic rewriter that (i) applies per-entity actions $a_i \in \{$RETAIN, ABSTRACT, REDACT$\}$ to produce $\tau(x;\mathbf{a})$ and a replacement map, and (ii) performs strict replace-back on model outputs for evaluation (Sec. 3.2). For utility, GPT-4o acts as judge (App. E): fixed-ground-truth tasks use the task's official scorer on $\mathcal{F}(\tau(x;\mathbf{a}))$; open-ended tasks are judged once on $(y, \text{restore}(\tilde{y}))$; single-answer QA runs $k=5$ independent decodes with early stop at the first mismatch, passing only if all $k$ are correct.

**Privacy Comparator.** We collect human ground-truth labels on 150 A/B pairs sampled from a PII-rich subset of the ShareGPT dataset (RyokoAI, 2023), with each pair annotated by at least five annotators. Independently, we create 4,840 additional pairs and obtain teacher labels from a strong zero-shot judge (OpenAI o3) for supervised LoRA finetuning (Hu et al., 2022), resulting in a latency-optimized comparator (finetuned Qwen2.5-7B-Instruct; hyperparameters in App. B) Compared with the human labels, the distilled comparator achieves **71%** overall and **89%** in high-human-consensus items ($\geq 0.8$) at **0.31**s/decision—yielding a $> 20\times$ speedup vs. the zero-shot judges with comparable high-consensus accuracy (Table 1). This choice materially reduces the $c\, T \log T \cdot t_\mathcal{C}$ term in §14 and enables practical Stage 2 search. Consensus among the 150 human-labeled pairs varies substantially: 73 items reach consensus $\geq 0.8$ and 121 reach $\geq 0.6$, with only a small subset achieving full agreement. Comparator accuracy improves with higher consensus, rising from 71% overall to 77% at $\geq 0.6$ and 89% at $\geq 0.8$.

| Comparator | Accuracy (All) | Acc. @ consensus $\geq 0.8$ | Latency (s) |
|---|---|---|---|
| o1 (zero-shot) | 70% | 90% | 8.05 |
| o3 (zero-shot) | 70% | 89% | 6.37 |
| o3-mini (zero-shot) | 69% | 88% | 4.32 |
| Qwen2.5-7B-Instruct (finetuned) | 71% | 89% | 0.31 |

Table 1: Privacy comparator alignment with human judgments and per-decision latency.

**Utility Evaluator.** For open-ended tasks, where utility cannot be measured deterministically, we validate GPT-4o as the utility judge using samples drawn from the oracle's search trace. We constructed 150 evaluation pairs, balanced between GPT-4o PASS and FAIL decisions, and collected judgments from 75 human annotators (five per item) on whether utility was preserved (ACCEPT) or

degraded (REJECT). Agreement between GPT-4o and humans increases with consensus strength, rising from approximately 0.69 at a 0.6 consensus threshold to approximately 0.94 under full agreement. This pattern parallels that of the privacy comparator and supports GPT-4o's reliability in the high-consensus regime where the utility predicate is most informative.

## 5 EXPERIMENTAL DETAILS

### 5.1 DATASETS AND PREFILTER

We sample test prompts from four datasets spanning open-ended and closed-ended tasks: ShareGPT (RyokoAI, 2023) (open-ended; 176 messages), WildChat (Zhao et al., 2024) (open-ended; 139), MedQA (Jin et al., 2020) (medical MCQ; 108), and CaseHOLD (Zheng et al., 2021) (legal MCQ; 110). All prompts contain PIIs (open-ended: $\geq 3$; close-ended: $\geq 1$).

For closed-ended datasets, we ensure that all test models can correctly answer the selected questions five times, so that any further accuracy drop can be attributed to reduced disclosure rather than intrinsic task difficulty. Open-ended datasets are prefiltered to only include PII-rich English text with a clear task. Detailed curation criteria are given in App. C.

### 5.2 MODEL SELECTION

We evaluate *nine* target models: *gpt-4.1-nano*, *gpt-4.1*, *gpt-5*, *claude-3-7-sonnet-20250219* (extended thinking disabled), *claude-sonnet-4-20250514* (extended thinking disabled), *lgai/exaone-deep-32b*, *mistralai/mistral-small-3.1-24b-instruct*, *qwen/qwen2.5-7b-instruct*, and *qwen/qwen2.5-0.5b-instruct*. This set covers a wide range of capacity model families, from frontier closed-weight models to small, open models suitable for on-device deployment. Two targets expose *reasoning modes* and are run with their default settings: *gpt-5* (default reasoning profile; `reasoning_effort=medium`) and *lgai_exaone-deep-32b* (provider default reasoning mode). All other targets are instruction-tuned chat models.

### 5.3 EXPERIMENT I: ESTABLISHING DATA MINIMIZATION ORACLES

We applied our framework to search data minimization, using the nine target models as the response-generation model $\mathcal{F}$ on prompts sampled from the four datasets. We report data minimization results as the optimal percentage of REDACT/ABSTRACT/RETAIN actions under the utility constraint.

To verify that minimization *robustly* reduces recoverability of masked information (redacted or abstracted) from the message itself, we run two black-box adversarial audits that attempt to simulate *on-text* inference by an adversary (Staab et al., 2024). **Type-wise recovery:** Given the text and the *set of types that were marked during minimization*, the attacker must output up to three *verbatim* candidates per requested type with confidences, relying only on the given text. We evaluate the same attacker on both the original input $x$ and the minimized input $\tilde{x}$ with an identical type set. For each type, we compute Hit@1/Hit@3 against the corresponding gold strings. **Span-wise recovery:** Given the minimized text $\tilde{x}$ and the list of replacement strings actually inserted by our pipeline (e.g., [NAME1] or abstraction phrases), the attacker must, for *each* span, return a single guess of its original string or ``Unknown'' with confidence 0 if it cannot be recovered from this message alone. We use two LLMs different from the nine target test models as attackers: one open-weight model (`meta-llama/llama-3.1-70b-instruct`) and one closed-weight model (`google/gemini-flash-1.5`).

### 5.4 EXPERIMENT II: BENCHMARKING ZERO-SHOT LLM DATA MINIMIZATION PREDICTORS

With the oracles in place, we evaluate the selected models in the *prediction* setting: given an input, the model must directly choose an action from {RETAIN, ABSTRACT, REDACT} for each detected span to produce the most privacy-preserving variant while preserving utility, **without comparator guidance, search, or any in-loop utility judge**.

The prompt provides the message, span types, span variants, and the replacement strings that would be applied if chosen. We parse the model output into an action map; invalid actions are repaired with a schema-only prompt, and undecided spans are marked and excluded from conditioned ratios.

For each item $i$ and predictor model $m$, we pair the oracle minimized prompt $\tilde{x}_i^\star$ with the predicted one $\tilde{x}_i^{(m)}$ to evelute with the same **pairwise privacy comparator** and **utility predicate** as in the search process. We classify (item, $m$) into four disjoint categories: *Overshare* if prediction leaks more privacy than oracle), *Undershare+Fail* if prediction is more protective but fails utility, *Undershare+Pass* if prediction is more protective and passes utility. *Fit* if prediction ties the oracle on privacy and passes utility. The first two categories are considered unsuccessful minimization, whereas the latter two represent successful minimization.

## 6 RESULTS

### 6.1 DATA MINIMIZATION ORACLE

Our minimization oracles show frontier models achieve the most privacy protection without violating the utility constraint (Table 2). On open-ended task prompts, *gpt-5* achieves the most aggressive removal—**85.7%** REDACT and **8.6%** ABSTRACT (only 5.7% RETAIN)—while the smallest model (*qwen2.5-0.5b*) sits at the bottom with **19.3%** REDACT and **11.0%** ABSTRACT (69.7% RETAIN). Closed-ended tasks admit even more minimization: *gpt-4.1* tops the board at **98.0%** REDACT and **1.0%** ABSTRACT (1.0% RETAIN), whereas *qwen2.5-0.5b* again trails with **32.1%** REDACT and **11.7%** ABSTRACT. The scatterplot in Fig. 2 shows frontier models clustered near the $x+y=1$ band, confirming that very little PII must be retained to preserve utility.

Overall, minimization is *redaction-heavy*: abstraction stays small (typically 1–12%), indicating that simply deleting sensitive spans is usually sufficient for the utility constraint. Smaller models accept far less minimization in both settings, which is acceptable in practice because they are more feasible to be deployed on-device, posing lower leakage risks. A cross-model Jaccard analysis (App. H) further shows that, despite differences in the exact minimized prompts, redaction decisions are highly consistent across model families. The majority of cross-model variation arises instead from the much smaller abstraction set, which both explains the larger fluctuations observed in abstraction and suggests that the core redactions transfer well across models.

| Response Generation Model | Open-ended | | | Closed-ended | | |
|---|---|---|---|---|---|---|
| | Redact ↑ | Abstract ↑ | Retain ↓ | Redact ↑ | Abstract ↑ | Retain ↓ |
| *gpt-5* | **85.7%** | **8.6%** | **5.7%** | 97.1% | 1.8% | 1.1% |
| *gpt-4.1* | 82.6% | 9.9% | 7.6% | **98.0%** | **1.0%** | **1.0%** |
| *gpt-4.1-nano* | 79.6% | 10.0% | 10.5% | 91.3% | 2.0% | 6.7% |
| *claude-sonnet-4-20250514*[†] | 74.8% | 11.2% | 14.0% | 97.2% | 1.9% | 0.9% |
| *claude-3-7-sonnet-20250219*[†] | 77.5% | 10.6% | 11.9% | 79.5% | 10.1% | 10.4% |
| *lgai_exaone-deep-32b* | 60.4% | 17.4% | 22.2% | 75.0% | 10.2% | 14.7% |
| *mistral-small-3.1-24b-instruct* | 75.3% | 12.5% | 12.2% | 96.4% | 1.7% | 1.9% |
| *qwen2.5-7b-instruct* | 69.9% | 12.0% | 18.1% | 91.7% | 4.6% | 3.7% |
| *qwen2.5-0.5b-instruct* | 19.3% | 11.0% | 69.7% | 32.1% | 11.7% | 56.2% |

Table 2: Optimal percentage of REDACT, ABSTRACT, and RETAIN actions for open-ended (ShareGPT, WildChat) and closed-ended (MedQA, CaseHold) task prompts across nine models. ↑ indicates that higher is better, and ↓ indicates that lower is better. [†] Extended thinking disabled.

**Span-wise Recovery.** Pooling across target models and grouping spans by action (Table 3), **abstraction** consistently yields higher overall recovery than **redaction** on every dataset: the correct-recovery rate $p_{\text{corr}}$ ranges **5.6–14.9%** for ABSTRACT versus only **2.7–7.7%** for REDACT. Importantly, the *absolute* rates are low across the board (all $p_{\text{corr}} < 0.15$, with REDACT $\leq 0.077$), indicating that on-text inference

Table 3: Span-wise recovery pooled across target models: $p_{\text{corr}}$ by action across (rows) datasets (columns).

| Action | CaseHOLD | MedQA | ShareGPT | WildChat |
|---|---|---|---|---|
| abstract | 0.092 | 0.056 | 0.149 | 0.119 |
| redact | 0.050 | 0.027 | 0.051 | 0.077 |

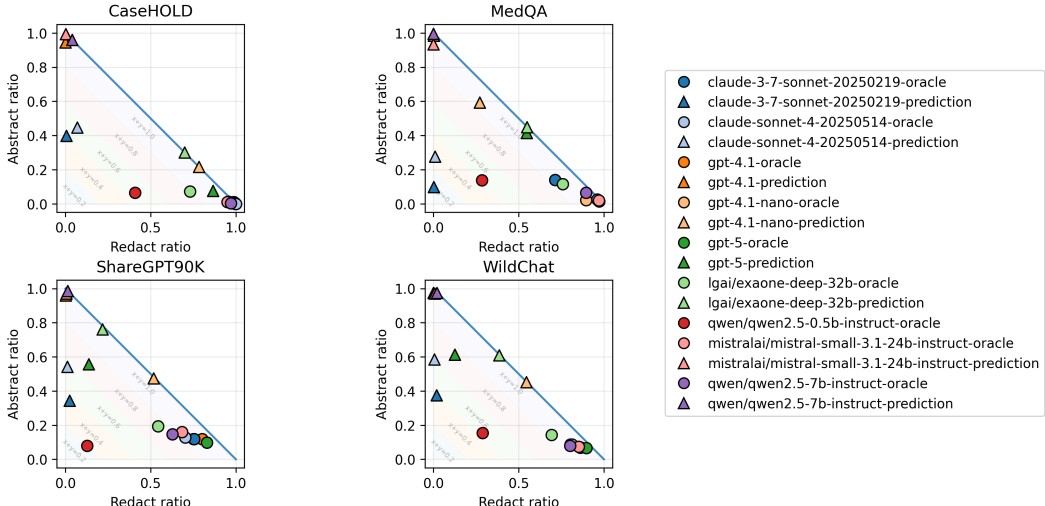

Figure 2: Oracle vs. Prediction REDACT and ABSTRACT Ratio.

is generally difficult under our setup. The separation is larger on open-ended data than on closed-ended data, suggesting that open-domain context leaves more clues. Redaction is more robust to on-text inference than abstraction—attackers both *attempt less* and *succeed less* after REDACT—and overall recovery remains low, reinforcing a *redact-first* policy when minimizing leakage, especially for open-ended inputs.

A parallel span-level evaluation with GPT-5 as the attacker on its own leads to the same conclusion. Across datasets, correct-recovery rates remain low for both abstraction and redaction spans, and masked spans are overwhelmingly labeled as UNKNOWN. Full results are reported in Appendix I.2. Together, these findings show that GPT-5 is unable to reconstruct the removed private information even when attacking its own oracle-minimized prompts.

**Type-wise Recovery (original vs. masked).** Aggregating by entity type, masking causes a sharp drop in recoverability relative to the original text. For example on WILDCHAT (Hit@1, %), NAME falls from 90.3 to 0.0, GEOLOCATION from 89.8 to 2.2, OCCUPATION from 85.4 to 8.0, and AF-FILIATION from 83.0 to 1.9; other datasets show the same pattern (Appendix I.1). Hit@3 mirrors these trends across types. In short, masking severely limits type-wise recovery. Consistent with the span-level results, a parallel test using GPT-5 as the attacker on its own minimized prompts shows the same pattern: masked Hit@1 for every type stays in the low single digits while the corresponding original values are often near the top of the scale, indicating that GPT-5 does not infer the removed PII.

Taken together, the span-wise and type-wise recovery checks confirm that our search-based data minimization method effectively strips sensitive information from prompts and prevents that information from being inferred indirectly from the remaining context.

## 6.2 PREDICTION VS. ORACLE

As shown in Fig. 3, single-pass predictions are generally less privacy-preserving than the `gpt-5` oracle—*Overshare* dominates across tasks—indicating that these direct predictions without comparator-guided search tend to under-protect privacy with frontier models which are most widely used and vulnerable to more privacy risks. Items counted as *Undershare+FAIL* reflect attempts to push masking beyond the oracle that break task utility. A meaningful slice—especially on open-ended datasets—falls into *Undershare+PASS*, signaling headroom to further tighten the oracle's comparator priorities or stop rule. The *Fit* mass (privacy tie + utility pass) is small, suggesting the prediction rarely sits close to a task-wise privacy/utility frontier. Oracles are harder to surpass in the close-ended, answer-verifiable tasks (MedQA is near-all Overshare, while CaseHOLD still

shows non-trivial Undershare+PASS and Fit). Minor stochasticity in `gpt-5` decoding is mitigated via replace-back, and $k=5$ repetition on verifiable tasks.

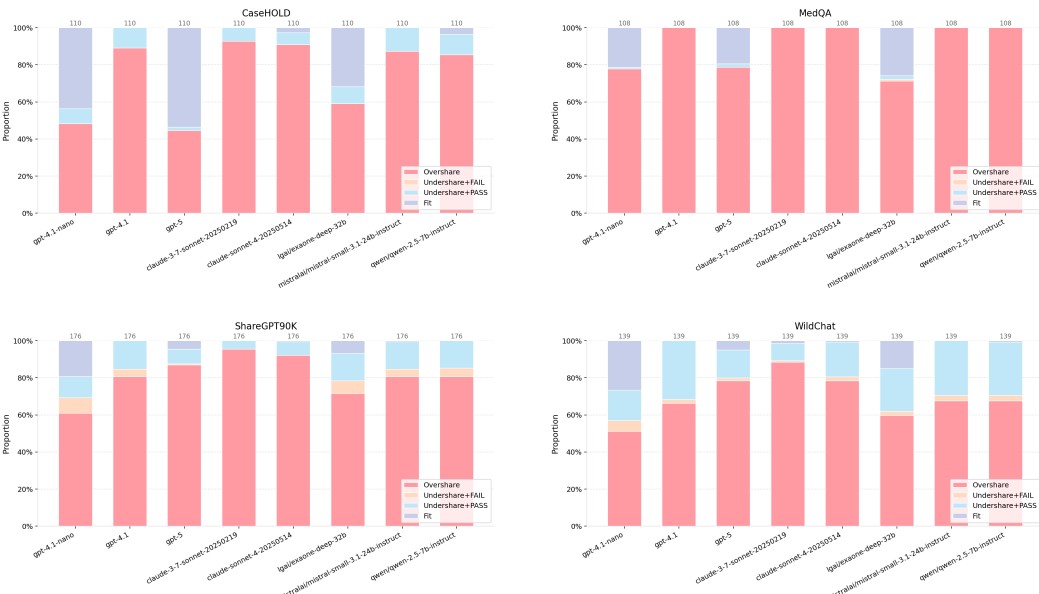

Figure 3: **Prediction vs. oracle minimization across datasets.** Each panel shows per-model stacked proportions that sum to 1. Outcomes are interpreted *relative to the `gpt-5` oracle* using our privacy comparator (Sec. 3.2) and utility predicate (Sec. 5): *Overshare*—the prediction disclosure is *less privacy-preserving* than the oracle; *Undershare+FAIL*—the prediction hides more but fails the utility check; *Undershare+PASS*—the prediction hides more and passes utility; and *Fit*—the prediction ties the oracle on privacy and passes utility.

**Prediction bias toward ABSTRACT.** In single-pass predictions, models consistently favor AB-STRACT over REDACT, showing an *abstraction-first* default on everyday user prompts (e.g., trip planning). Because ABSTRACT is less privacy-preserving in our setup, choosing it when REDACT would still retain utility implies unnecessary disclosure. This tendency persists even when instructions explicitly indicate that the protection strengths of REDACT is higher than ABSTRACT; and it contrasts with the oracles that cluster in the high-REDACT/low-ABSTRACT regime (cf. Fig. 2). We also tested whether the abstraction preference arises from our prompt design by ablating the minimization-order instruction. As detailed in App. G.4, removing either the "prefer stronger" clause or the entire minimization order line leaves model behavior nearly unchanged: all the selected models for this further test (GPT-5, Mistral-24B, and Qwen2.5-7B) still strongly prefer ABSTRACT, indicating that the bias is model-internal rather than prompt-induced.

**Ablation by model family.** Results show stable, high-level biases as illustrated by the prediction-side clusters in Fig. 2. **Mistral/Qwen/GPT-4.1** default to an *abstract-first* policy across datasets—even for *structured* identifiers—e.g., on ShareGPT and WildChat they abstract nearly all URL/EMAIL/ID_NUMBER spans with $\leq$1–2% redact, with slightly lower abstraction rates for contextual attributes such as GEOLOCATION/TIME, where abstraction remains predominant. **Claude** adds a pronounced RETAIN tail on open-ended prompts (large fractions of GEOLOCATION, TIME, AFFILIATION kept), with little redaction. By contrast, the two reasoning models **GPT-5** and **Exaone** are the only ones that *consistently redact* high-precision types: on closed-ended Case-HOLD/MedQA they heavily redact NAME/TIME/GEOLOCATION, and on open-ended chats they are far more willing than other families to redact URL/EMAIL/PHONE_NUMBER.

For completeness, we also observe that fully masking all detected PIIs, as would occur in a simple NER-based redaction, often breaks utility. Together with the oversharing behavior of single-pass predictions, this suggests that neither extreme is adequate, and an oracle is needed to determine how much masking each model can tolerate.

## 7 CONCLUSION AND DISCUSSION

We present a framework that formally defines and operationalizes data minimization in LLM prompting: for a given user prompt and response model, it quantifies the minimal privacy-revealing disclosure required to maintain utility. Our results show that data minimization offers a significant optimization space for reducing privacy exposure without compromising task performance, particularly for larger and more capable language models. However, we find that directly predicting this minimal disclosure is challenging, even for frontier models. This work lays the groundwork for research on quantifying data minimization and robust prediction methods, fostering both fundamental machine learning advances and interdisciplinary research in human-AI interaction.

**Novel Paradigm of Privacy-Preserving LLM Interactions.** We show that the more capable the model is, the more feasible data minimization becomes. This result shows that data minimization is a promising approach to addressing excessive disclosure problems in user interactions with LLM systems, as users tend to trade privacy for utility and therefore often choose frontier models hosted on the cloud for sensitive tasks despite privacy concerns (Zhang et al., 2024). The variances of data minimization across datasets and models suggest that model-specific predictors are needed, and we advocate that LLM providers include these as part of the released model package. Such predictors naturally align with an emerging line of work that explores a dual-model management approach: using small edge models for data-minimization-guided local sanitization before sharing data with the remote model (Li et al., 2025b; Zhou et al., 2025; Zhang et al., 2025; Chowdhury et al., 2025). Beyond these observations, our results also clarify the technical role of the oracle within this workflow. The oracle procedure identifies the upper bound of data minimization a target model can tolerate while preserving utility, providing high quality supervision for learning practical sanitization policies. This supervision can train or distill a small predictor that performs single pass span level decisions locally, complementing the dual model management approach described above. This establishes a natural path toward future on-device predictors that give users full control over the flow of private data before any interaction with a remote model.

**LLM Capabilities for Privacy Tasks.** We evaluate LLM capabilities on two novel privacy tasks: data minimization prediction and privacy sensitivity ranking (by the privacy comparator), extending prior work on using LLMs for PII detection and context-aware privacy judgments (Mireshghallah et al., 2024b; Shao et al., 2025; Li et al., 2025a). We find that data minimization prediction remains challenging for current state-of-the-art models. For the privacy sensitivity ranking task, we found that off-the-shelf reasoning models (e.g., GPT-o1, o3, and o3-mini) perform better than non-reasoning models (e.g., GPT-4o). Future research should further account for individual preference differences, as our results show that in over half of cases the five human raters reached a consensus score below 0.8. A failure case analysis of the best-performing models reveals where misalignment still occurs. In these cases, humans often choose "SAME," while models prefer "A" or "B," reflecting different thresholds for saliency: models overemphasize subtle distinctions that seem significant to them but are imperceptible or irrelevant to humans. Moreover, models tend to overvalue specificity and do not align with humans on how the specificity of certain data types corresponds to sensitivity (e.g., assigning more weight to time or date information than to names).

**Interpretation Methods for "What is Necessary."** Foundational understanding of what information or tokens are necessary is still required to explain the variance observed in data minimization oracles across models and datasets. Current methods can reveal what information is used at inference (Vig et al., 2020), but determining what is truly necessary remains an open research frontier. In addition, the potential impact of test set contamination (Oren et al., 2024) should be carefully taken into consideration in future investigations.

## ETHICS STATEMENT

All datasets are publicly available under their respective terms; we do not crawl private sources. All human-subjects studies have been approved by our institution's IRB. Our human evaluation collects no PII of the human raters. Annotators only state preferences over sanitized replacements; no demographics are recorded and no unanonymized content is shown.

REPRODUCIBILITY STATEMENT

We release our code at the project GitHub repository. The pipeline code used in the oracle experiment (Section 5.3) is in `run_pipeline.py`. We also release the trained privacy comparator as a LoRA adapter on HuggingFace, enabling end-to-end reproduction of our action-search pipeline. The folder `Prefiltered datasets` corresponds to the four test datasets described in Section 5.1. The file `human_annotation_vs_o3mini.jsonl` contains human annotator tallies (`tally`) and `o3mini` judgments used to (i) select the best teacher model, (ii) use the teacher to generate large input sets, and (iii) train the privacy comparator (Section 4.2); we also use the same human annotations to evaluate the comparator's accuracy. Setup notes, and example commands are provided in the repository README.

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

## A    LLM Usage

Outside of minor manuscript assistance, all LLM usage in this work was solely for running the experiments reported in the main text (PII detection/adjudication, abstraction-term synthesis, prediction runs, and adversarial audits), with model choices and procedures fully specified in Sec. 5

**Manuscript assistance.** We used LLMs only for light language polishing and LaTeX table formatting (e.g., caption phrasing, column alignment). No technical content or analysis was delegated, and all edits were author-verified.

## B PRIVACY COMPARATOR TRAINING HYPERPARAMETERS

**Qwen2.5-7B-Instruct (distilled final).** learning_rate $= 1e-4$; epochs $= 2$; LoRA rank $= 8$; context length $= 2048$; global batch size $= 2048$ (Fireworks AI). On human-annotated evals (consensus $\geq 0.8$), accuracy $= 89.04\%$.

## C DATASET CURATION

**Close-ended Dataset (MedQA and CaseHold)** Each item is answered by all nine target models with five runs at temperature $0$ on the original (unmasked) input; we keep an item only if *every* model is correct in *all five* runs. This ensures that any accuracy drop under minimization is attributable to masking rather than intrinsic task difficulty.

**Open-ended Dataset (ShareGPT and WildChat)** We curate a high-quality subset to study how far inputs can be minimized before utility degrades. Specifically, we (i) select PII-rich messages using Microsoft Presidio, requiring at least three *distinct* PII categories; (ii) retain primarily English content via fastText; (iii) remove near-duplicates using MiniLM embeddings; (iv) exclude code-heavy or prefixed web dumps and templated/meta prompts lacking self-contained intent; (v) discard prompts without a clear task intention; and (vi) validate PII triggers with a GPT-4o checker, confirming at least three *unique*, text-based entities per message. For WildChat, we additionally restrict to messages without pre-existing redactions. The final sets comprise over 500 ShareGPT messages and over 100 WildChat messages. The ShareGPT pool supports replacement-pair creation for the human study and comparator distillation, and both sets are later reused for pipeline operationalization.

## D DETECTION AND ABSTRACTION SYSTEM PROMPT

### D.1 DETECTION

You are an expert in cybersecurity and data privacy. You are now tasked to detect PII from the given text, using the following taxonomy only:

- `NAME`: Name
- `EMAIL`: Email address
- `PHONE_NUMBER`: Phone number
- `ID`: Identifiers, including ID Number, passport number, SSN, driver's license, taxpayer, identification number
- `ONLINE_IDENTITY`: IP address, username, URL, password, key
- `GEOLOCATION`: Places and locations and addresses, such as cities, provinces, countries, international regions, or named infrastructures (bus stops, bridges, etc.).
- `AFFILIATION`: Names of organizations, such as public and private companies, schools, universities, public institutions, prisons, healthcare institutions, non-governmental organizations, churches, etc.
- `DEMOGRAPHIC_ATTRIBUTE`: Demographic attributes of a person, such as native language, descent, heritage, ethnicity, nationality, religious or political group, birthmarks, ages, sexual orientation, gender and sex.
- `TIME`: Description of a specific date, time, or duration.
- `HEALTH_INFORMATION`: Details concerning an individual's health status, medical conditions, treatment records, and health insurance information.
- `FINANCIAL_INFORMATION`: Financial details such as bank account numbers, credit card numbers, investment records, salary information, and other financial statuses or activities.

- EDUCATIONAL_RECORD: Educational background details, including academic records, transcripts, degrees, and certification.

For the given message that a user sends to a chatbot, identify all the personally identifiable information using the above taxonomy only, and the entity_type should be selected from the all-caps categories. Note that the information should be related to a real person not in a public context, but okay if not uniquely identifiable. Result should be in its minimum possible unit.

Return *only* a JSON object in the following format: `"results": ["entity_type": YOU_DECIDE_THE_PII_TYPE, "text": PART_OF_MESSAGE_YOU_IDENTIFIED_AS_PII]`

## D.2 ABSTRACTION

Rewrite the text to abstract the protected information, without changing other parts. Please output in JSON format only.

*Input:*
```
<Text>I graduated from CMU, and I earn a six-figure salary.
Today a lady from China said...</Text>
<ProtectedInformation1>CMU</ProtectedInformation1>
<ProtectedInformation2>China</ProtectedInformation2>
```

*Output:*
```
{"results": [{"protected": "CMU", "abstracted": "a prestigious
American university"},
{"protected": "China", "abstracted": "A country in Asia"}]}
```

# E UTILITY CHECK

**Utility Evaluator System Prompt**   You are a utility evaluator. Your task is to assess whether a second response adequately addresses a user's original message. You will be provided with: (i) the user message (what the user is asking or talking about), (ii) a reference response A (a standard answer that is known to be good), and (iii) a candidate response B (a second version that needs to be evaluated). You must decide whether the candidate response is "PASS" or "FAIL". This is not a test of exact wording or stylistic similarity; focus on whether the candidate response addresses all key points or needs expressed in the user message. If it does, it passes. If it fails to address one or more key points, it fails.

Input will contain three tagged sections: `<user_message>`, `<response_A>`, and `<response_B>`.

Return JSON with the result and a one-line explanation: `"Result": "PASS/FAIL", "Reason": YOUR_EXPLANATION`

# F UTILITY THRESHOLD SENSITIVITY STUDY

To evaluate how sensitive our method is to the choice of the utility threshold $\gamma$ (§3.1) in the open-ended task setting, we performed an additional experiment on the ShareGPT and WildChat datasets. In the main paper, the utility predicate is binary because the evaluator (GPT-4o) returns categorical judgments. For this study, we modified the evaluator's instruction so that it produces a numerical utility score between 1 and 10, where the model's original response is defined as a score of 10. A candidate transformation passes the utility predicate if its score is at least $10 - \gamma$.

We evaluated three thresholds, $\gamma = 0, 1, 2$, corresponding to minimum required utility scores of 10, 9, and 8. All other components of the pipeline remained unchanged.

To determine whether the resulting quality differences are perceptible to users, we conducted a human evaluation. We recruited 90 English speaking participants located in the United States using

Prolific. For each of 90 sampled prompts, we generated pairwise comparisons between (i) the original response and (ii) a response produced under one of the $\gamma$ settings. Participants selected whether A, B, or SAME was of higher quality.

The results were as follows:

- $\gamma = 0$: SAME = 36.8%, original preferred = 32.6%, masked preferred = 30.5%

- $\gamma = 1$: SAME = 30.9%, original preferred = 42.5%, masked preferred = 26.6%

- $\gamma = 2$: SAME = 28.1%, original preferred = 44.1%, masked preferred = 27.8%

The trend is clear. As $\gamma$ increases, the proportion of SAME judgments decreases and preference for the original output increases. This shows that even small relaxations of the utility threshold produce user perceptible quality degradation. These findings justify the use of a strict binary predicate in 3.2, which prohibits any reduction in utility and therefore represents a conservative and safe choice for minimizing disclosure.

## G  PROMPT SPECIFICATIONS FOR ZERO-SHOT PREDICTION

### G.1  SMALL-MODEL FAILURE.

We mark the smallest-capacity model (Qwen2.5–0.5B) as N/A for the single-pass prediction due to persistent schema/IO failures: (i) emitting actions outside {RETAIN, ABSTRACT, REDACT}; (ii) using PII *types* as keys instead of entity IDs; and (iii) remaining unstable even with reduced prompts. Notably, when we prefill an all-RETAIN map *inside the prompt* to constrain decoding, the model either (a) echoes the seed map (no decision-making) verbatim or (b) still produces noncompliant/irrelevant text, failing to follow the output schema. Given this degenerate behavior, a single-pass prediction is not meaningful so we did not use this model as predictor in the experiments.

### G.2  SYSTEM PROMPTS

**Open-ended tasks (ShareGPT / WildChat).**  You are a PII Minimization Transformer. Return *one line*: a single JSON object (no markdown/prose). **Goal:** for each PII surface string in `pii_dict`, choose the most privacy-preserving action that keeps response quality for open-ended tasks. **Order:** RETAIN < ABSTRACT < REDACT (prefer stronger only if quality is unchanged). **Schema:** `"transformation":"<PII_KEY>":"retain|abstract|redact", ...,` `"explanation":"ONE SHORT SENTENCE, NO RAW PII"`.

**Rules:**  (1) Keys must be the exact surface strings from `pii_dict` (verbatim; do not add/remove/rename). (2) Values are exactly one of `retain|abstract|redact` (lowercase). (3) If an action is chosen, it applies to all variants in `variants_map`. (4) If the needed entry is missing in `abstract_map`/`redact_map`, default to `retain` for that key. (5) Keep key order aligned with `pii_dict` if possible.

**Deterministic tasks (MedQA / CaseHOLD).**  Same as (A) except the **Goal**: choose actions that still allow selecting the correct option in a multiple-choice setting. Use the same order, schema, and rules.

**Tiny-model strict prompt (values-only edits).** Return *one line* JSON only: `{"transformation":{...}}`.  Inputs:  `pii_dict` and `draft_transformation` where every key (verbatim from `pii_dict`) is preset to `"retain"`. **Edit only the values** to one of `retain|abstract|redact`. Do not add/remove/rename any key.

### G.3  USER PAYLOADS

**Regular user payload (with maps).**  Fields: `message` (original text), `pii_dict` ({surface: type}), `variants_map` ({surface: [aliases]}), `redact_map`/`abstract_map` (per-surface replacements).

"Edit values only to one of `retain|abstract|redact`; do not change keys. Return `{"transformation": <edited draft>}`."

### G.4 ABLATION OF MINIMIZATION-ORDER

To test whether the observed preference for ABSTRACT arises from our prompt design, we conducted two ablations of the system instruction used for single-pass prediction. The first variant (*order only*) removes the clause "prefer stronger only if quality is unchanged," while the second (*no order/notion*) removes the entire minimization-order line. We ran GPT–5, Mistral–24B, and Qwen2.5–7B on ShareGPT and MedQA using all three prompt types, keeping all other settings fixed. These three models were chosen to cover a representative range of capacities and training regimes, and ShareGPT (open-ended) and MedQA (closed-ended) serve as one exemplar dataset for each task type. As shown in Table 4, removing these instructions does not eliminate the strong preference for ABSTRACT; redaction increases marginally, but the overall pattern remains unchanged. This indicates that the abstraction bias is not prompt-induced but reflects a model-internal tendency toward fluency-preserving transformations.

| dataset | model | prompt_type | redact | abstract | retain | undecided |
|---|---|---|---|---|---|---|
| ShareGPT | gpt-5 | order+notion | 164 (13.8%) | 663 (55.7%) | 363 (30.5%) | 0 (0.0%) |
| ShareGPT | gpt-5 | order only | 225 (18.9%) | 656 (55.1%) | 309 (26.0%) | 0 (0.0%) |
| ShareGPT | gpt-5 | no order/notion | 153 (12.9%) | 728 (61.2%) | 309 (26.0%) | 0 (0.0%) |
| ShareGPT | mistral-small-24b | order+notion | 6 (0.5%) | 1155 (97.1%) | 29 (2.4%) | 0 (0.0%) |
| ShareGPT | mistral-small-24b | order only | 20 (1.7%) | 1070 (89.9%) | 100 (8.4%) | 0 (0.0%) |
| ShareGPT | mistral-small-24b | no order/notion | 20 (1.7%) | 1027 (86.3%) | 143 (12.0%) | 0 (0.0%) |
| ShareGPT | qwen2.5-7b | order+notion | 16 (1.3%) | 1174 (98.7%) | 0 (0.0%) | 0 (0.0%) |
| ShareGPT | qwen2.5-7b | order only | 38 (3.2%) | 1116 (93.8%) | 31 (2.6%) | 5 (0.4%) |
| ShareGPT | qwen2.5-7b | no order/notion | 139 (11.7%) | 1000 (84.0%) | 46 (3.9%) | 5 (0.4%) |
| MedQA | gpt-5 | order+notion | 533 (54.6%) | 405 (41.5%) | 38 (3.9%) | 0 (0.0%) |
| MedQA | gpt-5 | order only | 808 (82.8%) | 140 (14.3%) | 28 (2.9%) | 0 (0.0%) |
| MedQA | gpt-5 | no order/notion | 688 (70.5%) | 258 (26.4%) | 30 (3.1%) | 0 (0.0%) |
| MedQA | mistral-small-24b | order+notion | 0 (0.0%) | 912 (93.4%) | 64 (6.6%) | 0 (0.0%) |
| MedQA | mistral-small-24b | order only | 2 (0.2%) | 609 (62.4%) | 365 (37.4%) | 0 (0.0%) |
| MedQA | mistral-small-24b | no order/notion | 2 (0.2%) | 531 (54.4%) | 443 (45.4%) | 0 (0.0%) |
| MedQA | qwen2.5-7b | order+notion | 0 (0.0%) | 973 (99.7%) | 3 (0.3%) | 0 (0.0%) |
| MedQA | qwen2.5-7b | order only | 2 (0.2%) | 940 (96.3%) | 30 (3.1%) | 4 (0.4%) |
| MedQA | qwen2.5-7b | no order/notion | 19 (1.9%) | 928 (95.1%) | 28 (2.9%) | 1 (0.1%) |

Table 4: Ablation of minimization-order instructions across models and datasets (without total-pii column).

## H CROSS-MODEL DATA MINIMIZATION OVERLAP

This section details the cross-model overlap experiment from §6.1, which quantifies how similar data minimization decisions are across response generation models, and whether the minimal sufficient masking chosen by each model's oracle shows stable structure.

For each model and dataset, we compare the oracle-derived action map to the GPT-5 oracle (as reference), computing Jaccard overlap separately for REDACT and ABSTRACT. The overlap is the number of PII spans where both oracles choose the same action divided by the union of spans where either does, yielding model- and dataset-level consistency measures.

Tables 5 and 6 report the resulting overlaps. As summarized in §6.1, redaction overlap is high across most models and datasets, typically at or above eighty percent, and in some datasets such as CaseHOLD it reaches above ninety percent for nearly all models. These results indicate that the majority of removed spans form a shared core of non-essential sensitive information that models broadly agree upon. In contrast, abstraction overlap is lower but abstraction itself accounts for a much smaller fraction of actions, which makes its variability less consequential in practice. Taken together, these observations suggest that the essential privacy-preserving behavior, namely which spans can be safely removed while maintaining utility, generalizes well across model families even though the exact minimally sufficient prompts remain model specific by definition.

| Dataset | Model | Overlap |
|---------|-------|---------|
| ShareGPT | gpt-4.1-nano | 0.802493 |
| ShareGPT | gpt-4.1 | 0.845057 |
| ShareGPT | claude-3-7-sonnet-20250219 | 0.792776 |
| ShareGPT | claude-sonnet-4-20250514 | 0.754572 |
| ShareGPT | lgai/exaone-deep-32b | 0.583899 |
| ShareGPT | mistralai/mistral-small-3.1-24b-instruct | 0.743466 |
| ShareGPT | qwen/qwen2.5-7b-instruct | 0.686103 |
| ShareGPT | qwen/qwen2.5-0.5b-instruct | 0.145875 |
| WildChat | gpt-4.1-nano | 0.849807 |
| WildChat | gpt-4.1 | 0.860465 |
| WildChat | claude-3-7-sonnet-20250219 | 0.797419 |
| WildChat | claude-sonnet-4-20250514 | 0.800771 |
| WildChat | lgai/exaone-deep-32b | 0.701961 |
| WildChat | mistralai/mistral-small-3.1-24b-instruct | 0.853816 |
| WildChat | qwen/qwen2.5-7b-instruct | 0.794839 |
| WildChat | qwen/qwen2.5-0.5b-instruct | 0.298128 |
| MedQA | gpt-4.1-nano | 0.875905 |
| MedQA | gpt-4.1 | 0.950464 |
| MedQA | claude-3-7-sonnet-20250219 | 0.708768 |
| MedQA | claude-sonnet-4-20250514 | 0.952183 |
| MedQA | lgai/exaone-deep-32b | 0.762055 |
| MedQA | mistralai/mistral-small-3.1-24b-instruct | 0.954451 |
| MedQA | qwen/qwen2.5-7b-instruct | 0.889583 |
| MedQA | qwen/qwen2.5-0.5b-instruct | 0.292683 |
| CaseHOLD | gpt-4.1-nano | 0.945274 |
| CaseHOLD | gpt-4.1 | 0.985075 |
| CaseHOLD | claude-3-7-sonnet-20250219 | 0.987531 |
| CaseHOLD | claude-sonnet-4-20250514 | 0.987562 |
| CaseHOLD | lgai/exaone-deep-32b | 0.736181 |
| CaseHOLD | mistralai/mistral-small-3.1-24b-instruct | 0.937811 |
| CaseHOLD | qwen/qwen2.5-7b-instruct | 0.957711 |
| CaseHOLD | qwen/qwen2.5-0.5b-instruct | 0.395522 |

Table 5: Cross-Model Redaction Overlap with the GPT-5 Oracle

| Dataset | Model | Overlap |
|---------|-------|---------|
| ShareGPT | gpt-4.1-nano | 0.203704 |
| ShareGPT | gpt-4.1 | 0.256039 |
| ShareGPT | claude-3-7-sonnet-20250219 | 0.182648 |
| ShareGPT | claude-sonnet-4-20250514 | 0.178261 |
| ShareGPT | lgai/exaone-deep-32b | 0.090625 |
| ShareGPT | mistralai/mistral-small-3.1-24b-instruct | 0.189189 |
| ShareGPT | qwen/qwen2.5-7b-instruct | 0.152941 |
| ShareGPT | qwen/qwen2.5-0.5b-instruct | 0.014354 |
| WildChat | gpt-4.1-nano | 0.141414 |
| WildChat | gpt-4.1 | 0.132653 |
| WildChat | claude-3-7-sonnet-20250219 | 0.114035 |
| WildChat | claude-sonnet-4-20250514 | 0.085470 |
| WildChat | lgai/exaone-deep-32b | 0.108974 |
| WildChat | mistralai/mistral-small-3.1-24b-instruct | 0.126214 |
| WildChat | qwen/qwen2.5-7b-instruct | 0.090909 |
| WildChat | qwen/qwen2.5-0.5b-instruct | 0.052326 |
| MedQA | gpt-4.1-nano | 0.023810 |
| MedQA | gpt-4.1 | 0.093750 |
| MedQA | claude-3-7-sonnet-20250219 | 0.060403 |
| MedQA | claude-sonnet-4-20250514 | 0.175000 |
| MedQA | lgai/exaone-deep-32b | 0.031008 |
| MedQA | mistralai/mistral-small-3.1-24b-instruct | 0.111111 |
| MedQA | qwen/qwen2.5-7b-instruct | 0.090909 |
| MedQA | qwen/qwen2.5-0.5b-instruct | 0.026316 |
| CaseHOLD | gpt-4.1-nano | 0.000000 |
| CaseHOLD | gpt-4.1 | 0.000000 |
| CaseHOLD | claude-3-7-sonnet-20250219 | 0.200000 |
| CaseHOLD | claude-sonnet-4-20250514 | 0.000000 |
| CaseHOLD | lgai/exaone-deep-32b | 0.000000 |
| CaseHOLD | mistralai/mistral-small-3.1-24b-instruct | 0.000000 |
| CaseHOLD | qwen/qwen2.5-7b-instruct | 0.000000 |
| CaseHOLD | qwen/qwen2.5-0.5b-instruct | 0.000000 |

Table 6: Cross-Model Abstraction Overlap with the GPT-5 Oracle

# I PRIVACY AUDIT

## I.1 POOLED PRIVACY AUDIT ACROSS ORACLE-MINIMIZED PROMPTS - GOOGLE/GEMINI-FLASH-1.5 & META-LLAMA/LLAMA-3.1-70B-INSTRUCT AS ATTACKERS

| action | $N$ | $p_{corr}$ | $p_{corr,lo}$ | $p_{corr,hi}$ | $p_{unk}$ | $p_{unk,lo}$ | $p_{unk,hi}$ | $\overline{conf}$ |
|--------|-----|-----------|--------------|--------------|-----------|-------------|-------------|------|
| abstract | 679 | 0.119 | 0.097 | 0.146 | 0.323 | 0.288 | 0.359 | 0.627 |
| redact | 5627 | 0.077 | 0.070 | 0.084 | 0.762 | 0.750 | 0.773 | 0.175 |

Table 7: Span-wise recovery pooled across models by action on WildChat

| action | $N$ | $p_{corr}$ | $p_{corr,lo}$ | $p_{corr,hi}$ | $p_{unk}$ | $p_{unk,lo}$ | $p_{unk,hi}$ | $\overline{conf}$ |
|---|---|---|---|---|---|---|---|---|
| abstract | 1376 | 0.149 | 0.131 | 0.169 | 0.310 | 0.286 | 0.335 | 0.630 |
| redact | 6929 | 0.051 | 0.046 | 0.056 | 0.803 | 0.793 | 0.812 | 0.138 |

Table 8: Span-wise recovery pooled across models by action on ShareGPT

| action | $N$ | $p_{corr}$ | $p_{corr,lo}$ | $p_{corr,hi}$ | $p_{unk}$ | $p_{unk,lo}$ | $p_{unk,hi}$ | $\overline{conf}$ |
|---|---|---|---|---|---|---|---|---|
| abstract | 142 | 0.092 | 0.054 | 0.150 | 0.338 | 0.265 | 0.419 | 0.613 |
| redact | 6430 | 0.050 | 0.045 | 0.056 | 0.731 | 0.720 | 0.742 | 0.190 |

Table 9: Span-wise recovery pooled across models by action on CaseHOLD

| action | $N$ | $p_{corr}$ | $p_{corr,lo}$ | $p_{corr,hi}$ | $p_{unk}$ | $p_{unk,lo}$ | $p_{unk,hi}$ | $\overline{conf}$ |
|---|---|---|---|---|---|---|---|---|
| abstract | 935 | 0.056 | 0.043 | 0.072 | 0.030 | 0.021 | 0.043 | 0.964 |
| redact | 12835 | 0.027 | 0.024 | 0.030 | 0.790 | 0.783 | 0.797 | 0.158 |

Table 10: Span-wise recovery pooled across models by action on MedQA

| Type | Hit@1 (orig) | Hit@1 (mask) | Hit@3 (orig) | Hit@3 (mask) |
|---|---|---|---|---|
| ADDRESS | 1.000 | 0.000 | 1.000 | 0.000 |
| AFFILIATION | 0.684 | 0.017 | 0.725 | 0.020 |
| AGE | 1.000 | 0.000 | 1.000 | 0.000 |
| ETHNICITY | 0.000 | 0.000 | 0.000 | 0.000 |
| FINANCIAL_INFORMATION | 0.000 | 0.000 | 0.000 | 0.000 |
| GEOLOCATION | 0.759 | 0.140 | 0.788 | 0.157 |
| HEALTH_INFORMATION | 0.500 | 0.000 | 0.500 | 0.000 |
| INCOME | 1.000 | 0.000 | 1.000 | 0.000 |
| NAME | 0.915 | 0.000 | 0.997 | 0.000 |
| RACE | 0.735 | 0.000 | 0.735 | 0.000 |
| TIME | 0.870 | 0.004 | 0.916 | 0.004 |

Table 11: Type-wise recovery pooled by type on CaseHOLD

| Type | $N$ | H@1 CI | H@1 CI$^\sim$ | H@3 CI | H@3 CI$^\sim$ | $\overline{conf}$ | $\overline{conf}^\sim$ |
|---|---|---|---|---|---|---|---|
| ADDRESS | 8 | [67.6%, 100.0%] | [0.0%, 32.4%] | [67.6%, 100.0%] | [0.0%, 32.4%] | 1.000000 | 1.000000 |
| AFFILIATION | 459 | [64.0%, 72.5%] | [0.9%, 3.4%] | [68.3%, 76.4%] | [1.0%, 3.7%] | 0.869000 | 0.800000 |
| AGE | 8 | [67.6%, 100.0%] | [0.0%, 32.4%] | [67.6%, 100.0%] | [0.0%, 32.4%] | 1.000000 | 0.000000 |
| ETHNICITY | 9 | [0.0%, 29.9%] | [0.0%, 29.9%] | [0.0%, 29.9%] | [0.0%, 29.9%] | 0.667000 | 0.067000 |
| FINANCIAL_INFORMATION | 9 | [0.0%, 29.9%] | [0.0%, 29.9%] | [0.0%, 29.9%] | [0.0%, 29.9%] | 1.000000 | 0.000000 |
| GEOLOCATION | 344 | [71.1%, 80.1%] | [10.7%, 18.0%] | [74.2%, 82.8%] | [12.2%, 19.9%] | 0.830000 | 0.530000 |
| HEALTH_INFORMATION | 8 | [21.5%, 78.5%] | [0.0%, 32.4%] | [21.5%, 78.5%] | [0.0%, 32.4%] | 1.000000 | 1.000000 |
| INCOME | 7 | [64.6%, 100.0%] | [0.0%, 35.4%] | [64.6%, 100.0%] | [0.0%, 35.4%] | 1.000000 | 0.257000 |
| NAME | 377 | [88.3%, 93.9%] | [0.0%, 1.0%] | [98.5%, 100.0%] | [0.0%, 1.0%] | 0.999000 | 0.601000 |
| RACE | 34 | [56.9%, 85.4%] | [0.0%, 10.2%] | [56.9%, 85.4%] | [0.0%, 10.2%] | 0.735000 | 0.500000 |
| TIME | 238 | [82.1%, 90.7%] | [0.1%, 2.3%] | [87.4%, 94.5%] | [0.1%, 2.3%] | 0.954000 | 0.633000 |

Table 12: Type-wise recovery on casehold: 95% confidence intervals (H@1/H@3; original and minimized) and mean top-1 confidence.

| Type | Hit@1 (orig) | Hit@1 (mask) | Hit@3 (orig) | Hit@3 (mask) |
|---|---|---|---|---|
| AFFILIATION | 0.000 | 0.000 | 0.000 | 0.000 |
| AGE | 0.990 | 0.000 | 0.994 | 0.000 |
| DIETARY_PREFERENCE | 0.111 | 0.000 | 0.111 | 0.000 |
| GENDER | 1.000 | 0.120 | 1.000 | 0.120 |
| GEOLOCATION | 0.704 | 0.000 | 0.704 | 0.000 |
| HEALTH_INFORMATION | 0.919 | 0.031 | 0.970 | 0.048 |
| MARITAL_STATUS | 1.000 | 0.000 | 1.000 | 0.000 |
| OCCUPATION | 0.768 | 0.000 | 0.768 | 0.000 |
| RACE | 1.000 | 0.015 | 1.000 | 0.015 |
| SEXUAL_ORIENTATION | 0.375 | 0.000 | 0.375 | 0.000 |
| TIME | 0.541 | 0.000 | 0.824 | 0.000 |

Table 13: Type-wise recovery pooled by type on MedQA

| Type | $N$ | H@1 CI | H@1 CI$^{\sim}$ | H@3 CI | H@3 CI$^{\sim}$ | $\overline{\text{conf}}$ | $\overline{\text{conf}}^{\sim}$ |
|---|---|---|---|---|---|---|---|
| AFFILIATION | 9 | [0.0%, 29.9%] | [0.0%, 29.9%] | [0.0%, 29.9%] | [0.0%, 29.9%] | 0.000000 | 0.000000 |
| AGE | 864 | [98.0%, 99.5%] | [0.0%, 0.4%] | [98.7%, 99.8%] | [0.0%, 0.4%] | 1.000000 | 0.117000 |
| DIETARY_PREFERENCE | 9 | [2.0%, 43.5%] | [0.0%, 29.9%] | [2.0%, 43.5%] | [0.0%, 29.9%] | 1.000000 | 0.333000 |
| GENDER | 475 | [99.2%, 100.0%] | [9.4%, 15.2%] | [99.2%, 100.0%] | [9.4%, 15.2%] | 1.000000 | 0.839000 |
| GEOLOCATION | 108 | [61.2%, 78.2%] | [0.0%, 3.4%] | [61.2%, 78.2%] | [0.0%, 3.4%] | 0.704000 | 0.131000 |
| HEALTH_INFORMATION | 800 | [89.8%, 93.6%] | [2.1%, 4.6%] | [95.6%, 98.0%] | [3.5%, 6.5%] | 0.999000 | 0.766000 |
| MARITAL_STATUS | 9 | [70.1%, 100.0%] | [0.0%, 29.9%] | [70.1%, 100.0%] | [0.0%, 29.9%] | 1.000000 | 0.222000 |
| OCCUPATION | 69 | [65.6%, 85.2%] | [0.0%, 5.3%] | [65.6%, 85.2%] | [0.0%, 5.3%] | 0.884000 | 0.130000 |
| RACE | 68 | [94.7%, 100.0%] | [0.3%, 7.9%] | [94.7%, 100.0%] | [0.3%, 7.9%] | 1.000000 | 0.015000 |
| SEXUAL_ORIENTATION | 24 | [21.2%, 57.3%] | [0.0%, 13.8%] | [21.2%, 57.3%] | [0.0%, 13.8%] | 0.904000 | 0.567000 |
| TIME | 85 | [43.6%, 64.3%] | [0.0%, 4.3%] | [72.9%, 89.0%] | [0.0%, 4.3%] | 1.000000 | 0.205000 |

Table 14: Type-wise recovery on MedQA: 95% confidence intervals (H@1/H@3; original and minimized) and mean top-1 confidence.

| Type | Hit@1 (orig) | Hit@1 (mask) | Hit@3 (orig) | Hit@3 (mask) |
|---|---|---|---|---|
| ADDRESS | 1.000 | 0.000 | 1.000 | 0.000 |
| AFFILIATION | 0.845 | 0.042 | 0.892 | 0.044 |
| AGE | 0.746 | 0.029 | 0.787 | 0.029 |
| EDUCATIONAL_RECORD | 0.413 | 0.000 | 0.413 | 0.000 |
| EMAIL | 1.000 | 0.000 | 1.000 | 0.000 |
| ETHNICITY | 1.000 | 0.000 | 1.000 | 0.000 |
| FINANCIAL_INFORMATION | 0.833 | 0.148 | 0.852 | 0.148 |
| GENDER | 1.000 | 0.038 | 1.000 | 0.038 |
| GEOLOCATION | 0.858 | 0.051 | 0.961 | 0.058 |
| HEALTH_INFORMATION | 0.855 | 0.000 | 0.964 | 0.024 |
| INCOME | 0.729 | 0.000 | 0.729 | 0.000 |
| IP_ADDRESS | 0.429 | 0.000 | 1.000 | 0.000 |
| MARITAL_STATUS | 0.703 | 0.000 | 0.781 | 0.000 |
| NAME | 0.853 | 0.018 | 0.937 | 0.018 |
| OCCUPATION | 0.775 | 0.086 | 0.823 | 0.105 |
| RACE | 1.000 | 0.000 | 1.000 | 0.000 |
| RELIGION | 1.000 | 0.000 | 1.000 | 0.000 |
| TIME | 0.861 | 0.046 | 0.918 | 0.052 |
| URL | 0.922 | 0.000 | 0.933 | 0.000 |
| VEHICLE | 1.000 | 0.000 | 1.000 | 0.000 |

Table 15: Type-wise recovery pooled by type on ShareGPT

| Type | N | H@1 CI | H@1 CI~ | H@3 CI | H@3 CI~ | $\overline{\text{conf}}$ | $\overline{\text{conf}}$~ |
|---|---|---|---|---|---|---|---|
| ADDRESS | 8 | [67.6%, 100.0%] | [0.0%, 32.4%] | [67.6%, 100.0%] | [0.0%, 32.4%] | 1.000000 | 0.125000 |
| AFFILIATION | 548 | [81.2%, 87.3%] | [2.8%, 6.2%] | [86.4%, 91.6%] | [3.0%, 6.4%] | 0.934000 | 0.703000 |
| AGE | 272 | [69.1%, 79.4%] | [1.5%, 5.7%] | [73.4%, 83.1%] | [1.5%, 5.7%] | 0.982000 | 0.293000 |
| EDUCATIONAL_RECORD | 46 | [28.3%, 55.7%] | [0.0%, 7.7%] | [28.3%, 55.7%] | [0.0%, 7.7%] | 0.900000 | 0.680000 |
| EMAIL | 17 | [81.6%, 100.0%] | [0.0%, 18.4%] | [81.6%, 100.0%] | [0.0%, 18.4%] | 1.000000 | 0.188000 |
| ETHNICITY | 31 | [89.0%, 100.0%] | [0.0%, 11.0%] | [89.0%, 100.0%] | [0.0%, 11.0%] | 1.000000 | 0.226000 |
| FINANCIAL_INFORMATION | 54 | [71.3%, 91.0%] | [7.7%, 26.6%] | [73.4%, 92.3%] | [7.7%, 26.6%] | 0.852000 | 0.907000 |
| GENDER | 78 | [95.3%, 100.0%] | [1.3%, 10.7%] | [95.3%, 100.0%] | [1.3%, 10.7%] | 1.000000 | 0.342000 |
| GEOLOCATION | 935 | [83.4%, 87.9%] | [3.9%, 6.7%] | [94.7%, 97.2%] | [4.5%, 7.5%] | 0.992000 | 0.674000 |
| HEALTH_INFORMATION | 83 | [76.4%, 91.5%] | [0.0%, 4.4%] | [89.9%, 98.8%] | [0.7%, 8.4%] | 1.000000 | 0.714000 |
| INCOME | 48 | [59.0%, 83.4%] | [0.0%, 7.4%] | [59.0%, 83.4%] | [0.0%, 7.4%] | 0.833000 | 0.677000 |
| IP_ADDRESS | 7 | [15.8%, 75.0%] | [0.0%, 35.4%] | [64.6%, 100.0%] | [0.0%, 35.4%] | 1.000000 | 0.857000 |
| MARITAL_STATUS | 64 | [58.2%, 80.1%] | [0.0%, 5.7%] | [66.6%, 86.5%] | [0.0%, 5.7%] | 0.969000 | 0.262000 |
| NAME | 621 | [82.3%, 87.9%] | [1.0%, 3.1%] | [91.5%, 95.4%] | [1.0%, 3.1%] | 0.958000 | 0.597000 |
| OCCUPATION | 209 | [71.4%, 82.6%] | [5.5%, 13.2%] | [76.6%, 86.9%] | [7.1%, 15.4%] | 0.911000 | 0.588000 |
| RACE | 13 | [77.2%, 100.0%] | [0.0%, 22.8%] | [77.2%, 100.0%] | [0.0%, 22.8%] | 1.000000 | 0.154000 |
| RELIGION | 7 | [64.6%, 100.0%] | [0.0%, 35.4%] | [64.6%, 100.0%] | [0.0%, 35.4%] | 1.000000 | 1.000000 |
| TIME | 656 | [83.3%, 88.6%] | [3.2%, 6.5%] | [89.4%, 93.6%] | [3.7%, 7.2%] | 0.998000 | 0.695000 |
| URL | 90 | [84.8%, 96.2%] | [0.0%, 4.1%] | [86.2%, 96.9%] | [0.0%, 4.1%] | 0.933000 | 0.561000 |
| VEHICLE | 6 | [61.0%, 100.0%] | [0.0%, 39.0%] | [61.0%, 100.0%] | [0.0%, 39.0%] | 1.000000 | 1.000000 |

Table 16: Type-wise recovery on ShareGPT: 95% confidence intervals (H@1/H@3; original and minimized) and mean top-1 confidence.

| Type | Hit@1 (orig) | Hit@1 (mask) | Hit@3 (orig) | Hit@3 (mask) |
|---|---|---|---|---|
| AFFILIATION | 0.830 | 0.019 | 0.871 | 0.019 |
| AGE | 0.691 | 0.000 | 0.764 | 0.000 |
| EDUCATIONAL_RECORD | 0.667 | 0.000 | 1.000 | 0.000 |
| EMAIL | 0.000 | 0.000 | 0.000 | 0.000 |
| ETHNICITY | 0.630 | 0.000 | 1.000 | 0.000 |
| FINANCIAL_INFORMATION | 0.923 | 0.000 | 0.923 | 0.000 |
| GENDER | 1.000 | 0.026 | 1.000 | 0.026 |
| GEOLOCATION | 0.898 | 0.022 | 0.954 | 0.031 |
| GPA | 1.000 | 0.000 | 1.000 | 0.000 |
| HEALTH_INFORMATION | 1.000 | 0.000 | 1.000 | 0.000 |
| ID_NUMBER | 1.000 | 0.000 | 1.000 | 0.000 |
| INCOME | 0.727 | 0.000 | 0.727 | 0.030 |
| KEYS | 1.000 | 0.000 | 1.000 | 0.000 |
| NAME | 0.903 | 0.000 | 0.981 | 0.000 |
| OCCUPATION | 0.854 | 0.080 | 0.934 | 0.080 |
| PHONE_NUMBER | 1.000 | 0.000 | 1.000 | 0.000 |
| PRODUCT | 1.000 | 0.000 | 1.000 | 0.000 |
| QUANTITY | 0.500 | 0.000 | 0.500 | 0.000 |
| RACE | 1.000 | 0.000 | 1.000 | 0.000 |
| TIME | 0.733 | 0.000 | 0.862 | 0.000 |
| URL | 0.886 | 0.000 | 0.886 | 0.000 |
| USERNAME | 0.533 | 0.000 | 0.533 | 0.000 |

Table 17: Type-wise recovery pooled by type on WildChat

| Type | N | H@1 CI | H@1 CI$^\sim$ | H@3 CI | H@3 CI$^\sim$ | $\overline{\text{conf}}$ | $\overline{\text{conf}}^\sim$ |
|---|---|---|---|---|---|---|---|
| AFFILIATION | 535 | [79.6%, 85.9%] | [1.0%, 3.4%] | [84.0%, 89.7%] | [1.0%, 3.4%] | 0.923000 | 0.671000 |
| AGE | 123 | [60.5%, 76.6%] | [0.0%, 3.0%] | [68.2%, 83.1%] | [0.0%, 3.0%] | 0.927000 | 0.263000 |
| EDUCATIONAL_RECORD | 24 | [46.7%, 82.0%] | [0.0%, 13.8%] | [86.2%, 100.0%] | [0.0%, 13.8%] | 1.000000 | 0.750000 |
| EMAIL | 18 | [0.0%, 17.6%] | [0.0%, 17.6%] | [0.0%, 17.6%] | [0.0%, 17.6%] | 0.200000 | 0.111000 |
| ETHNICITY | 27 | [44.2%, 78.5%] | [0.0%, 12.5%] | [87.5%, 100.0%] | [0.0%, 12.5%] | 1.000000 | 0.289000 |
| FINANCIAL_INFORMATION | 39 | [79.7%, 97.3%] | [0.0%, 9.0%] | [79.7%, 97.3%] | [0.0%, 9.0%] | 1.000000 | 0.949000 |
| GENDER | 38 | [90.8%, 100.0%] | [0.5%, 13.5%] | [90.8%, 100.0%] | [0.5%, 13.5%] | 1.000000 | 0.337000 |
| GEOLOCATION | 677 | [87.3%, 91.9%] | [1.3%, 3.6%] | [93.6%, 96.8%] | [2.0%, 4.7%] | 0.972000 | 0.577000 |
| GPA | 8 | [67.6%, 100.0%] | [0.0%, 32.4%] | [67.6%, 100.0%] | [0.0%, 32.4%] | 1.000000 | 0.250000 |
| HEALTH_INFORMATION | 39 | [91.0%, 100.0%] | [0.0%, 9.0%] | [91.0%, 100.0%] | [0.0%, 9.0%] | 1.000000 | 0.610000 |
| ID_NUMBER | 9 | [70.1%, 100.0%] | [0.0%, 29.9%] | [70.1%, 100.0%] | [0.0%, 29.9%] | 1.000000 | 0.111000 |
| INCOME | 33 | [55.8%, 84.9%] | [0.0%, 10.4%] | [55.8%, 84.9%] | [0.5%, 15.3%] | 0.758000 | 0.515000 |
| KEYS | 9 | [70.1%, 100.0%] | [0.0%, 29.9%] | [70.1%, 100.0%] | [0.0%, 29.9%] | 1.000000 | 0.444000 |
| NAME | 621 | [87.8%, 92.4%] | [0.0%, 0.6%] | [96.7%, 98.9%] | [0.0%, 0.6%] | 0.986000 | 0.558000 |
| OCCUPATION | 137 | [78.5%, 90.3%] | [4.5%, 13.8%] | [88.0%, 96.5%] | [4.5%, 13.8%] | 1.000000 | 0.531000 |
| PHONE_NUMBER | 9 | [70.1%, 100.0%] | [0.0%, 29.9%] | [70.1%, 100.0%] | [0.0%, 29.9%] | 1.000000 | 1.000000 |
| PRODUCT | 8 | [67.6%, 100.0%] | [0.0%, 32.4%] | [67.6%, 100.0%] | [0.0%, 32.4%] | 1.000000 | 0.750000 |
| QUANTITY | 18 | [29.0%, 71.0%] | [0.0%, 17.6%] | [29.0%, 71.0%] | [0.0%, 17.6%] | 1.000000 | 1.000000 |
| RACE | 8 | [67.6%, 100.0%] | [0.0%, 32.4%] | [67.6%, 100.0%] | [0.0%, 32.4%] | 1.000000 | 0.250000 |
| TIME | 536 | [69.4%, 76.9%] | [0.0%, 0.7%] | [83.0%, 88.9%] | [0.0%, 0.7%] | 0.998000 | 0.566000 |
| URL | 44 | [76.0%, 95.0%] | [0.0%, 8.0%] | [76.0%, 95.0%] | [0.0%, 8.0%] | 0.886000 | 0.443000 |
| USERNAME | 15 | [30.1%, 75.2%] | [0.0%, 20.4%] | [30.1%, 75.2%] | [0.0%, 20.4%] | 0.533000 | 0.533000 |

Table 18: Type-wise recovery on WildChat: 95% confidence intervals (H@1/H@3; original and minimized) and mean top-1 confidence.

## I.2 GPT–5 AS ATTACKER ON ITS OWN ORACLE-MINIMIZED PROMPTS

| action | N | $p_{\text{corr}}$ | $p_{\text{corr,lo}}$ | $p_{\text{corr,hi}}$ | $p_{\text{unk}}$ | $p_{\text{unk,lo}}$ | $p_{\text{unk,hi}}$ | $\overline{\text{conf}}$ |
|---|---|---|---|---|---|---|---|---|
| abstract | 679 | 0.119 | 0.097 | 0.146 | 0.323 | 0.288 | 0.359 | 0.627 |
| redact | 5627 | 0.077 | 0.070 | 0.084 | 0.762 | 0.750 | 0.773 | 0.175 |

Table 19: Span-wise recovery with GPT–5 as attacker on its own oracle-minimized prompts by action on WildChat

| action | N | $p_{\text{corr}}$ | $p_{\text{corr,lo}}$ | $p_{\text{corr,hi}}$ | $p_{\text{unk}}$ | $p_{\text{unk,lo}}$ | $p_{\text{unk,hi}}$ | $\overline{\text{conf}}$ |
|---|---|---|---|---|---|---|---|---|
| abstract | 118 | 0.127 | 0.068 | 0.195 | 0.280 | 0.203 | 0.364 | 0.719 |
| redact | 987 | 0.020 | 0.012 | 0.029 | 0.967 | 0.954 | 0.978 | 0.030 |

Table 20: Span-wise recovery with GPT–5 as attacker on its own oracle-minimized prompts by action on ShareGPT

| action | N | $p_{\text{corr}}$ | $p_{\text{corr,lo}}$ | $p_{\text{corr,hi}}$ | $p_{\text{unk}}$ | $p_{\text{unk,lo}}$ | $p_{\text{unk,hi}}$ | $\overline{\text{conf}}$ |
|---|---|---|---|---|---|---|---|---|
| abstract | 4 | 0.250 | 0.000 | 0.750 | 0.500 | 0.000 | 1.000 | 0.487 |
| redact | 397 | 0.055 | 0.035 | 0.081 | 0.922 | 0.894 | 0.947 | 0.069 |

Table 21: Span-wise recovery with GPT–5 as attacker on its own oracle-minimized prompts by action on CaseHOLD

| action | N | $p_{\text{corr}}$ | $p_{\text{corr,lo}}$ | $p_{\text{corr,hi}}$ | $p_{\text{unk}}$ | $p_{\text{unk,lo}}$ | $p_{\text{unk,hi}}$ | $\overline{\text{conf}}$ |
|---|---|---|---|---|---|---|---|---|
| abstract | 21 | 0.000 | 0.000 | 0.000 | 0.000 | 0.000 | 0.000 | 1.000 |
| redact | 941 | 0.003 | 0.000 | 0.006 | 0.995 | 0.990 | 0.999 | 0.004 |

Table 22: Span-wise recovery with GPT–5 as attacker on its own oracle-minimized prompts by action on MedQA

| Type | Hit@1 (orig) | Hit@1 (mask) | Hit@3 (orig) | Hit@3 (mask) |
|---|---|---|---|---|
| AFFILIATION | 74.6% | 0.0% | 85.7% | 0.0% |
| AGE | 94.1% | 0.0% | 94.1% | 0.0% |
| EDUCATIONAL_RECORD | 66.7% | 0.0% | 66.7% | 0.0% |
| EMAIL | 0.0% | 0.0% | 0.0% | 0.0% |
| ETHNICITY | 50.0% | 0.0% | 75.0% | 0.0% |
| FINANCIAL_INFORMATION | 25.0% | 0.0% | 50.0% | 0.0% |
| GENDER | 100.0% | 0.0% | 100.0% | 0.0% |
| GEOLOCATION | 81.5% | 0.0% | 93.8% | 0.0% |
| GPA | 0.0% | 0.0% | 100.0% | 0.0% |
| HEALTH_INFORMATION | 20.0% | 0.0% | 40.0% | 0.0% |
| ID_NUMBER | 100.0% | 0.0% | 100.0% | 0.0% |
| INCOME | 50.0% | 0.0% | 50.0% | 0.0% |
| KEYS | 0.0% | 0.0% | 0.0% | 0.0% |
| NAME | 92.1% | 0.0% | 97.4% | 0.0% |
| OCCUPATION | 88.2% | 0.0% | 88.2% | 0.0% |
| PHONE_NUMBER | 100.0% | 0.0% | 100.0% | 0.0% |
| PRODUCT | 100.0% | 0.0% | 100.0% | 0.0% |
| QUANTITY | 0.0% | 0.0% | 50.0% | 0.0% |
| RACE | 100.0% | 0.0% | 100.0% | 0.0% |
| TIME | 73.0% | 0.0% | 82.5% | 0.0% |
| URL | 60.0% | 0.0% | 60.0% | 0.0% |
| USERNAME | 50.0% | 0.0% | 50.0% | 0.0% |

Table 23: Type-wise recovery with GPT–5 as attacker on its own oracle-minimized prompts on WildChat.

| Type | Hit@1 (orig) | Hit@1 (mask) | Hit@3 (orig) | Hit@3 (mask) |
|---|---|---|---|---|
| ADDRESS | 100.0% | 0.0% | 100.0% | 0.0% |
| AFFILIATION | 74.2% | 1.5% | 81.8% | 1.5% |
| AGE | 58.3% | 0.0% | 63.9% | 0.0% |
| EDUCATIONAL_RECORD | 33.3% | 0.0% | 33.3% | 0.0% |
| EMAIL | 100.0% | 0.0% | 100.0% | 0.0% |
| ETHNICITY | 75.0% | 0.0% | 75.0% | 0.0% |
| FINANCIAL_INFORMATION | 42.9% | 0.0% | 57.1% | 0.0% |
| GENDER | 90.9% | 0.0% | 100.0% | 0.0% |
| GEOLOCATION | 82.2% | 1.7% | 93.2% | 1.7% |
| HEALTH_INFORMATION | 50.0% | 0.0% | 60.0% | 0.0% |
| INCOME | 33.3% | 0.0% | 50.0% | 0.0% |
| IP_ADDRESS | 100.0% | 0.0% | 100.0% | 0.0% |
| MARITAL_STATUS | 18.8% | 0.0% | 25.0% | 0.0% |
| NAME | 91.0% | 1.3% | 94.9% | 1.3% |
| OCCUPATION | 59.3% | 3.7% | 70.4% | 3.7% |
| RACE | 100.0% | 0.0% | 100.0% | 0.0% |
| RELIGION | 100.0% | 0.0% | 100.0% | 0.0% |
| TIME | 71.4% | 2.4% | 83.3% | 2.4% |
| URL | 90.9% | 0.0% | 90.9% | 0.0% |
| VEHICLE | 100.0% | 0.0% | 100.0% | 0.0% |

Table 24: Type-wise recovery with GPT–5 as attacker on its own oracle-minimized prompts on ShareGPT.

| Type | Hit@1 (orig) | Hit@1 (mask) | Hit@3 (orig) | Hit@3 (mask) |
|---|---|---|---|---|
| ADDRESS | 100.0% | 0.0% | 100.0% | 0.0% |
| AFFILIATION | 74.5% | 1.8% | 81.8% | 1.8% |
| AGE | 0.0% | 0.0% | 0.0% | 0.0% |
| ETHNICITY | 0.0% | 0.0% | 0.0% | 0.0% |
| FINANCIAL_INFORMATION | 100.0% | 0.0% | 100.0% | 0.0% |
| GEOLOCATION | 83.3% | 0.0% | 95.2% | 2.4% |
| HEALTH_INFORMATION | 0.0% | 0.0% | 0.0% | 0.0% |
| INCOME | 0.0% | 0.0% | 100.0% | 0.0% |
| NAME | 84.4% | 0.0% | 88.9% | 0.0% |
| RACE | 50.0% | 0.0% | 50.0% | 0.0% |
| TIME | 82.8% | 0.0% | 93.1% | 0.0% |

Table 25: Type-wise recovery with GPT–5 as attacker on its own oracle-minimized prompts on CaseHOLD.

| Type | Hit@1 (orig) | Hit@1 (mask) | Hit@3 (orig) | Hit@3 (mask) |
|---|---|---|---|---|
| AFFILIATION | 0.0% | 0.0% | 0.0% | 0.0% |
| AGE | 99.0% | 0.0% | 99.0% | 0.0% |
| DIETARY_PREFERENCE | 100.0% | 0.0% | 100.0% | 0.0% |
| GENDER | 100.0% | 0.0% | 100.0% | 0.0% |
| GEOLOCATION | 46.2% | 0.0% | 53.8% | 0.0% |
| HEALTH_INFORMATION | 83.7% | 0.0% | 92.4% | 0.0% |
| MARITAL_STATUS | 100.0% | 0.0% | 100.0% | 0.0% |
| OCCUPATION | 50.0% | 0.0% | 50.0% | 0.0% |
| RACE | 100.0% | 0.0% | 100.0% | 0.0% |
| SEXUAL_ORIENTATION | 100.0% | 0.0% | 100.0% | 0.0% |
| TIME | 50.0% | 0.0% | 80.0% | 0.0% |

Table 26: Type-wise recovery with GPT–5 as attacker on its own oracle-minimized prompts on MedQA.

