# SUPPLEMENTARY MATERIAL FOR "OPERATIONALIZING DATA MINIMIZATION FOR PRIVACY-PRESERVING LLM PROMPTING"

## 1 FULL PER-TYPE BREAKDOWN RESULTS

We report full per-type weighted results for all datasets (CaseHOLD, MedQA, ShareGPT90K, and WildChat), under both Oracle and Prediction settings, across all nine evaluated models.

For each (dataset, setting, model) combination, we provide detailed breakdowns over entity types, including weighted Redact, Abstract, and Retain rates.

### 1.1 CASEHOLD

| Type | Weighted Redact | Weighted Abstract | Weighted Retain |
|---|---|---|---|
| NAME | 98.94% (93/94) | 0.00% (0/94) | 1.06% (1/94) |
| TIME | 96.61% (57/59) | 1.69% (1/59) | 1.69% (1/59) |
| GEOLOCATION | 96.15% (75/78) | 1.28% (1/78) | 2.56% (2/78) |
| AFFILIATION | 93.17% (150/161) | 1.86% (3/161) | 4.97% (8/161) |
| RACE | 100.00% (4/4) | 0.00% (0/4) | 0.00% (0/4) |
| ETHNICITY | 100.00% (1/1) | 0.00% (0/1) | 0.00% (0/1) |
| ADDRESS | 100.00% (1/1) | 0.00% (0/1) | 0.00% (0/1) |
| INCOME | 100.00% (1/1) | 0.00% (0/1) | 0.00% (0/1) |
| AGE | 100.00% (1/1) | 0.00% (0/1) | 0.00% (0/1) |
| HEALTH_INFORMATION | 100.00% (1/1) | 0.00% (0/1) | 0.00% (0/1) |
| FINANCIAL_INFORMATION | 100.00% (1/1) | 0.00% (0/1) | 0.00% (0/1) |
| **Overall** | 95.77% (385/402) | 1.24% (5/402) | 2.99% (12/402) |

Table 1: Weighted Results per Type and Overall (Oracle for **CaseHOLD**), Model: **gpt-4.1-nano**

| Type | Weighted Redact | Weighted Abstract | Weighted Retain |
|---|---|---|---|
| NAME | 98.94% (93/94) | 0.00% (0/94) | 1.06% (1/94) |
| TIME | 100.00% (59/59) | 0.00% (0/59) | 0.00% (0/59) |
| GEOLOCATION | 100.00% (78/78) | 0.00% (0/78) | 0.00% (0/78) |
| AFFILIATION | 100.00% (161/161) | 0.00% (0/161) | 0.00% (0/161) |
| RACE | 100.00% (4/4) | 0.00% (0/4) | 0.00% (0/4) |
| ETHNICITY | 100.00% (1/1) | 0.00% (0/1) | 0.00% (0/1) |
| ADDRESS | 100.00% (1/1) | 0.00% (0/1) | 0.00% (0/1) |
| INCOME | 100.00% (1/1) | 0.00% (0/1) | 0.00% (0/1) |
| AGE | 100.00% (1/1) | 0.00% (0/1) | 0.00% (0/1) |
| HEALTH_INFORMATION | 100.00% (1/1) | 0.00% (0/1) | 0.00% (0/1) |
| FINANCIAL_INFORMATION | 100.00% (1/1) | 0.00% (0/1) | 0.00% (0/1) |
| **Overall** | 99.75% (401/402) | 0.00% (0/402) | 0.25% (1/402) |

Table 2: Weighted Results per Type and Overall (Oracle for **CaseHOLD**), Model: **gpt-4.1**

| Type | Weighted Redact | Weighted Abstract | Weighted Retain |
|---|---|---|---|
| NAME | 97.87% (92/94) | 2.13% (2/94) | 0.00% (0/94) |
| TIME | 96.61% (57/59) | 1.69% (1/59) | 1.69% (1/59) |
| GEOLOCATION | 98.72% (77/78) | 1.28% (1/78) | 0.00% (0/78) |
| AFFILIATION | 100.00% (161/161) | 0.00% (0/161) | 0.00% (0/161) |
| RACE | 100.00% (4/4) | 0.00% (0/4) | 0.00% (0/4) |
| ETHNICITY | 100.00% (1/1) | 0.00% (0/1) | 0.00% (0/1) |
| ADDRESS | 100.00% (1/1) | 0.00% (0/1) | 0.00% (0/1) |
| INCOME | 100.00% (1/1) | 0.00% (0/1) | 0.00% (0/1) |
| AGE | 100.00% (1/1) | 0.00% (0/1) | 0.00% (0/1) |
| HEALTH_INFORMATION | 100.00% (1/1) | 0.00% (0/1) | 0.00% (0/1) |
| FINANCIAL_INFORMATION | 100.00% (1/1) | 0.00% (0/1) | 0.00% (0/1) |
| **Overall** | **98.76% (397/402)** | **1.00% (4/402)** | **0.25% (1/402)** |

Table 3: Weighted Results per Type and Overall (Oracle for **CaseHOLD**), Model: **gpt-5**

| Type | Weighted Redact | Weighted Abstract | Weighted Retain |
|---|---|---|---|
| NAME | 100.00% (94/94) | 0.00% (0/94) | 0.00% (0/94) |
| TIME | 98.31% (58/59) | 1.69% (1/59) | 0.00% (0/59) |
| GEOLOCATION | 100.00% (78/78) | 0.00% (0/78) | 0.00% (0/78) |
| AFFILIATION | 99.38% (160/161) | 0.62% (1/161) | 0.00% (0/161) |
| RACE | 100.00% (4/4) | 0.00% (0/4) | 0.00% (0/4) |
| ETHNICITY | 100.00% (1/1) | 0.00% (0/1) | 0.00% (0/1) |
| ADDRESS | 100.00% (1/1) | 0.00% (0/1) | 0.00% (0/1) |
| INCOME | 100.00% (1/1) | 0.00% (0/1) | 0.00% (0/1) |
| AGE | 100.00% (1/1) | 0.00% (0/1) | 0.00% (0/1) |
| HEALTH_INFORMATION | 100.00% (1/1) | 0.00% (0/1) | 0.00% (0/1) |
| FINANCIAL_INFORMATION | 100.00% (1/1) | 0.00% (0/1) | 0.00% (0/1) |
| **Overall** | **99.50% (400/402)** | **0.50% (2/402)** | **0.00% (0/402)** |

Table 4: Weighted Results per Type and Overall (Oracle for **CaseHOLD**), Model: **claude-3-7-sonnet-20250219**

| Type | Weighted Redact | Weighted Abstract | Weighted Retain |
|---|---|---|---|
| NAME | 100.00% (94/94) | 0.00% (0/94) | 0.00% (0/94) |
| TIME | 100.00% (59/59) | 0.00% (0/59) | 0.00% (0/59) |
| GEOLOCATION | 100.00% (78/78) | 0.00% (0/78) | 0.00% (0/78) |
| AFFILIATION | 100.00% (161/161) | 0.00% (0/161) | 0.00% (0/161) |
| RACE | 100.00% (4/4) | 0.00% (0/4) | 0.00% (0/4) |
| ETHNICITY | 100.00% (1/1) | 0.00% (0/1) | 0.00% (0/1) |
| ADDRESS | 100.00% (1/1) | 0.00% (0/1) | 0.00% (0/1) |
| INCOME | 100.00% (1/1) | 0.00% (0/1) | 0.00% (0/1) |
| AGE | 100.00% (1/1) | 0.00% (0/1) | 0.00% (0/1) |
| HEALTH_INFORMATION | 100.00% (1/1) | 0.00% (0/1) | 0.00% (0/1) |
| FINANCIAL_INFORMATION | 100.00% (1/1) | 0.00% (0/1) | 0.00% (0/1) |
| **Overall** | **100.00% (402/402)** | **0.00% (0/402)** | **0.00% (0/402)** |

Table 5: Weighted Results per Type and Overall (Oracle for **CaseHOLD**), Model: **claude-sonnet-4-20250514**

| Type | Weighted Redact | Weighted Abstract | Weighted Retain |
|---|---|---|---|
| NAME | 84.04% (79/94) | 5.32% (5/94) | 10.64% (10/94) |
| TIME | 57.63% (34/59) | 6.78% (4/59) | 35.59% (21/59) |
| GEOLOCATION | 74.36% (58/78) | 5.13% (4/78) | 20.51% (16/78) |
| AFFILIATION | 71.43% (115/161) | 9.94% (16/161) | 18.63% (30/161) |
| RACE | 100.00% (4/4) | 0.00% (0/4) | 0.00% (0/4) |
| ETHNICITY | 100.00% (1/1) | 0.00% (0/1) | 0.00% (0/1) |
| ADDRESS | 100.00% (1/1) | 0.00% (0/1) | 0.00% (0/1) |
| INCOME | 100.00% (1/1) | 0.00% (0/1) | 0.00% (0/1) |
| AGE | 0.00% (0/1) | 0.00% (0/1) | 100.00% (1/1) |
| HEALTH_INFORMATION | 0.00% (0/1) | 0.00% (0/1) | 100.00% (1/1) |
| FINANCIAL_INFORMATION | 100.00% (1/1) | 0.00% (0/1) | 0.00% (0/1) |
| **Overall** | 73.13% (294/402) | 7.21% (29/402) | 19.65% (79/402) |

Table 6: Weighted Results per Type and Overall (Oracle for **CaseHOLD**), Model: **lgai/exaone-deep-32b**

| Type | Weighted Redact | Weighted Abstract | Weighted Retain |
|---|---|---|---|
| NAME | 95.74% (90/94) | 0.00% (0/94) | 4.26% (4/94) |
| TIME | 96.61% (57/59) | 1.69% (1/59) | 1.69% (1/59) |
| GEOLOCATION | 92.31% (72/78) | 1.28% (1/78) | 6.41% (5/78) |
| AFFILIATION | 95.65% (154/161) | 0.62% (1/161) | 3.73% (6/161) |
| RACE | 75.00% (3/4) | 25.00% (1/4) | 0.00% (0/4) |
| ETHNICITY | 100.00% (1/1) | 0.00% (0/1) | 0.00% (0/1) |
| ADDRESS | 100.00% (1/1) | 0.00% (0/1) | 0.00% (0/1) |
| INCOME | 100.00% (1/1) | 0.00% (0/1) | 0.00% (0/1) |
| AGE | 100.00% (1/1) | 0.00% (0/1) | 0.00% (0/1) |
| HEALTH_INFORMATION | 100.00% (1/1) | 0.00% (0/1) | 0.00% (0/1) |
| FINANCIAL_INFORMATION | 100.00% (1/1) | 0.00% (0/1) | 0.00% (0/1) |
| **Overall** | 95.02% (382/402) | 1.00% (4/402) | 3.98% (16/402) |

Table 7: Weighted Results per Type and Overall (Oracle for **CaseHOLD**), Model: **mistralai/mistral-small-3.1-24b-instruct**

| Type | Weighted Redact | Weighted Abstract | Weighted Retain |
|---|---|---|---|
| NAME | 95.74% (90/94) | 1.06% (1/94) | 3.19% (3/94) |
| TIME | 98.31% (58/59) | 0.00% (0/59) | 1.69% (1/59) |
| GEOLOCATION | 97.44% (76/78) | 0.00% (0/78) | 2.56% (2/78) |
| AFFILIATION | 97.52% (157/161) | 0.00% (0/161) | 2.48% (4/161) |
| RACE | 100.00% (4/4) | 0.00% (0/4) | 0.00% (0/4) |
| ETHNICITY | 100.00% (1/1) | 0.00% (0/1) | 0.00% (0/1) |
| ADDRESS | 100.00% (1/1) | 0.00% (0/1) | 0.00% (0/1) |
| INCOME | 0.00% (0/1) | 0.00% (0/1) | 100.00% (1/1) |
| AGE | 100.00% (1/1) | 0.00% (0/1) | 0.00% (0/1) |
| HEALTH_INFORMATION | 100.00% (1/1) | 0.00% (0/1) | 0.00% (0/1) |
| FINANCIAL_INFORMATION | 100.00% (1/1) | 0.00% (0/1) | 0.00% (0/1) |
| **Overall** | 97.01% (390/402) | 0.25% (1/402) | 2.74% (11/402) |

Table 8: Weighted Results per Type and Overall (Oracle for **CaseHOLD**), Model: **qwen/qwen2.5-7b-instruct**

| Type | Weighted Redact | Weighted Abstract | Weighted Retain |
|---|---|---|---|
| NAME | 39.36% (37/94) | 5.32% (5/94) | 55.32% (52/94) |
| TIME | 35.59% (21/59) | 5.08% (3/59) | 59.32% (35/59) |
| GEOLOCATION | 38.46% (30/78) | 8.97% (7/78) | 52.56% (41/78) |
| AFFILIATION | 44.10% (71/161) | 6.21% (10/161) | 49.69% (80/161) |
| RACE | 25.00% (1/4) | 25.00% (1/4) | 50.00% (2/4) |
| ETHNICITY | 100.00% (1/1) | 0.00% (0/1) | 0.00% (0/1) |
| ADDRESS | 0.00% (0/1) | 0.00% (0/1) | 100.00% (1/1) |
| INCOME | 0.00% (0/1) | 0.00% (0/1) | 100.00% (1/1) |
| AGE | 100.00% (1/1) | 0.00% (0/1) | 0.00% (0/1) |
| HEALTH_INFORMATION | 100.00% (1/1) | 0.00% (0/1) | 0.00% (0/1) |
| FINANCIAL_INFORMATION | 100.00% (1/1) | 0.00% (0/1) | 0.00% (0/1) |
| **Overall** | 40.80% (164/402) | 6.47% (26/402) | 52.74% (212/402) |

Table 9: Weighted Results per Type and Overall (Oracle for **CaseHOLD**), Model: **qwen/qwen2.5-0.5b-instruct**

| Type | Weighted Redact | Weighted Abstract | Weighted Retain |
|---|---|---|---|
| NAME | 71.28% (67/94) | 28.72% (27/94) | 0.00% (0/94) |
| TIME | 79.66% (47/59) | 20.34% (12/59) | 0.00% (0/59) |
| GEOLOCATION | 80.77% (63/78) | 19.23% (15/78) | 0.00% (0/78) |
| AFFILIATION | 81.99% (132/161) | 18.01% (29/161) | 0.00% (0/161) |
| RACE | 75.00% (3/4) | 25.00% (1/4) | 0.00% (0/4) |
| ETHNICITY | 100.00% (1/1) | 0.00% (0/1) | 0.00% (0/1) |
| ADDRESS | 100.00% (1/1) | 0.00% (0/1) | 0.00% (0/1) |
| INCOME | 100.00% (1/1) | 0.00% (0/1) | 0.00% (0/1) |
| AGE | 0.00% (0/1) | 100.00% (1/1) | 0.00% (0/1) |
| HEALTH_INFORMATION | 0.00% (0/1) | 100.00% (1/1) | 0.00% (0/1) |
| FINANCIAL_INFORMATION | 0.00% (0/1) | 100.00% (1/1) | 0.00% (0/1) |
| **Overall** | 78.36% (315/402) | 21.64% (87/402) | 0.00% (0/402) |

Table 10: Weighted Results per Type and Overall (Prediction for **CaseHOLD**), Model: **gpt-4.1-nano**

| Type | Weighted Redact | Weighted Abstract | Weighted Retain |
|---|---|---|---|
| NAME | 0.00% (0/94) | 100.00% (94/94) | 0.00% (0/94) |
| TIME | 0.00% (0/59) | 100.00% (59/59) | 0.00% (0/59) |
| GEOLOCATION | 0.00% (0/78) | 84.62% (66/78) | 15.38% (12/78) |
| AFFILIATION | 0.00% (0/161) | 95.03% (153/161) | 4.97% (8/161) |
| RACE | 0.00% (0/4) | 75.00% (3/4) | 25.00% (1/4) |
| ETHNICITY | 0.00% (0/1) | 0.00% (0/1) | 100.00% (1/1) |
| ADDRESS | 0.00% (0/1) | 100.00% (1/1) | 0.00% (0/1) |
| INCOME | 0.00% (0/1) | 100.00% (1/1) | 0.00% (0/1) |
| AGE | 0.00% (0/1) | 100.00% (1/1) | 0.00% (0/1) |
| HEALTH_INFORMATION | 0.00% (0/1) | 100.00% (1/1) | 0.00% (0/1) |
| FINANCIAL_INFORMATION | 0.00% (0/1) | 100.00% (1/1) | 0.00% (0/1) |
| **Overall** | 0.00% (0/402) | 94.53% (380/402) | 5.47% (22/402) |

Table 11: Weighted Results per Type and Overall (Prediction for **CaseHOLD**), Model: **gpt-4.1**

| Type | Weighted Redact | Weighted Abstract | Weighted Retain |
|---|---|---|---|
| NAME | 88.30% (83/94) | 6.38% (6/94) | 5.32% (5/94) |
| TIME | 86.44% (51/59) | 13.56% (8/59) | 0.00% (0/59) |
| GEOLOCATION | 76.92% (60/78) | 10.26% (8/78) | 12.82% (10/78) |
| AFFILIATION | 89.44% (144/161) | 5.59% (9/161) | 4.97% (8/161) |
| RACE | 100.00% (4/4) | 0.00% (0/4) | 0.00% (0/4) |
| ETHNICITY | 100.00% (1/1) | 0.00% (0/1) | 0.00% (0/1) |
| ADDRESS | 100.00% (1/1) | 0.00% (0/1) | 0.00% (0/1) |
| INCOME | 100.00% (1/1) | 0.00% (0/1) | 0.00% (0/1) |
| AGE | 100.00% (1/1) | 0.00% (0/1) | 0.00% (0/1) |
| HEALTH_INFORMATION | 100.00% (1/1) | 0.00% (0/1) | 0.00% (0/1) |
| FINANCIAL_INFORMATION | 100.00% (1/1) | 0.00% (0/1) | 0.00% (0/1) |
| **Overall** | 86.57% (348/402) | 7.71% (31/402) | 5.72% (23/402) |

Table 12: Weighted Results per Type and Overall (Prediction for **CaseHOLD**), Model: **gpt-5**

| Type | Weighted Redact | Weighted Abstract | Weighted Retain |
|---|---|---|---|
| NAME | 1.06% (1/94) | 62.77% (59/94) | 36.17% (34/94) |
| TIME | 1.69% (1/59) | 45.76% (27/59) | 52.54% (31/59) |
| GEOLOCATION | 0.00% (0/78) | 20.51% (16/78) | 79.49% (62/78) |
| AFFILIATION | 0.00% (0/161) | 32.30% (52/161) | 67.70% (109/161) |
| RACE | 0.00% (0/4) | 50.00% (2/4) | 50.00% (2/4) |
| ETHNICITY | 0.00% (0/1) | 0.00% (0/1) | 100.00% (1/1) |
| ADDRESS | 0.00% (0/1) | 100.00% (1/1) | 0.00% (0/1) |
| INCOME | 0.00% (0/1) | 100.00% (1/1) | 0.00% (0/1) |
| AGE | 0.00% (0/1) | 0.00% (0/1) | 100.00% (1/1) |
| HEALTH_INFORMATION | 0.00% (0/1) | 100.00% (1/1) | 0.00% (0/1) |
| FINANCIAL_INFORMATION | 0.00% (0/1) | 100.00% (1/1) | 0.00% (0/1) |
| **Overall** | 0.50% (2/402) | 39.80% (160/402) | 59.70% (240/402) |

Table 13: Weighted Results per Type and Overall (Prediction for **CaseHOLD**), Model: **claude-3-7-sonnet-20250219**

| Type | Weighted Redact | Weighted Abstract | Weighted Retain |
|---|---|---|---|
| NAME | 17.02% (16/94) | 60.64% (57/94) | 22.34% (21/94) |
| TIME | 11.86% (7/59) | 62.71% (37/59) | 25.42% (15/59) |
| GEOLOCATION | 0.00% (0/78) | 29.49% (23/78) | 70.51% (55/78) |
| AFFILIATION | 3.11% (5/161) | 35.40% (57/161) | 61.49% (99/161) |
| RACE | 0.00% (0/4) | 50.00% (2/4) | 50.00% (2/4) |
| ETHNICITY | 0.00% (0/1) | 0.00% (0/1) | 100.00% (1/1) |
| ADDRESS | 0.00% (0/1) | 100.00% (1/1) | 0.00% (0/1) |
| INCOME | 0.00% (0/1) | 0.00% (0/1) | 100.00% (1/1) |
| AGE | 0.00% (0/1) | 100.00% (1/1) | 0.00% (0/1) |
| HEALTH_INFORMATION | 0.00% (0/1) | 100.00% (1/1) | 0.00% (0/1) |
| FINANCIAL_INFORMATION | 0.00% (0/1) | 100.00% (1/1) | 0.00% (0/1) |
| **Overall** | 6.97% (28/402) | 44.78% (180/402) | 48.26% (194/402) |

Table 14: Weighted Results per Type and Overall (Prediction for **CaseHOLD**), Model: **claude-sonnet-4-20250514**

| Type | Weighted Redact | Weighted Abstract | Weighted Retain |
|---|---|---|---|
| NAME | 70.21% (66/94) | 29.79% (28/94) | 0.00% (0/94) |
| TIME | 66.10% (39/59) | 33.90% (20/59) | 0.00% (0/59) |
| GEOLOCATION | 66.67% (52/78) | 33.33% (26/78) | 0.00% (0/78) |
| AFFILIATION | 72.05% (116/161) | 27.95% (45/161) | 0.00% (0/161) |
| RACE | 75.00% (3/4) | 25.00% (1/4) | 0.00% (0/4) |
| ETHNICITY | 100.00% (1/1) | 0.00% (0/1) | 0.00% (0/1) |
| ADDRESS | 100.00% (1/1) | 0.00% (0/1) | 0.00% (0/1) |
| INCOME | 0.00% (0/1) | 100.00% (1/1) | 0.00% (0/1) |
| AGE | 100.00% (1/1) | 0.00% (0/1) | 0.00% (0/1) |
| HEALTH_INFORMATION | 100.00% (1/1) | 0.00% (0/1) | 0.00% (0/1) |
| FINANCIAL_INFORMATION | 100.00% (1/1) | 0.00% (0/1) | 0.00% (0/1) |
| **Overall** | 69.90% (281/402) | 30.10% (121/402) | 0.00% (0/402) |

Table 15: Weighted Results per Type and Overall (Prediction for **CaseHOLD**), Model: **lgai/exaone-deep-32b**

| Type | Weighted Redact | Weighted Abstract | Weighted Retain |
|---|---|---|---|
| NAME | 0.00% (0/94) | 100.00% (94/94) | 0.00% (0/94) |
| TIME | 0.00% (0/59) | 100.00% (59/59) | 0.00% (0/59) |
| GEOLOCATION | 1.28% (1/78) | 98.72% (77/78) | 0.00% (0/78) |
| AFFILIATION | 0.00% (0/161) | 99.38% (160/161) | 0.62% (1/161) |
| RACE | 0.00% (0/4) | 100.00% (4/4) | 0.00% (0/4) |
| ETHNICITY | 0.00% (0/1) | 100.00% (1/1) | 0.00% (0/1) |
| ADDRESS | 0.00% (0/1) | 100.00% (1/1) | 0.00% (0/1) |
| INCOME | 0.00% (0/1) | 100.00% (1/1) | 0.00% (0/1) |
| AGE | 0.00% (0/1) | 100.00% (1/1) | 0.00% (0/1) |
| HEALTH_INFORMATION | 0.00% (0/1) | 100.00% (1/1) | 0.00% (0/1) |
| FINANCIAL_INFORMATION | 0.00% (0/1) | 100.00% (1/1) | 0.00% (0/1) |
| **Overall** | 0.25% (1/402) | 99.50% (400/402) | 0.25% (1/402) |

Table 16: Weighted Results per Type and Overall (Prediction for **CaseHOLD**), Model: **mistralai/mistral-small-3.1-24b-instruct**

| Type | Weighted Redact | Weighted Abstract | Weighted Retain |
|---|---|---|---|
| NAME | 3.19% (3/94) | 96.81% (91/94) | 0.00% (0/94) |
| TIME | 5.08% (3/59) | 94.92% (56/59) | 0.00% (0/59) |
| GEOLOCATION | 1.28% (1/78) | 98.72% (77/78) | 0.00% (0/78) |
| AFFILIATION | 4.97% (8/161) | 95.03% (153/161) | 0.00% (0/161) |
| RACE | 25.00% (1/4) | 75.00% (3/4) | 0.00% (0/4) |
| ETHNICITY | 0.00% (0/1) | 100.00% (1/1) | 0.00% (0/1) |
| ADDRESS | 0.00% (0/1) | 100.00% (1/1) | 0.00% (0/1) |
| INCOME | 0.00% (0/1) | 100.00% (1/1) | 0.00% (0/1) |
| AGE | 0.00% (0/1) | 100.00% (1/1) | 0.00% (0/1) |
| HEALTH_INFORMATION | 0.00% (0/1) | 100.00% (1/1) | 0.00% (0/1) |
| FINANCIAL_INFORMATION | 0.00% (0/1) | 100.00% (1/1) | 0.00% (0/1) |
| **Overall** | 3.98% (16/402) | 96.02% (386/402) | 0.00% (0/402) |

Table 17: Weighted Results per Type and Overall (Prediction for **CaseHOLD**), Model: **qwen/qwen2.5-7b-instruct**

## 1.2 MEDQA

| Type | Weighted Redact | Weighted Abstract | Weighted Retain |
|------|----------------|-------------------|-----------------|
| AGE | 96.46% (109/113) | 0.00% (0/113) | 3.54% (4/113) |
| GENDER | 98.28% (57/58) | 0.00% (0/58) | 1.72% (1/58) |
| OCCUPATION | 100.00% (9/9) | 0.00% (0/9) | 0.00% (0/9) |
| HEALTH_INFORMATION | 87.08% (647/743) | 2.96% (22/743) | 9.96% (74/743) |
| GEOLOCATION | 94.74% (18/19) | 0.00% (0/19) | 5.26% (1/19) |
| RACE | 100.00% (8/8) | 0.00% (0/8) | 0.00% (0/8) |
| MARITAL_STATUS | 100.00% (1/1) | 0.00% (0/1) | 0.00% (0/1) |
| TIME | 95.00% (19/20) | 0.00% (0/20) | 5.00% (1/20) |
| SEXUAL_ORIENTATION | 100.00% (3/3) | 0.00% (0/3) | 0.00% (0/3) |
| AFFILIATION | 100.00% (1/1) | 0.00% (0/1) | 0.00% (0/1) |
| DIETARY_PREFERENCE | 100.00% (1/1) | 0.00% (0/1) | 0.00% (0/1) |
| **Overall** | 89.45% (873/976) | 2.25% (22/976) | 8.30% (81/976) |

Table 18: Weighted Results per Type and Overall (Oracle for **MedQA**), Model: **gpt-4.1-nano**

| Type | Weighted Redact | Weighted Abstract | Weighted Retain |
|------|----------------|-------------------|-----------------|
| AGE | 98.23% (111/113) | 1.77% (2/113) | 0.00% (0/113) |
| GENDER | 96.55% (56/58) | 1.72% (1/58) | 1.72% (1/58) |
| OCCUPATION | 100.00% (9/9) | 0.00% (0/9) | 0.00% (0/9) |
| HEALTH_INFORMATION | 96.90% (720/743) | 1.48% (11/743) | 1.62% (12/743) |
| GEOLOCATION | 100.00% (19/19) | 0.00% (0/19) | 0.00% (0/19) |
| RACE | 100.00% (8/8) | 0.00% (0/8) | 0.00% (0/8) |
| MARITAL_STATUS | 100.00% (1/1) | 0.00% (0/1) | 0.00% (0/1) |
| TIME | 100.00% (20/20) | 0.00% (0/20) | 0.00% (0/20) |
| SEXUAL_ORIENTATION | 100.00% (3/3) | 0.00% (0/3) | 0.00% (0/3) |
| AFFILIATION | 100.00% (1/1) | 0.00% (0/1) | 0.00% (0/1) |
| DIETARY_PREFERENCE | 100.00% (1/1) | 0.00% (0/1) | 0.00% (0/1) |
| **Overall** | 97.23% (949/976) | 1.43% (14/976) | 1.33% (13/976) |

Table 19: Weighted Results per Type and Overall (Oracle for **MedQA**), Model: **gpt-4.1**

| Type | Weighted Redact | Weighted Abstract | Weighted Retain |
|------|----------------|-------------------|-----------------|
| AGE | 100.00% (113/113) | 0.00% (0/113) | 0.00% (0/113) |
| GENDER | 100.00% (58/58) | 0.00% (0/58) | 0.00% (0/58) |
| OCCUPATION | 100.00% (9/9) | 0.00% (0/9) | 0.00% (0/9) |
| HEALTH_INFORMATION | 95.42% (709/743) | 2.69% (20/743) | 1.88% (14/743) |
| GEOLOCATION | 94.74% (18/19) | 5.26% (1/19) | 0.00% (0/19) |
| RACE | 100.00% (8/8) | 0.00% (0/8) | 0.00% (0/8) |
| MARITAL_STATUS | 100.00% (1/1) | 0.00% (0/1) | 0.00% (0/1) |
| TIME | 100.00% (20/20) | 0.00% (0/20) | 0.00% (0/20) |
| SEXUAL_ORIENTATION | 100.00% (3/3) | 0.00% (0/3) | 0.00% (0/3) |
| AFFILIATION | 100.00% (1/1) | 0.00% (0/1) | 0.00% (0/1) |
| DIETARY_PREFERENCE | 100.00% (1/1) | 0.00% (0/1) | 0.00% (0/1) |
| **Overall** | 96.41% (941/976) | 2.15% (21/976) | 1.43% (14/976) |

Table 20: Weighted Results per Type and Overall (Oracle for **MedQA**), Model: **gpt-5**

| Type | Weighted Redact | Weighted Abstract | Weighted Retain |
|---|---|---|---|
| AGE | 79.65% (90/113) | 10.62% (12/113) | 9.73% (11/113) |
| GENDER | 70.69% (41/58) | 13.79% (8/58) | 15.52% (9/58) |
| OCCUPATION | 88.89% (8/9) | 11.11% (1/9) | 0.00% (0/9) |
| HEALTH_INFORMATION | 69.99% (520/743) | 14.27% (106/743) | 15.75% (117/743) |
| GEOLOCATION | 78.95% (15/19) | 21.05% (4/19) | 0.00% (0/19) |
| RACE | 87.50% (7/8) | 0.00% (0/8) | 12.50% (1/8) |
| MARITAL_STATUS | 100.00% (1/1) | 0.00% (0/1) | 0.00% (0/1) |
| TIME | 55.00% (11/20) | 25.00% (5/20) | 20.00% (4/20) |
| SEXUAL_ORIENTATION | 33.33% (1/3) | 33.33% (1/3) | 33.33% (1/3) |
| AFFILIATION | 100.00% (1/1) | 0.00% (0/1) | 0.00% (0/1) |
| DIETARY_PREFERENCE | 100.00% (1/1) | 0.00% (0/1) | 0.00% (0/1) |
| **Overall** | 71.31% (696/976) | 14.04% (137/976) | 14.65% (143/976) |

Table 21: Weighted Results per Type and Overall (Oracle for **MedQA**), Model: **claude-3-7-sonnet-20250219**

| Type | Weighted Redact | Weighted Abstract | Weighted Retain |
|---|---|---|---|
| AGE | 100.00% (113/113) | 0.00% (0/113) | 0.00% (0/113) |
| GENDER | 100.00% (58/58) | 0.00% (0/58) | 0.00% (0/58) |
| OCCUPATION | 100.00% (9/9) | 0.00% (0/9) | 0.00% (0/9) |
| HEALTH_INFORMATION | 94.75% (704/743) | 3.50% (26/743) | 1.75% (13/743) |
| GEOLOCATION | 100.00% (19/19) | 0.00% (0/19) | 0.00% (0/19) |
| RACE | 100.00% (8/8) | 0.00% (0/8) | 0.00% (0/8) |
| MARITAL_STATUS | 100.00% (1/1) | 0.00% (0/1) | 0.00% (0/1) |
| TIME | 100.00% (20/20) | 0.00% (0/20) | 0.00% (0/20) |
| SEXUAL_ORIENTATION | 100.00% (3/3) | 0.00% (0/3) | 0.00% (0/3) |
| AFFILIATION | 100.00% (1/1) | 0.00% (0/1) | 0.00% (0/1) |
| DIETARY_PREFERENCE | 100.00% (1/1) | 0.00% (0/1) | 0.00% (0/1) |
| **Overall** | 96.00% (937/976) | 2.66% (26/976) | 1.33% (13/976) |

Table 22: Weighted Results per Type and Overall (Oracle for **MedQA**), Model: **claude-sonnet-4-20250514**

| Type | Weighted Redact | Weighted Abstract | Weighted Retain |
|---|---|---|---|
| AGE | 80.53% (91/113) | 11.50% (13/113) | 7.96% (9/113) |
| GENDER | 82.76% (48/58) | 6.90% (4/58) | 10.34% (6/58) |
| OCCUPATION | 77.78% (7/9) | 22.22% (2/9) | 0.00% (0/9) |
| HEALTH_INFORMATION | 74.16% (551/743) | 11.98% (89/743) | 13.86% (103/743) |
| GEOLOCATION | 84.21% (16/19) | 5.26% (1/19) | 10.53% (2/19) |
| RACE | 100.00% (8/8) | 0.00% (0/8) | 0.00% (0/8) |
| MARITAL_STATUS | 100.00% (1/1) | 0.00% (0/1) | 0.00% (0/1) |
| TIME | 70.00% (14/20) | 15.00% (3/20) | 15.00% (3/20) |
| SEXUAL_ORIENTATION | 66.67% (2/3) | 0.00% (0/3) | 33.33% (1/3) |
| AFFILIATION | 100.00% (1/1) | 0.00% (0/1) | 0.00% (0/1) |
| DIETARY_PREFERENCE | 100.00% (1/1) | 0.00% (0/1) | 0.00% (0/1) |
| **Overall** | 75.82% (740/976) | 11.48% (112/976) | 12.70% (124/976) |

Table 23: Weighted Results per Type and Overall (Oracle for **MedQA**), Model: **lgai/exaone-deep-32b**

| Type | Weighted Redact | Weighted Abstract | Weighted Retain |
|---|---|---|---|
| AGE | 100.00% (113/113) | 0.00% (0/113) | 0.00% (0/113) |
| GENDER | 100.00% (58/58) | 0.00% (0/58) | 0.00% (0/58) |
| OCCUPATION | 88.89% (8/9) | 11.11% (1/9) | 0.00% (0/9) |
| HEALTH_INFORMATION | 96.37% (716/743) | 2.29% (17/743) | 1.35% (10/743) |
| GEOLOCATION | 100.00% (19/19) | 0.00% (0/19) | 0.00% (0/19) |
| RACE | 100.00% (8/8) | 0.00% (0/8) | 0.00% (0/8) |
| MARITAL_STATUS | 100.00% (1/1) | 0.00% (0/1) | 0.00% (0/1) |
| TIME | 100.00% (20/20) | 0.00% (0/20) | 0.00% (0/20) |
| SEXUAL_ORIENTATION | 100.00% (3/3) | 0.00% (0/3) | 0.00% (0/3) |
| AFFILIATION | 100.00% (1/1) | 0.00% (0/1) | 0.00% (0/1) |
| DIETARY_PREFERENCE | 0.00% (0/1) | 100.00% (1/1) | 0.00% (0/1) |
| **Overall** | 97.03% (947/976) | 1.95% (19/976) | 1.02% (10/976) |

Table 24: Weighted Results per Type and Overall (Oracle for **MedQA**), Model: **mistralai/mistral-small-3.1-24b-instruct**

| Type | Weighted Redact | Weighted Abstract | Weighted Retain |
|---|---|---|---|
| AGE | 94.69% (107/113) | 2.65% (3/113) | 2.65% (3/113) |
| GENDER | 93.10% (54/58) | 5.17% (3/58) | 1.72% (1/58) |
| OCCUPATION | 88.89% (8/9) | 11.11% (1/9) | 0.00% (0/9) |
| HEALTH_INFORMATION | 88.29% (656/743) | 6.86% (51/743) | 4.85% (36/743) |
| GEOLOCATION | 89.47% (17/19) | 10.53% (2/19) | 0.00% (0/19) |
| RACE | 100.00% (8/8) | 0.00% (0/8) | 0.00% (0/8) |
| MARITAL_STATUS | 100.00% (1/1) | 0.00% (0/1) | 0.00% (0/1) |
| TIME | 90.00% (18/20) | 10.00% (2/20) | 0.00% (0/20) |
| SEXUAL_ORIENTATION | 100.00% (3/3) | 0.00% (0/3) | 0.00% (0/3) |
| AFFILIATION | 100.00% (1/1) | 0.00% (0/1) | 0.00% (0/1) |
| DIETARY_PREFERENCE | 0.00% (0/1) | 100.00% (1/1) | 0.00% (0/1) |
| **Overall** | 89.45% (873/976) | 6.45% (63/976) | 4.10% (40/976) |

Table 25: Weighted Results per Type and Overall (Oracle for **MedQA**), Model: **qwen/qwen2.5-7b-instruct**

| Type | Weighted Redact | Weighted Abstract | Weighted Retain |
|---|---|---|---|
| AGE | 42.48% (48/113) | 13.27% (15/113) | 44.25% (50/113) |
| GENDER | 39.66% (23/58) | 10.34% (6/58) | 50.00% (29/58) |
| OCCUPATION | 55.56% (5/9) | 11.11% (1/9) | 33.33% (3/9) |
| HEALTH_INFORMATION | 24.09% (179/743) | 14.27% (106/743) | 61.64% (458/743) |
| GEOLOCATION | 31.58% (6/19) | 15.79% (3/19) | 52.63% (10/19) |
| RACE | 50.00% (4/8) | 12.50% (1/8) | 37.50% (3/8) |
| MARITAL_STATUS | 100.00% (1/1) | 0.00% (0/1) | 0.00% (0/1) |
| TIME | 45.00% (9/20) | 10.00% (2/20) | 45.00% (9/20) |
| SEXUAL_ORIENTATION | 33.33% (1/3) | 33.33% (1/3) | 33.33% (1/3) |
| AFFILIATION | 100.00% (1/1) | 0.00% (0/1) | 0.00% (0/1) |
| DIETARY_PREFERENCE | 100.00% (1/1) | 0.00% (0/1) | 0.00% (0/1) |
| **Overall** | 28.48% (278/976) | 13.83% (135/976) | 57.68% (563/976) |

Table 26: Weighted Results per Type and Overall (Oracle for **MedQA**), Model: **qwen/qwen2.5-0.5b-instruct**

| Type | Weighted Redact | Weighted Abstract | Weighted Retain |
|---|---|---|---|
| AGE | 41.59% (47/113) | 58.41% (66/113) | 0.00% (0/113) |
| GENDER | 37.93% (22/58) | 60.34% (35/58) | 1.72% (1/58) |
| OCCUPATION | 66.67% (6/9) | 33.33% (3/9) | 0.00% (0/9) |
| HEALTH_INFORMATION | 21.27% (158/743) | 60.97% (453/743) | 17.77% (132/743) |
| GEOLOCATION | 57.89% (11/19) | 42.11% (8/19) | 0.00% (0/19) |
| RACE | 37.50% (3/8) | 62.50% (5/8) | 0.00% (0/8) |
| MARITAL_STATUS | 100.00% (1/1) | 0.00% (0/1) | 0.00% (0/1) |
| TIME | 65.00% (13/20) | 35.00% (7/20) | 0.00% (0/20) |
| SEXUAL_ORIENTATION | 33.33% (1/3) | 66.67% (2/3) | 0.00% (0/3) |
| AFFILIATION | 100.00% (1/1) | 0.00% (0/1) | 0.00% (0/1) |
| DIETARY_PREFERENCE | 100.00% (1/1) | 0.00% (0/1) | 0.00% (0/1) |
| **Overall** | 27.05% (264/976) | 59.32% (579/976) | 13.63% (133/976) |

Table 27: Weighted Results per Type and Overall (Prediction for **MedQA**), Model: **gpt-4.1-nano**

| Type | Weighted Redact | Weighted Abstract | Weighted Retain |
|---|---|---|---|
| AGE | 0.00% (0/113) | 99.12% (112/113) | 0.88% (1/113) |
| GENDER | 0.00% (0/58) | 98.28% (57/58) | 1.72% (1/58) |
| OCCUPATION | 0.00% (0/9) | 100.00% (9/9) | 0.00% (0/9) |
| HEALTH_INFORMATION | 0.00% (0/743) | 98.52% (732/743) | 1.48% (11/743) |
| GEOLOCATION | 10.53% (2/19) | 89.47% (17/19) | 0.00% (0/19) |
| RACE | 0.00% (0/8) | 100.00% (8/8) | 0.00% (0/8) |
| MARITAL_STATUS | 0.00% (0/1) | 100.00% (1/1) | 0.00% (0/1) |
| TIME | 0.00% (0/20) | 100.00% (20/20) | 0.00% (0/20) |
| SEXUAL_ORIENTATION | 0.00% (0/3) | 100.00% (3/3) | 0.00% (0/3) |
| AFFILIATION | 0.00% (0/1) | 100.00% (1/1) | 0.00% (0/1) |
| DIETARY_PREFERENCE | 0.00% (0/1) | 100.00% (1/1) | 0.00% (0/1) |
| **Overall** | 0.20% (2/976) | 98.46% (961/976) | 1.33% (13/976) |

Table 28: Weighted Results per Type and Overall (Prediction for **MedQA**), Model: **gpt-4.1**

| Type | Weighted Redact | Weighted Abstract | Weighted Retain |
|---|---|---|---|
| AGE | 57.52% (65/113) | 41.59% (47/113) | 0.88% (1/113) |
| GENDER | 74.14% (43/58) | 22.41% (13/58) | 3.45% (2/58) |
| OCCUPATION | 66.67% (6/9) | 33.33% (3/9) | 0.00% (0/9) |
| HEALTH_INFORMATION | 51.82% (385/743) | 43.47% (323/743) | 4.71% (35/743) |
| GEOLOCATION | 57.89% (11/19) | 42.11% (8/19) | 0.00% (0/19) |
| RACE | 100.00% (8/8) | 0.00% (0/8) | 0.00% (0/8) |
| MARITAL_STATUS | 100.00% (1/1) | 0.00% (0/1) | 0.00% (0/1) |
| TIME | 50.00% (10/20) | 50.00% (10/20) | 0.00% (0/20) |
| SEXUAL_ORIENTATION | 100.00% (3/3) | 0.00% (0/3) | 0.00% (0/3) |
| AFFILIATION | 100.00% (1/1) | 0.00% (0/1) | 0.00% (0/1) |
| DIETARY_PREFERENCE | 0.00% (0/1) | 100.00% (1/1) | 0.00% (0/1) |
| **Overall** | 54.61% (533/976) | 41.50% (405/976) | 3.89% (38/976) |

Table 29: Weighted Results per Type and Overall (Prediction for **MedQA**), Model: **gpt-5**

| Type | Weighted Redact | Weighted Abstract | Weighted Retain |
|---|---|---|---|
| AGE | 0.00% (0/113) | 30.09% (34/113) | 69.91% (79/113) |
| GENDER | 0.00% (0/58) | 22.41% (13/58) | 77.59% (45/58) |
| OCCUPATION | 0.00% (0/9) | 55.56% (5/9) | 44.44% (4/9) |
| HEALTH_INFORMATION | 0.00% (0/743) | 2.96% (22/743) | 97.04% (721/743) |
| GEOLOCATION | 0.00% (0/19) | 52.63% (10/19) | 47.37% (9/19) |
| RACE | 25.00% (2/8) | 62.50% (5/8) | 12.50% (1/8) |
| MARITAL_STATUS | 0.00% (0/1) | 100.00% (1/1) | 0.00% (0/1) |
| TIME | 0.00% (0/20) | 20.00% (4/20) | 80.00% (16/20) |
| SEXUAL_ORIENTATION | 0.00% (0/3) | 66.67% (2/3) | 33.33% (1/3) |
| AFFILIATION | 0.00% (0/1) | 100.00% (1/1) | 0.00% (0/1) |
| DIETARY_PREFERENCE | 0.00% (0/1) | 0.00% (0/1) | 100.00% (1/1) |
| **Overall** | 0.20% (2/976) | 9.94% (97/976) | 89.86% (877/976) |

Table 30: Weighted Results per Type and Overall (Prediction for **MedQA**), Model: **claude-3-7-sonnet-20250219**

| Type | Weighted Redact | Weighted Abstract | Weighted Retain |
|---|---|---|---|
| AGE | 0.00% (0/113) | 70.80% (80/113) | 29.20% (33/113) |
| GENDER | 1.72% (1/58) | 74.14% (43/58) | 24.14% (14/58) |
| OCCUPATION | 0.00% (0/9) | 100.00% (9/9) | 0.00% (0/9) |
| HEALTH_INFORMATION | 0.00% (0/743) | 15.21% (113/743) | 84.79% (630/743) |
| GEOLOCATION | 5.26% (1/19) | 63.16% (12/19) | 31.58% (6/19) |
| RACE | 87.50% (7/8) | 0.00% (0/8) | 12.50% (1/8) |
| MARITAL_STATUS | 0.00% (0/1) | 100.00% (1/1) | 0.00% (0/1) |
| TIME | 0.00% (0/20) | 40.00% (8/20) | 60.00% (12/20) |
| SEXUAL_ORIENTATION | 0.00% (0/3) | 100.00% (3/3) | 0.00% (0/3) |
| AFFILIATION | 0.00% (0/1) | 100.00% (1/1) | 0.00% (0/1) |
| DIETARY_PREFERENCE | 0.00% (0/1) | 0.00% (0/1) | 100.00% (1/1) |
| **Overall** | 0.92% (9/976) | 27.66% (270/976) | 71.41% (697/976) |

Table 31: Weighted Results per Type and Overall (Prediction for **MedQA**), Model: **claude-sonnet-4-20250514**

| Type | Weighted Redact | Weighted Abstract | Weighted Retain |
|---|---|---|---|
| AGE | 84.07% (95/113) | 15.93% (18/113) | 0.00% (0/113) |
| GENDER | 82.76% (48/58) | 15.52% (9/58) | 1.72% (1/58) |
| OCCUPATION | 88.89% (8/9) | 11.11% (1/9) | 0.00% (0/9) |
| HEALTH_INFORMATION | 45.76% (340/743) | 54.10% (402/743) | 0.13% (1/743) |
| GEOLOCATION | 94.74% (18/19) | 5.26% (1/19) | 0.00% (0/19) |
| RACE | 87.50% (7/8) | 12.50% (1/8) | 0.00% (0/8) |
| MARITAL_STATUS | 100.00% (1/1) | 0.00% (0/1) | 0.00% (0/1) |
| TIME | 80.00% (16/20) | 20.00% (4/20) | 0.00% (0/20) |
| SEXUAL_ORIENTATION | 66.67% (2/3) | 33.33% (1/3) | 0.00% (0/3) |
| AFFILIATION | 100.00% (1/1) | 0.00% (0/1) | 0.00% (0/1) |
| DIETARY_PREFERENCE | 0.00% (0/1) | 100.00% (1/1) | 0.00% (0/1) |
| **Overall** | 54.92% (536/976) | 44.88% (438/976) | 0.20% (2/976) |

Table 32: Weighted Results per Type and Overall (Prediction for **MedQA**), Model: **lgai/exaone-deep-32b**

| Type | Weighted Redact | Weighted Abstract | Weighted Retain |
|---|---|---|---|
| AGE | 0.00% (0/113) | 100.00% (113/113) | 0.00% (0/113) |
| GENDER | 0.00% (0/58) | 100.00% (58/58) | 0.00% (0/58) |
| OCCUPATION | 0.00% (0/9) | 100.00% (9/9) | 0.00% (0/9) |
| HEALTH_INFORMATION | 0.00% (0/743) | 91.39% (679/743) | 8.61% (64/743) |
| GEOLOCATION | 0.00% (0/19) | 100.00% (19/19) | 0.00% (0/19) |
| RACE | 0.00% (0/8) | 100.00% (8/8) | 0.00% (0/8) |
| MARITAL_STATUS | 0.00% (0/1) | 100.00% (1/1) | 0.00% (0/1) |
| TIME | 0.00% (0/20) | 100.00% (20/20) | 0.00% (0/20) |
| SEXUAL_ORIENTATION | 0.00% (0/3) | 100.00% (3/3) | 0.00% (0/3) |
| AFFILIATION | 0.00% (0/1) | 100.00% (1/1) | 0.00% (0/1) |
| DIETARY_PREFERENCE | 0.00% (0/1) | 100.00% (1/1) | 0.00% (0/1) |
| **Overall** | 0.00% (0/976) | 93.44% (912/976) | 6.56% (64/976) |

Table 33: Weighted Results per Type and Overall (Prediction for **MedQA**), Model: **mistralai/mistral-small-3.1-24b-instruct**

| Type | Weighted Redact | Weighted Abstract | Weighted Retain |
|---|---|---|---|
| AGE | 0.00% (0/113) | 100.00% (113/113) | 0.00% (0/113) |
| GENDER | 0.00% (0/58) | 96.55% (56/58) | 3.45% (2/58) |
| OCCUPATION | 0.00% (0/9) | 100.00% (9/9) | 0.00% (0/9) |
| HEALTH_INFORMATION | 0.00% (0/743) | 99.87% (742/743) | 0.13% (1/743) |
| GEOLOCATION | 0.00% (0/19) | 100.00% (19/19) | 0.00% (0/19) |
| RACE | 0.00% (0/8) | 100.00% (8/8) | 0.00% (0/8) |
| MARITAL_STATUS | 0.00% (0/1) | 100.00% (1/1) | 0.00% (0/1) |
| TIME | 0.00% (0/20) | 100.00% (20/20) | 0.00% (0/20) |
| SEXUAL_ORIENTATION | 0.00% (0/3) | 100.00% (3/3) | 0.00% (0/3) |
| AFFILIATION | 0.00% (0/1) | 100.00% (1/1) | 0.00% (0/1) |
| DIETARY_PREFERENCE | 0.00% (0/1) | 100.00% (1/1) | 0.00% (0/1) |
| **Overall** | 0.00% (0/976) | 99.69% (973/976) | 0.31% (3/976) |

Table 34: Weighted Results per Type and Overall (Prediction for **MedQA**), Model: **qwen/qwen2.5-7b-instruct**

## 1.3 SHAREGPT

| Type | Weighted Redact | Weighted Abstract | Weighted Retain |
|------|-----------------|-------------------|-----------------|
| NAME | 91.28% (157/172) | 5.23% (9/172) | 3.49% (6/172) |
| AFFILIATION | 90.64% (155/171) | 5.26% (9/171) | 4.09% (7/171) |
| TIME | 67.05% (177/264) | 7.58% (20/264) | 25.38% (67/264) |
| URL | 75.00% (15/20) | 20.00% (4/20) | 5.00% (1/20) |
| EMAIL | 100.00% (4/4) | 0.00% (0/4) | 0.00% (0/4) |
| GEOLOCATION | 66.67% (218/327) | 17.13% (56/327) | 16.21% (53/327) |
| RELIGION | 0.00% (0/2) | 50.00% (1/2) | 50.00% (1/2) |
| FINANCIAL_INFORMATION | 72.73% (8/11) | 18.18% (2/11) | 9.09% (1/11) |
| MARITAL_STATUS | 63.64% (7/11) | 27.27% (3/11) | 9.09% (1/11) |
| OCCUPATION | 83.33% (50/60) | 11.67% (7/60) | 5.00% (3/60) |
| VEHICLE | 0.00% (0/1) | 100.00% (1/1) | 0.00% (0/1) |
| INCOME | 87.50% (7/8) | 12.50% (1/8) | 0.00% (0/8) |
| HEALTH_INFORMATION | 65.31% (32/49) | 16.33% (8/49) | 18.37% (9/49) |
| EDUCATIONAL_RECORD | 100.00% (10/10) | 0.00% (0/10) | 0.00% (0/10) |
| AGE | 67.86% (38/56) | 26.79% (15/56) | 5.36% (3/56) |
| GENDER | 61.54% (8/13) | 23.08% (3/13) | 15.38% (2/13) |
| ETHNICITY | 80.00% (4/5) | 20.00% (1/5) | 0.00% (0/5) |
| ADDRESS | 0.00% (0/1) | 100.00% (1/1) | 0.00% (0/1) |
| IP_ADDRESS | 66.67% (2/3) | 0.00% (0/3) | 33.33% (1/3) |
| RACE | 50.00% (1/2) | 50.00% (1/2) | 0.00% (0/2) |
| **Overall** | 75.04% (893/1190) | 11.93% (142/1190) | 13.03% (155/1190) |

Table 35: Weighted Results per Type and Overall (Oracle for **ShareGPT90K**), Model: **gpt-4.1-nano**

| Type | Weighted Redact | Weighted Abstract | Weighted Retain |
|------|-----------------|-------------------|-----------------|
| NAME | 87.79% (151/172) | 6.98% (12/172) | 5.23% (9/172) |
| AFFILIATION | 96.49% (165/171) | 1.75% (3/171) | 1.75% (3/171) |
| TIME | 78.03% (206/264) | 10.61% (28/264) | 11.36% (30/264) |
| URL | 85.00% (17/20) | 15.00% (3/20) | 0.00% (0/20) |
| EMAIL | 100.00% (4/4) | 0.00% (0/4) | 0.00% (0/4) |
| GEOLOCATION | 66.97% (219/327) | 22.32% (73/327) | 10.70% (35/327) |
| RELIGION | 50.00% (1/2) | 50.00% (1/2) | 0.00% (0/2) |
| FINANCIAL_INFORMATION | 81.82% (9/11) | 18.18% (2/11) | 0.00% (0/11) |
| MARITAL_STATUS | 63.64% (7/11) | 0.00% (0/11) | 36.36% (4/11) |
| OCCUPATION | 86.67% (52/60) | 8.33% (5/60) | 5.00% (3/60) |
| VEHICLE | 0.00% (0/1) | 100.00% (1/1) | 0.00% (0/1) |
| INCOME | 100.00% (8/8) | 0.00% (0/8) | 0.00% (0/8) |
| HEALTH_INFORMATION | 81.63% (40/49) | 10.20% (5/49) | 8.16% (4/49) |
| EDUCATIONAL_RECORD | 90.00% (9/10) | 10.00% (1/10) | 0.00% (0/10) |
| AGE | 78.57% (44/56) | 14.29% (8/56) | 7.14% (4/56) |
| GENDER | 92.31% (12/13) | 0.00% (0/13) | 7.69% (1/13) |
| ETHNICITY | 100.00% (5/5) | 0.00% (0/5) | 0.00% (0/5) |
| ADDRESS | 100.00% (1/1) | 0.00% (0/1) | 0.00% (0/1) |
| IP_ADDRESS | 66.67% (2/3) | 0.00% (0/3) | 33.33% (1/3) |
| RACE | 100.00% (2/2) | 0.00% (0/2) | 0.00% (0/2) |
| **Overall** | 80.17% (954/1190) | 11.93% (142/1190) | 7.90% (94/1190) |

Table 36: Weighted Results per Type and Overall (Oracle for **ShareGPT90K**), Model: **gpt-4.1**

| Type | Weighted Redact | Weighted Abstract | Weighted Retain |
|---|---|---|---|
| NAME | 93.60% (161/172) | 5.81% (10/172) | 0.58% (1/172) |
| AFFILIATION | 95.91% (164/171) | 2.92% (5/171) | 1.17% (2/171) |
| TIME | 77.65% (205/264) | 10.61% (28/264) | 11.74% (31/264) |
| URL | 100.00% (20/20) | 0.00% (0/20) | 0.00% (0/20) |
| EMAIL | 100.00% (4/4) | 0.00% (0/4) | 0.00% (0/4) |
| GEOLOCATION | 68.81% (225/327) | 18.04% (59/327) | 13.15% (43/327) |
| RELIGION | 100.00% (2/2) | 0.00% (0/2) | 0.00% (0/2) |
| FINANCIAL_INFORMATION | 81.82% (9/11) | 18.18% (2/11) | 0.00% (0/11) |
| MARITAL_STATUS | 90.91% (10/11) | 0.00% (0/11) | 9.09% (1/11) |
| OCCUPATION | 88.33% (53/60) | 8.33% (5/60) | 3.33% (2/60) |
| VEHICLE | 0.00% (0/1) | 100.00% (1/1) | 0.00% (0/1) |
| INCOME | 87.50% (7/8) | 12.50% (1/8) | 0.00% (0/8) |
| HEALTH_INFORMATION | 87.76% (43/49) | 6.12% (3/49) | 6.12% (3/49) |
| EDUCATIONAL_RECORD | 100.00% (10/10) | 0.00% (0/10) | 0.00% (0/10) |
| AGE | 91.07% (51/56) | 5.36% (3/56) | 3.57% (2/56) |
| GENDER | 92.31% (12/13) | 7.69% (1/13) | 0.00% (0/13) |
| ETHNICITY | 100.00% (5/5) | 0.00% (0/5) | 0.00% (0/5) |
| ADDRESS | 100.00% (1/1) | 0.00% (0/1) | 0.00% (0/1) |
| IP_ADDRESS | 100.00% (3/3) | 0.00% (0/3) | 0.00% (0/3) |
| RACE | 100.00% (2/2) | 0.00% (0/2) | 0.00% (0/2) |
| **Overall** | 82.94% (987/1190) | 9.92% (118/1190) | 7.14% (85/1190) |

Table 37: Weighted Results per Type and Overall (Oracle for **ShareGPT90K**), Model: **gpt-5**

| Type | Weighted Redact | Weighted Abstract | Weighted Retain |
|---|---|---|---|
| NAME | 80.81% (139/172) | 9.88% (17/172) | 9.30% (16/172) |
| AFFILIATION | 94.15% (161/171) | 2.92% (5/171) | 2.92% (5/171) |
| TIME | 76.14% (201/264) | 8.33% (22/264) | 15.53% (41/264) |
| URL | 85.00% (17/20) | 10.00% (2/20) | 5.00% (1/20) |
| EMAIL | 100.00% (4/4) | 0.00% (0/4) | 0.00% (0/4) |
| GEOLOCATION | 63.00% (206/327) | 18.35% (60/327) | 18.65% (61/327) |
| RELIGION | 0.00% (0/2) | 0.00% (0/2) | 100.00% (2/2) |
| FINANCIAL_INFORMATION | 63.64% (7/11) | 27.27% (3/11) | 9.09% (1/11) |
| MARITAL_STATUS | 81.82% (9/11) | 18.18% (2/11) | 0.00% (0/11) |
| OCCUPATION | 88.33% (53/60) | 11.67% (7/60) | 0.00% (0/60) |
| VEHICLE | 0.00% (0/1) | 100.00% (1/1) | 0.00% (0/1) |
| INCOME | 87.50% (7/8) | 12.50% (1/8) | 0.00% (0/8) |
| HEALTH_INFORMATION | 65.31% (32/49) | 12.24% (6/49) | 22.45% (11/49) |
| EDUCATIONAL_RECORD | 90.00% (9/10) | 0.00% (0/10) | 10.00% (1/10) |
| AGE | 67.86% (38/56) | 21.43% (12/56) | 10.71% (6/56) |
| GENDER | 76.92% (10/13) | 15.38% (2/13) | 7.69% (1/13) |
| ETHNICITY | 60.00% (3/5) | 0.00% (0/5) | 40.00% (2/5) |
| ADDRESS | 100.00% (1/1) | 0.00% (0/1) | 0.00% (0/1) |
| IP_ADDRESS | 33.33% (1/3) | 0.00% (0/3) | 66.67% (2/3) |
| RACE | 50.00% (1/2) | 50.00% (1/2) | 0.00% (0/2) |
| **Overall** | 75.55% (899/1190) | 11.85% (141/1190) | 12.61% (150/1190) |

Table 38: Weighted Results per Type and Overall (Oracle for **ShareGPT90K**), Model: **claude-3-7-sonnet-20250219**

| Type | Weighted Redact | Weighted Abstract | Weighted Retain |
|------|-----------------|-------------------|-----------------|
| NAME | 82.56% (142/172) | 6.40% (11/172) | 11.05% (19/172) |
| AFFILIATION | 91.23% (156/171) | 7.60% (13/171) | 1.17% (2/171) |
| TIME | 61.74% (163/264) | 14.39% (38/264) | 23.86% (63/264) |
| URL | 75.00% (15/20) | 15.00% (3/20) | 10.00% (2/20) |
| EMAIL | 100.00% (4/4) | 0.00% (0/4) | 0.00% (0/4) |
| GEOLOCATION | 58.41% (191/327) | 17.13% (56/327) | 24.46% (80/327) |
| RELIGION | 100.00% (2/2) | 0.00% (0/2) | 0.00% (0/2) |
| FINANCIAL_INFORMATION | 63.64% (7/11) | 36.36% (4/11) | 0.00% (0/11) |
| MARITAL_STATUS | 81.82% (9/11) | 9.09% (1/11) | 9.09% (1/11) |
| OCCUPATION | 80.00% (48/60) | 16.67% (10/60) | 3.33% (2/60) |
| VEHICLE | 100.00% (1/1) | 0.00% (0/1) | 0.00% (0/1) |
| INCOME | 75.00% (6/8) | 12.50% (1/8) | 12.50% (1/8) |
| HEALTH_INFORMATION | 65.31% (32/49) | 12.24% (6/49) | 22.45% (11/49) |
| EDUCATIONAL_RECORD | 90.00% (9/10) | 0.00% (0/10) | 10.00% (1/10) |
| AGE | 66.07% (37/56) | 12.50% (7/56) | 21.43% (12/56) |
| GENDER | 69.23% (9/13) | 7.69% (1/13) | 23.08% (3/13) |
| ETHNICITY | 60.00% (3/5) | 40.00% (2/5) | 0.00% (0/5) |
| ADDRESS | 100.00% (1/1) | 0.00% (0/1) | 0.00% (0/1) |
| IP_ADDRESS | 0.00% (0/3) | 0.00% (0/3) | 100.00% (3/3) |
| RACE | 50.00% (1/2) | 0.00% (0/2) | 50.00% (1/2) |
| **Overall** | 70.25% (836/1190) | 12.86% (153/1190) | 16.89% (201/1190) |

Table 39: Weighted Results per Type and Overall (Oracle for **ShareGPT90K**), Model: **claude-sonnet-4-20250514**

| Type | Weighted Redact | Weighted Abstract | Weighted Retain |
|------|-----------------|-------------------|-----------------|
| NAME | 56.40% (97/172) | 18.02% (31/172) | 25.58% (44/172) |
| AFFILIATION | 72.51% (124/171) | 14.62% (25/171) | 12.87% (22/171) |
| TIME | 46.21% (122/264) | 28.79% (76/264) | 25.00% (66/264) |
| URL | 60.00% (12/20) | 25.00% (5/20) | 15.00% (3/20) |
| EMAIL | 75.00% (3/4) | 0.00% (0/4) | 25.00% (1/4) |
| GEOLOCATION | 45.57% (149/327) | 18.04% (59/327) | 36.39% (119/327) |
| RELIGION | 0.00% (0/2) | 100.00% (2/2) | 0.00% (0/2) |
| FINANCIAL_INFORMATION | 36.36% (4/11) | 27.27% (3/11) | 36.36% (4/11) |
| MARITAL_STATUS | 36.36% (4/11) | 18.18% (2/11) | 45.45% (5/11) |
| OCCUPATION | 70.00% (42/60) | 6.67% (4/60) | 23.33% (14/60) |
| VEHICLE | 0.00% (0/1) | 0.00% (0/1) | 100.00% (1/1) |
| INCOME | 87.50% (7/8) | 12.50% (1/8) | 0.00% (0/8) |
| HEALTH_INFORMATION | 63.27% (31/49) | 10.20% (5/49) | 26.53% (13/49) |
| EDUCATIONAL_RECORD | 80.00% (8/10) | 0.00% (0/10) | 20.00% (2/10) |
| AGE | 48.21% (27/56) | 30.36% (17/56) | 21.43% (12/56) |
| GENDER | 76.92% (10/13) | 0.00% (0/13) | 23.08% (3/13) |
| ETHNICITY | 60.00% (3/5) | 20.00% (1/5) | 20.00% (1/5) |
| ADDRESS | 100.00% (1/1) | 0.00% (0/1) | 0.00% (0/1) |
| IP_ADDRESS | 33.33% (1/3) | 0.00% (0/3) | 66.67% (2/3) |
| RACE | 50.00% (1/2) | 0.00% (0/2) | 50.00% (1/2) |
| **Overall** | 54.29% (646/1190) | 19.41% (231/1190) | 26.30% (313/1190) |

Table 40: Weighted Results per Type and Overall (Oracle for **ShareGPT90K**), Model: **lgai/exaone-deep-32b**

| Type | Weighted Redact | Weighted Abstract | Weighted Retain |
|---|---|---|---|
| NAME | 79.07% (136/172) | 6.98% (12/172) | 13.95% (24/172) |
| AFFILIATION | 89.47% (153/171) | 7.02% (12/171) | 3.51% (6/171) |
| TIME | 60.98% (161/264) | 19.70% (52/264) | 19.32% (51/264) |
| URL | 95.00% (19/20) | 0.00% (0/20) | 5.00% (1/20) |
| EMAIL | 100.00% (4/4) | 0.00% (0/4) | 0.00% (0/4) |
| GEOLOCATION | 55.05% (180/327) | 22.32% (73/327) | 22.63% (74/327) |
| RELIGION | 100.00% (2/2) | 0.00% (0/2) | 0.00% (0/2) |
| FINANCIAL_INFORMATION | 54.55% (6/11) | 18.18% (2/11) | 27.27% (3/11) |
| MARITAL_STATUS | 63.64% (7/11) | 18.18% (2/11) | 18.18% (2/11) |
| OCCUPATION | 86.67% (52/60) | 10.00% (6/60) | 3.33% (2/60) |
| VEHICLE | 0.00% (0/1) | 0.00% (0/1) | 100.00% (1/1) |
| INCOME | 87.50% (7/8) | 12.50% (1/8) | 0.00% (0/8) |
| HEALTH_INFORMATION | 63.27% (31/49) | 22.45% (11/49) | 14.29% (7/49) |
| EDUCATIONAL_RECORD | 90.00% (9/10) | 10.00% (1/10) | 0.00% (0/10) |
| AGE | 62.50% (35/56) | 19.64% (11/56) | 17.86% (10/56) |
| GENDER | 61.54% (8/13) | 15.38% (2/13) | 23.08% (3/13) |
| ETHNICITY | 60.00% (3/5) | 40.00% (2/5) | 0.00% (0/5) |
| ADDRESS | 100.00% (1/1) | 0.00% (0/1) | 0.00% (0/1) |
| IP_ADDRESS | 0.00% (0/3) | 66.67% (2/3) | 33.33% (1/3) |
| RACE | 0.00% (0/2) | 50.00% (1/2) | 50.00% (1/2) |
| **Overall** | 68.40% (814/1190) | 15.97% (190/1190) | 15.63% (186/1190) |

Table 41: Weighted Results per Type and Overall (Oracle for **ShareGPT90K**), Model: **mistralai/mistral-small-3.1-24b-instruct**

| Type | Weighted Redact | Weighted Abstract | Weighted Retain |
|---|---|---|---|
| NAME | 75.00% (129/172) | 9.88% (17/172) | 15.12% (26/172) |
| AFFILIATION | 81.87% (140/171) | 11.70% (20/171) | 6.43% (11/171) |
| TIME | 52.65% (139/264) | 15.53% (41/264) | 31.82% (84/264) |
| URL | 80.00% (16/20) | 15.00% (3/20) | 5.00% (1/20) |
| EMAIL | 100.00% (4/4) | 0.00% (0/4) | 0.00% (0/4) |
| GEOLOCATION | 53.21% (174/327) | 18.96% (62/327) | 27.83% (91/327) |
| RELIGION | 100.00% (2/2) | 0.00% (0/2) | 0.00% (0/2) |
| FINANCIAL_INFORMATION | 54.55% (6/11) | 9.09% (1/11) | 36.36% (4/11) |
| MARITAL_STATUS | 54.55% (6/11) | 9.09% (1/11) | 36.36% (4/11) |
| OCCUPATION | 65.00% (39/60) | 15.00% (9/60) | 20.00% (12/60) |
| VEHICLE | 100.00% (1/1) | 0.00% (0/1) | 0.00% (0/1) |
| INCOME | 100.00% (8/8) | 0.00% (0/8) | 0.00% (0/8) |
| HEALTH_INFORMATION | 61.22% (30/49) | 12.24% (6/49) | 26.53% (13/49) |
| EDUCATIONAL_RECORD | 90.00% (9/10) | 0.00% (0/10) | 10.00% (1/10) |
| AGE | 53.57% (30/56) | 23.21% (13/56) | 23.21% (13/56) |
| GENDER | 69.23% (9/13) | 7.69% (1/13) | 23.08% (3/13) |
| ETHNICITY | 60.00% (3/5) | 20.00% (1/5) | 20.00% (1/5) |
| ADDRESS | 100.00% (1/1) | 0.00% (0/1) | 0.00% (0/1) |
| IP_ADDRESS | 33.33% (1/3) | 33.33% (1/3) | 33.33% (1/3) |
| RACE | 50.00% (1/2) | 0.00% (0/2) | 50.00% (1/2) |
| **Overall** | 62.86% (748/1190) | 14.79% (176/1190) | 22.35% (266/1190) |

Table 42: Weighted Results per Type and Overall (Oracle for **ShareGPT90K**), Model: **qwen/qwen2.5-7b-instruct**

| Type | Weighted Redact | Weighted Abstract | Weighted Retain |
|---|---|---|---|
| NAME | 17.44% (30/172) | 8.14% (14/172) | 74.42% (128/172) |
| AFFILIATION | 18.71% (32/171) | 14.04% (24/171) | 67.25% (115/171) |
| TIME | 10.23% (27/264) | 6.44% (17/264) | 83.33% (220/264) |
| URL | 30.00% (6/20) | 0.00% (0/20) | 70.00% (14/20) |
| EMAIL | 0.00% (0/4) | 25.00% (1/4) | 75.00% (3/4) |
| GEOLOCATION | 8.26% (27/327) | 6.42% (21/327) | 85.32% (279/327) |
| RELIGION | 0.00% (0/2) | 0.00% (0/2) | 100.00% (2/2) |
| FINANCIAL_INFORMATION | 9.09% (1/11) | 0.00% (0/11) | 90.91% (10/11) |
| MARITAL_STATUS | 9.09% (1/11) | 9.09% (1/11) | 81.82% (9/11) |
| OCCUPATION | 6.67% (4/60) | 3.33% (2/60) | 90.00% (54/60) |
| VEHICLE | 0.00% (0/1) | 0.00% (0/1) | 100.00% (1/1) |
| INCOME | 0.00% (0/8) | 0.00% (0/8) | 100.00% (8/8) |
| HEALTH_INFORMATION | 22.45% (11/49) | 12.24% (6/49) | 65.31% (32/49) |
| EDUCATIONAL_RECORD | 50.00% (5/10) | 10.00% (1/10) | 40.00% (4/10) |
| AGE | 10.71% (6/56) | 8.93% (5/56) | 80.36% (45/56) |
| GENDER | 7.69% (1/13) | 7.69% (1/13) | 84.62% (11/13) |
| ETHNICITY | 0.00% (0/5) | 20.00% (1/5) | 80.00% (4/5) |
| ADDRESS | 0.00% (0/1) | 0.00% (0/1) | 100.00% (1/1) |
| IP_ADDRESS | 0.00% (0/3) | 0.00% (0/3) | 100.00% (3/3) |
| RACE | 50.00% (1/2) | 0.00% (0/2) | 50.00% (1/2) |
| **Overall** | 12.77% (152/1190) | 7.90% (94/1190) | 79.33% (944/1190) |

Table 43: Weighted Results per Type and Overall (Oracle for **ShareGPT90K**), Model: **qwen/qwen2.5-0.5b-instruct**

| Type | Weighted Redact | Weighted Abstract | Weighted Retain |
|---|---|---|---|
| NAME | 69.77% (120/172) | 29.07% (50/172) | 1.16% (2/172) |
| AFFILIATION | 53.80% (92/171) | 45.03% (77/171) | 1.17% (2/171) |
| TIME | 45.83% (121/264) | 53.41% (141/264) | 0.76% (2/264) |
| URL | 75.00% (15/20) | 25.00% (5/20) | 0.00% (0/20) |
| EMAIL | 100.00% (4/4) | 0.00% (0/4) | 0.00% (0/4) |
| GEOLOCATION | 42.51% (139/327) | 57.49% (188/327) | 0.00% (0/327) |
| RELIGION | 100.00% (2/2) | 0.00% (0/2) | 0.00% (0/2) |
| FINANCIAL_INFORMATION | 81.82% (9/11) | 18.18% (2/11) | 0.00% (0/11) |
| MARITAL_STATUS | 63.64% (7/11) | 36.36% (4/11) | 0.00% (0/11) |
| OCCUPATION | 71.67% (43/60) | 21.67% (13/60) | 6.67% (4/60) |
| VEHICLE | 0.00% (0/1) | 100.00% (1/1) | 0.00% (0/1) |
| INCOME | 62.50% (5/8) | 37.50% (3/8) | 0.00% (0/8) |
| HEALTH_INFORMATION | 34.69% (17/49) | 65.31% (32/49) | 0.00% (0/49) |
| EDUCATIONAL_RECORD | 20.00% (2/10) | 80.00% (8/10) | 0.00% (0/10) |
| AGE | 53.57% (30/56) | 46.43% (26/56) | 0.00% (0/56) |
| GENDER | 38.46% (5/13) | 61.54% (8/13) | 0.00% (0/13) |
| ETHNICITY | 80.00% (4/5) | 20.00% (1/5) | 0.00% (0/5) |
| ADDRESS | 0.00% (0/1) | 100.00% (1/1) | 0.00% (0/1) |
| IP_ADDRESS | 0.00% (0/3) | 100.00% (3/3) | 0.00% (0/3) |
| RACE | 0.00% (0/2) | 100.00% (2/2) | 0.00% (0/2) |
| **Overall** | 51.68% (615/1190) | 47.48% (565/1190) | 0.84% (10/1190) |

Table 44: Weighted Results per Type and Overall (Prediction for **ShareGPT90K**), Model: **gpt-4.1-nano**

| Type | Weighted Redact | Weighted Abstract | Weighted Retain |
|---|---|---|---|
| NAME | 1.74% (3/172) | 97.67% (168/172) | 0.58% (1/172) |
| AFFILIATION | 0.00% (0/171) | 98.83% (169/171) | 1.17% (2/171) |
| TIME | 0.00% (0/264) | 100.00% (264/264) | 0.00% (0/264) |
| URL | 0.00% (0/20) | 100.00% (20/20) | 0.00% (0/20) |
| EMAIL | 0.00% (0/4) | 100.00% (4/4) | 0.00% (0/4) |
| GEOLOCATION | 0.00% (0/327) | 91.13% (298/327) | 8.87% (29/327) |
| RELIGION | 0.00% (0/2) | 100.00% (2/2) | 0.00% (0/2) |
| FINANCIAL_INFORMATION | 0.00% (0/11) | 100.00% (11/11) | 0.00% (0/11) |
| MARITAL_STATUS | 0.00% (0/11) | 100.00% (11/11) | 0.00% (0/11) |
| OCCUPATION | 0.00% (0/60) | 96.67% (58/60) | 3.33% (2/60) |
| VEHICLE | 0.00% (0/1) | 100.00% (1/1) | 0.00% (0/1) |
| INCOME | 0.00% (0/8) | 100.00% (8/8) | 0.00% (0/8) |
| HEALTH_INFORMATION | 0.00% (0/49) | 79.59% (39/49) | 20.41% (10/49) |
| EDUCATIONAL_RECORD | 0.00% (0/10) | 100.00% (10/10) | 0.00% (0/10) |
| AGE | 0.00% (0/56) | 100.00% (56/56) | 0.00% (0/56) |
| GENDER | 0.00% (0/13) | 100.00% (13/13) | 0.00% (0/13) |
| ETHNICITY | 0.00% (0/5) | 100.00% (5/5) | 0.00% (0/5) |
| ADDRESS | 0.00% (0/1) | 100.00% (1/1) | 0.00% (0/1) |
| IP_ADDRESS | 0.00% (0/3) | 100.00% (3/3) | 0.00% (0/3) |
| RACE | 0.00% (0/2) | 100.00% (2/2) | 0.00% (0/2) |
| **Overall** | 0.25% (3/1190) | 96.05% (1143/1190) | 3.70% (44/1190) |

Table 45: Weighted Results per Type and Overall (Prediction for **ShareGPT90K**), Model: **gpt-4.1**

| Type | Weighted Redact | Weighted Abstract | Weighted Retain |
|---|---|---|---|
| NAME | 28.49% (49/172) | 62.21% (107/172) | 9.30% (16/172) |
| AFFILIATION | 11.70% (20/171) | 64.91% (111/171) | 23.39% (40/171) |
| TIME | 8.71% (23/264) | 55.68% (147/264) | 35.61% (94/264) |
| URL | 30.00% (6/20) | 30.00% (6/20) | 40.00% (8/20) |
| EMAIL | 75.00% (3/4) | 0.00% (0/4) | 25.00% (1/4) |
| GEOLOCATION | 6.73% (22/327) | 41.59% (136/327) | 51.68% (169/327) |
| RELIGION | 50.00% (1/2) | 0.00% (0/2) | 50.00% (1/2) |
| FINANCIAL_INFORMATION | 0.00% (0/11) | 54.55% (6/11) | 45.45% (5/11) |
| MARITAL_STATUS | 18.18% (2/11) | 81.82% (9/11) | 0.00% (0/11) |
| OCCUPATION | 1.67% (1/60) | 75.00% (45/60) | 23.33% (14/60) |
| VEHICLE | 0.00% (0/1) | 0.00% (0/1) | 100.00% (1/1) |
| INCOME | 0.00% (0/8) | 62.50% (5/8) | 37.50% (3/8) |
| HEALTH_INFORMATION | 28.57% (14/49) | 63.27% (31/49) | 8.16% (4/49) |
| EDUCATIONAL_RECORD | 0.00% (0/10) | 100.00% (10/10) | 0.00% (0/10) |
| AGE | 23.21% (13/56) | 67.86% (38/56) | 8.93% (5/56) |
| GENDER | 46.15% (6/13) | 46.15% (6/13) | 7.69% (1/13) |
| ETHNICITY | 40.00% (2/5) | 40.00% (2/5) | 20.00% (1/5) |
| ADDRESS | 100.00% (1/1) | 0.00% (0/1) | 0.00% (0/1) |
| IP_ADDRESS | 0.00% (0/3) | 100.00% (3/3) | 0.00% (0/3) |
| RACE | 50.00% (1/2) | 50.00% (1/2) | 0.00% (0/2) |
| **Overall** | 13.78% (164/1190) | 55.71% (663/1190) | 30.50% (363/1190) |

Table 46: Weighted Results per Type and Overall (Prediction for **ShareGPT90K**), Model: **gpt-5**

| Type | Weighted Redact | Weighted Abstract | Weighted Retain |
|---|---|---|---|
| NAME | 2.91% (5/172) | 76.74% (132/172) | 20.35% (35/172) |
| AFFILIATION | 3.51% (6/171) | 34.50% (59/171) | 61.99% (106/171) |
| TIME | 1.14% (3/264) | 20.45% (54/264) | 78.41% (207/264) |
| URL | 40.00% (8/20) | 25.00% (5/20) | 35.00% (7/20) |
| EMAIL | 0.00% (0/4) | 75.00% (3/4) | 25.00% (1/4) |
| GEOLOCATION | 0.92% (3/327) | 22.63% (74/327) | 76.45% (250/327) |
| RELIGION | 0.00% (0/2) | 100.00% (2/2) | 0.00% (0/2) |
| FINANCIAL_INFORMATION | 0.00% (0/11) | 54.55% (6/11) | 45.45% (5/11) |
| MARITAL_STATUS | 0.00% (0/11) | 81.82% (9/11) | 18.18% (2/11) |
| OCCUPATION | 0.00% (0/60) | 11.67% (7/60) | 88.33% (53/60) |
| VEHICLE | 0.00% (0/1) | 100.00% (1/1) | 0.00% (0/1) |
| INCOME | 0.00% (0/8) | 75.00% (6/8) | 25.00% (2/8) |
| HEALTH_INFORMATION | 0.00% (0/49) | 20.41% (10/49) | 79.59% (39/49) |
| EDUCATIONAL_RECORD | 0.00% (0/10) | 30.00% (3/10) | 70.00% (7/10) |
| AGE | 5.36% (3/56) | 51.79% (29/56) | 42.86% (24/56) |
| GENDER | 0.00% (0/13) | 38.46% (5/13) | 61.54% (8/13) |
| ETHNICITY | 20.00% (1/5) | 60.00% (3/5) | 20.00% (1/5) |
| ADDRESS | 0.00% (0/1) | 100.00% (1/1) | 0.00% (0/1) |
| IP_ADDRESS | 0.00% (0/3) | 0.00% (0/3) | 100.00% (3/3) |
| RACE | 0.00% (0/2) | 50.00% (1/2) | 50.00% (1/2) |
| **Overall** | 2.44% (29/1190) | 34.45% (410/1190) | 63.11% (751/1190) |

Table 47: Weighted Results per Type and Overall (Prediction for **ShareGPT90K**), Model: **claude-3-7-sonnet-20250219**

| Type | Weighted Redact | Weighted Abstract | Weighted Retain |
|---|---|---|---|
| NAME | 2.33% (4/172) | 85.47% (147/172) | 12.21% (21/172) |
| AFFILIATION | 0.00% (0/171) | 57.89% (99/171) | 42.11% (72/171) |
| TIME | 0.38% (1/264) | 48.48% (128/264) | 51.14% (135/264) |
| URL | 0.00% (0/20) | 70.00% (14/20) | 30.00% (6/20) |
| EMAIL | 0.00% (0/4) | 75.00% (3/4) | 25.00% (1/4) |
| GEOLOCATION | 0.00% (0/327) | 35.78% (117/327) | 64.22% (210/327) |
| RELIGION | 0.00% (0/2) | 0.00% (0/2) | 100.00% (2/2) |
| FINANCIAL_INFORMATION | 0.00% (0/11) | 45.45% (5/11) | 54.55% (6/11) |
| MARITAL_STATUS | 0.00% (0/11) | 100.00% (11/11) | 0.00% (0/11) |
| OCCUPATION | 0.00% (0/60) | 45.00% (27/60) | 55.00% (33/60) |
| VEHICLE | 0.00% (0/1) | 100.00% (1/1) | 0.00% (0/1) |
| INCOME | 0.00% (0/8) | 100.00% (8/8) | 0.00% (0/8) |
| HEALTH_INFORMATION | 0.00% (0/49) | 20.41% (10/49) | 79.59% (39/49) |
| EDUCATIONAL_RECORD | 0.00% (0/10) | 100.00% (10/10) | 0.00% (0/10) |
| AGE | 10.71% (6/56) | 78.57% (44/56) | 10.71% (6/56) |
| GENDER | 7.69% (1/13) | 92.31% (12/13) | 0.00% (0/13) |
| ETHNICITY | 20.00% (1/5) | 80.00% (4/5) | 0.00% (0/5) |
| ADDRESS | 0.00% (0/1) | 100.00% (1/1) | 0.00% (0/1) |
| IP_ADDRESS | 0.00% (0/3) | 100.00% (3/3) | 0.00% (0/3) |
| RACE | 50.00% (1/2) | 50.00% (1/2) | 0.00% (0/2) |
| **Overall** | 1.18% (14/1190) | 54.20% (645/1190) | 44.62% (531/1190) |

Table 48: Weighted Results per Type and Overall (Prediction for **ShareGPT90K**), Model: **claude-sonnet-4-20250514**

| Type | Weighted Redact | Weighted Abstract | Weighted Retain |
|---|---|---|---|
| NAME | 34.88% (60/172) | 65.12% (112/172) | 0.00% (0/172) |
| AFFILIATION | 26.32% (45/171) | 72.51% (124/171) | 1.17% (2/171) |
| TIME | 14.77% (39/264) | 81.82% (216/264) | 3.41% (9/264) |
| URL | 25.00% (5/20) | 75.00% (15/20) | 0.00% (0/20) |
| EMAIL | 75.00% (3/4) | 25.00% (1/4) | 0.00% (0/4) |
| GEOLOCATION | 15.60% (51/327) | 81.65% (267/327) | 2.75% (9/327) |
| RELIGION | 0.00% (0/2) | 100.00% (2/2) | 0.00% (0/2) |
| FINANCIAL_INFORMATION | 9.09% (1/11) | 90.91% (10/11) | 0.00% (0/11) |
| MARITAL_STATUS | 0.00% (0/11) | 100.00% (11/11) | 0.00% (0/11) |
| OCCUPATION | 28.33% (17/60) | 71.67% (43/60) | 0.00% (0/60) |
| VEHICLE | 0.00% (0/1) | 100.00% (1/1) | 0.00% (0/1) |
| INCOME | 25.00% (2/8) | 75.00% (6/8) | 0.00% (0/8) |
| HEALTH_INFORMATION | 18.37% (9/49) | 81.63% (40/49) | 0.00% (0/49) |
| EDUCATIONAL_RECORD | 20.00% (2/10) | 80.00% (8/10) | 0.00% (0/10) |
| AGE | 23.21% (13/56) | 67.86% (38/56) | 8.93% (5/56) |
| GENDER | 46.15% (6/13) | 53.85% (7/13) | 0.00% (0/13) |
| ETHNICITY | 0.00% (0/5) | 100.00% (5/5) | 0.00% (0/5) |
| ADDRESS | 100.00% (1/1) | 0.00% (0/1) | 0.00% (0/1) |
| IP_ADDRESS | 100.00% (3/3) | 0.00% (0/3) | 0.00% (0/3) |
| RACE | 100.00% (2/2) | 0.00% (0/2) | 0.00% (0/2) |
| **Overall** | 21.76% (259/1190) | 76.13% (906/1190) | 2.10% (25/1190) |

Table 49: Weighted Results per Type and Overall (Prediction for **ShareGPT90K**), Model: **lgai/exaone-deep-32b**

| Type | Weighted Redact | Weighted Abstract | Weighted Retain |
|---|---|---|---|
| NAME | 0.58% (1/172) | 98.84% (170/172) | 0.58% (1/172) |
| AFFILIATION | 0.00% (0/171) | 98.25% (168/171) | 1.75% (3/171) |
| TIME | 0.00% (0/264) | 98.11% (259/264) | 1.89% (5/264) |
| URL | 15.00% (3/20) | 75.00% (15/20) | 10.00% (2/20) |
| EMAIL | 0.00% (0/4) | 100.00% (4/4) | 0.00% (0/4) |
| GEOLOCATION | 0.31% (1/327) | 97.55% (319/327) | 2.14% (7/327) |
| RELIGION | 0.00% (0/2) | 50.00% (1/2) | 50.00% (1/2) |
| FINANCIAL_INFORMATION | 9.09% (1/11) | 81.82% (9/11) | 9.09% (1/11) |
| MARITAL_STATUS | 0.00% (0/11) | 100.00% (11/11) | 0.00% (0/11) |
| OCCUPATION | 0.00% (0/60) | 98.33% (59/60) | 1.67% (1/60) |
| VEHICLE | 0.00% (0/1) | 100.00% (1/1) | 0.00% (0/1) |
| INCOME | 0.00% (0/8) | 100.00% (8/8) | 0.00% (0/8) |
| HEALTH_INFORMATION | 0.00% (0/49) | 87.76% (43/49) | 12.24% (6/49) |
| EDUCATIONAL_RECORD | 0.00% (0/10) | 80.00% (8/10) | 20.00% (2/10) |
| AGE | 0.00% (0/56) | 100.00% (56/56) | 0.00% (0/56) |
| GENDER | 0.00% (0/13) | 100.00% (13/13) | 0.00% (0/13) |
| ETHNICITY | 0.00% (0/5) | 100.00% (5/5) | 0.00% (0/5) |
| ADDRESS | 0.00% (0/1) | 100.00% (1/1) | 0.00% (0/1) |
| IP_ADDRESS | 0.00% (0/3) | 100.00% (3/3) | 0.00% (0/3) |
| RACE | 0.00% (0/2) | 100.00% (2/2) | 0.00% (0/2) |
| **Overall** | 0.50% (6/1190) | 97.06% (1155/1190) | 2.44% (29/1190) |

Table 50: Weighted Results per Type and Overall (Prediction for **ShareGPT90K**), Model: **mistralai/mistral-small-3.1-24b-instruct**

| Type | Weighted Redact | Weighted Abstract | Weighted Retain |
|------|-----------------|-------------------|-----------------|
| NAME | 0.58% (1/172) | 99.42% (171/172) | 0.00% (0/172) |
| AFFILIATION | 1.17% (2/171) | 98.83% (169/171) | 0.00% (0/171) |
| TIME | 1.14% (3/264) | 98.86% (261/264) | 0.00% (0/264) |
| URL | 10.00% (2/20) | 90.00% (18/20) | 0.00% (0/20) |
| EMAIL | 75.00% (3/4) | 25.00% (1/4) | 0.00% (0/4) |
| GEOLOCATION | 0.61% (2/327) | 99.39% (325/327) | 0.00% (0/327) |
| RELIGION | 0.00% (0/2) | 100.00% (2/2) | 0.00% (0/2) |
| FINANCIAL_INFORMATION | 18.18% (2/11) | 81.82% (9/11) | 0.00% (0/11) |
| MARITAL_STATUS | 9.09% (1/11) | 90.91% (10/11) | 0.00% (0/11) |
| OCCUPATION | 0.00% (0/60) | 100.00% (60/60) | 0.00% (0/60) |
| VEHICLE | 0.00% (0/1) | 100.00% (1/1) | 0.00% (0/1) |
| INCOME | 0.00% (0/8) | 100.00% (8/8) | 0.00% (0/8) |
| HEALTH_INFORMATION | 0.00% (0/49) | 100.00% (49/49) | 0.00% (0/49) |
| EDUCATIONAL_RECORD | 0.00% (0/10) | 100.00% (10/10) | 0.00% (0/10) |
| AGE | 0.00% (0/56) | 100.00% (56/56) | 0.00% (0/56) |
| GENDER | 0.00% (0/13) | 100.00% (13/13) | 0.00% (0/13) |
| ETHNICITY | 0.00% (0/5) | 100.00% (5/5) | 0.00% (0/5) |
| ADDRESS | 0.00% (0/1) | 100.00% (1/1) | 0.00% (0/1) |
| IP_ADDRESS | 0.00% (0/3) | 100.00% (3/3) | 0.00% (0/3) |
| RACE | 0.00% (0/2) | 100.00% (2/2) | 0.00% (0/2) |
| **Overall** | 1.34% (16/1190) | 98.66% (1174/1190) | 0.00% (0/1190) |

Table 51: Weighted Results per Type and Overall (Prediction for **ShareGPT90K**), Model: **qwen/qwen2.5-7b-instruct**

## 1.4 WILDCHAT

| Type | Weighted Redact | Weighted Abstract | Weighted Retain |
|------|-----------------|-------------------|-----------------|
| NAME | 88.82% (135/152) | 3.95% (6/152) | 7.24% (11/152) |
| AFFILIATION | 90.91% (130/143) | 6.99% (10/143) | 2.10% (3/143) |
| GEOLOCATION | 84.39% (200/237) | 6.75% (16/237) | 8.86% (21/237) |
| USERNAME | 100.00% (2/2) | 0.00% (0/2) | 0.00% (0/2) |
| TIME | 85.23% (127/149) | 8.05% (12/149) | 6.71% (10/149) |
| AGE | 63.64% (14/22) | 18.18% (4/22) | 18.18% (4/22) |
| OCCUPATION | 81.08% (30/37) | 10.81% (4/37) | 8.11% (3/37) |
| QUANTITY | 100.00% (6/6) | 0.00% (0/6) | 0.00% (0/6) |
| ETHNICITY | 77.78% (7/9) | 11.11% (1/9) | 11.11% (1/9) |
| GENDER | 85.71% (6/7) | 14.29% (1/7) | 0.00% (0/7) |
| EMAIL | 100.00% (2/2) | 0.00% (0/2) | 0.00% (0/2) |
| URL | 100.00% (8/8) | 0.00% (0/8) | 0.00% (0/8) |
| HEALTH_INFORMATION | 50.00% (3/6) | 33.33% (2/6) | 16.67% (1/6) |
| RACE | 100.00% (1/1) | 0.00% (0/1) | 0.00% (0/1) |
| INCOME | 85.71% (12/14) | 14.29% (2/14) | 0.00% (0/14) |
| PRODUCT | 100.00% (1/1) | 0.00% (0/1) | 0.00% (0/1) |
| FINANCIAL_INFORMATION | 83.33% (5/6) | 0.00% (0/6) | 16.67% (1/6) |
| PHONE_NUMBER | 100.00% (2/2) | 0.00% (0/2) | 0.00% (0/2) |
| EDUCATIONAL_RECORD | 100.00% (11/11) | 0.00% (0/11) | 0.00% (0/11) |
| ID_NUMBER | 100.00% (2/2) | 0.00% (0/2) | 0.00% (0/2) |
| KEYS | 100.00% (1/1) | 0.00% (0/1) | 0.00% (0/1) |
| GPA | 100.00% (1/1) | 0.00% (0/1) | 0.00% (0/1) |
| **Overall** | 86.20% (706/819) | 7.08% (58/819) | 6.72% (55/819) |

Table 52: Weighted Results per Type and Overall (Oracle for **WildChat**), Model: **gpt-4.1-nano**

| Type | Weighted Redact | Weighted Abstract | Weighted Retain |
|---|---|---|---|
| NAME | 88.82% (135/152) | 5.92% (9/152) | 5.26% (8/152) |
| AFFILIATION | 91.61% (131/143) | 4.20% (6/143) | 4.20% (6/143) |
| GEOLOCATION | 81.01% (192/237) | 9.28% (22/237) | 9.70% (23/237) |
| USERNAME | 100.00% (2/2) | 0.00% (0/2) | 0.00% (0/2) |
| TIME | 85.23% (127/149) | 6.71% (10/149) | 8.05% (12/149) |
| AGE | 72.73% (16/22) | 9.09% (2/22) | 18.18% (4/22) |
| OCCUPATION | 89.19% (33/37) | 5.41% (2/37) | 5.41% (2/37) |
| QUANTITY | 100.00% (6/6) | 0.00% (0/6) | 0.00% (0/6) |
| ETHNICITY | 77.78% (7/9) | 0.00% (0/9) | 22.22% (2/9) |
| GENDER | 71.43% (5/7) | 14.29% (1/7) | 14.29% (1/7) |
| EMAIL | 100.00% (2/2) | 0.00% (0/2) | 0.00% (0/2) |
| URL | 100.00% (8/8) | 0.00% (0/8) | 0.00% (0/8) |
| HEALTH_INFORMATION | 83.33% (5/6) | 16.67% (1/6) | 0.00% (0/6) |
| RACE | 100.00% (1/1) | 0.00% (0/1) | 0.00% (0/1) |
| INCOME | 100.00% (14/14) | 0.00% (0/14) | 0.00% (0/14) |
| PRODUCT | 100.00% (1/1) | 0.00% (0/1) | 0.00% (0/1) |
| FINANCIAL_INFORMATION | 83.33% (5/6) | 16.67% (1/6) | 0.00% (0/6) |
| PHONE_NUMBER | 100.00% (2/2) | 0.00% (0/2) | 0.00% (0/2) |
| EDUCATIONAL_RECORD | 81.82% (9/11) | 18.18% (2/11) | 0.00% (0/11) |
| ID_NUMBER | 100.00% (2/2) | 0.00% (0/2) | 0.00% (0/2) |
| KEYS | 100.00% (1/1) | 0.00% (0/1) | 0.00% (0/1) |
| GPA | 100.00% (1/1) | 0.00% (0/1) | 0.00% (0/1) |
| **Overall** | 86.08% (705/819) | 6.84% (56/819) | 7.08% (58/819) |

Table 53: Weighted Results per Type and Overall (Oracle for **WildChat**), Model: **gpt-4.1**

| Type | Weighted Redact | Weighted Abstract | Weighted Retain |
|---|---|---|---|
| NAME | 89.47% (136/152) | 7.89% (12/152) | 2.63% (4/152) |
| AFFILIATION | 93.71% (134/143) | 4.90% (7/143) | 1.40% (2/143) |
| GEOLOCATION | 86.50% (205/237) | 9.70% (23/237) | 3.80% (9/237) |
| USERNAME | 100.00% (2/2) | 0.00% (0/2) | 0.00% (0/2) |
| TIME | 91.28% (136/149) | 4.03% (6/149) | 4.70% (7/149) |
| AGE | 77.27% (17/22) | 13.64% (3/22) | 9.09% (2/22) |
| OCCUPATION | 86.49% (32/37) | 8.11% (3/37) | 5.41% (2/37) |
| QUANTITY | 100.00% (6/6) | 0.00% (0/6) | 0.00% (0/6) |
| ETHNICITY | 77.78% (7/9) | 11.11% (1/9) | 11.11% (1/9) |
| GENDER | 85.71% (6/7) | 0.00% (0/7) | 14.29% (1/7) |
| EMAIL | 100.00% (2/2) | 0.00% (0/2) | 0.00% (0/2) |
| URL | 100.00% (8/8) | 0.00% (0/8) | 0.00% (0/8) |
| HEALTH_INFORMATION | 100.00% (6/6) | 0.00% (0/6) | 0.00% (0/6) |
| RACE | 100.00% (1/1) | 0.00% (0/1) | 0.00% (0/1) |
| INCOME | 100.00% (14/14) | 0.00% (0/14) | 0.00% (0/14) |
| PRODUCT | 100.00% (1/1) | 0.00% (0/1) | 0.00% (0/1) |
| FINANCIAL_INFORMATION | 83.33% (5/6) | 0.00% (0/6) | 16.67% (1/6) |
| PHONE_NUMBER | 100.00% (2/2) | 0.00% (0/2) | 0.00% (0/2) |
| EDUCATIONAL_RECORD | 100.00% (11/11) | 0.00% (0/11) | 0.00% (0/11) |
| ID_NUMBER | 100.00% (2/2) | 0.00% (0/2) | 0.00% (0/2) |
| KEYS | 100.00% (1/1) | 0.00% (0/1) | 0.00% (0/1) |
| GPA | 100.00% (1/1) | 0.00% (0/1) | 0.00% (0/1) |
| **Overall** | 89.74% (735/819) | 6.72% (55/819) | 3.54% (29/819) |

Table 54: Weighted Results per Type and Overall (Oracle for **WildChat**), Model: **gpt-5**

| Type | Weighted Redact | Weighted Abstract | Weighted Retain |
|---|---|---|---|
| NAME | 77.63% (118/152) | 9.87% (15/152) | 12.50% (19/152) |
| AFFILIATION | 83.92% (120/143) | 7.69% (11/143) | 8.39% (12/143) |
| GEOLOCATION | 78.06% (185/237) | 9.28% (22/237) | 12.66% (30/237) |
| USERNAME | 100.00% (2/2) | 0.00% (0/2) | 0.00% (0/2) |
| TIME | 83.89% (125/149) | 10.74% (16/149) | 5.37% (8/149) |
| AGE | 63.64% (14/22) | 13.64% (3/22) | 22.73% (5/22) |
| OCCUPATION | 78.38% (29/37) | 5.41% (2/37) | 16.22% (6/37) |
| QUANTITY | 100.00% (6/6) | 0.00% (0/6) | 0.00% (0/6) |
| ETHNICITY | 33.33% (3/9) | 0.00% (0/9) | 66.67% (6/9) |
| GENDER | 85.71% (6/7) | 0.00% (0/7) | 14.29% (1/7) |
| EMAIL | 100.00% (2/2) | 0.00% (0/2) | 0.00% (0/2) |
| URL | 75.00% (6/8) | 12.50% (1/8) | 12.50% (1/8) |
| HEALTH_INFORMATION | 100.00% (6/6) | 0.00% (0/6) | 0.00% (0/6) |
| RACE | 0.00% (0/1) | 100.00% (1/1) | 0.00% (0/1) |
| INCOME | 100.00% (14/14) | 0.00% (0/14) | 0.00% (0/14) |
| PRODUCT | 100.00% (1/1) | 0.00% (0/1) | 0.00% (0/1) |
| FINANCIAL_INFORMATION | 66.67% (4/6) | 16.67% (1/6) | 16.67% (1/6) |
| PHONE_NUMBER | 100.00% (2/2) | 0.00% (0/2) | 0.00% (0/2) |
| EDUCATIONAL_RECORD | 100.00% (11/11) | 0.00% (0/11) | 0.00% (0/11) |
| ID_NUMBER | 100.00% (2/2) | 0.00% (0/2) | 0.00% (0/2) |
| KEYS | 100.00% (1/1) | 0.00% (0/1) | 0.00% (0/1) |
| GPA | 100.00% (1/1) | 0.00% (0/1) | 0.00% (0/1) |
| **Overall** | 80.34% (658/819) | 8.79% (72/819) | 10.87% (89/819) |

Table 55: Weighted Results per Type and Overall (Oracle for **WildChat**), Model: **claude-3-7-sonnet-20250219**

| Type | Weighted Redact | Weighted Abstract | Weighted Retain |
|---|---|---|---|
| NAME | 81.58% (124/152) | 7.24% (11/152) | 11.18% (17/152) |
| AFFILIATION | 85.31% (122/143) | 6.29% (9/143) | 8.39% (12/143) |
| GEOLOCATION | 78.90% (187/237) | 9.70% (23/237) | 11.39% (27/237) |
| USERNAME | 100.00% (2/2) | 0.00% (0/2) | 0.00% (0/2) |
| TIME | 81.21% (121/149) | 12.08% (18/149) | 6.71% (10/149) |
| AGE | 63.64% (14/22) | 13.64% (3/22) | 22.73% (5/22) |
| OCCUPATION | 83.78% (31/37) | 5.41% (2/37) | 10.81% (4/37) |
| QUANTITY | 100.00% (6/6) | 0.00% (0/6) | 0.00% (0/6) |
| ETHNICITY | 44.44% (4/9) | 0.00% (0/9) | 55.56% (5/9) |
| GENDER | 71.43% (5/7) | 28.57% (2/7) | 0.00% (0/7) |
| EMAIL | 100.00% (2/2) | 0.00% (0/2) | 0.00% (0/2) |
| URL | 100.00% (8/8) | 0.00% (0/8) | 0.00% (0/8) |
| HEALTH_INFORMATION | 83.33% (5/6) | 16.67% (1/6) | 0.00% (0/6) |
| RACE | 100.00% (1/1) | 0.00% (0/1) | 0.00% (0/1) |
| INCOME | 92.86% (13/14) | 0.00% (0/14) | 7.14% (1/14) |
| PRODUCT | 100.00% (1/1) | 0.00% (0/1) | 0.00% (0/1) |
| FINANCIAL_INFORMATION | 83.33% (5/6) | 16.67% (1/6) | 0.00% (0/6) |
| PHONE_NUMBER | 100.00% (2/2) | 0.00% (0/2) | 0.00% (0/2) |
| EDUCATIONAL_RECORD | 81.82% (9/11) | 18.18% (2/11) | 0.00% (0/11) |
| ID_NUMBER | 100.00% (2/2) | 0.00% (0/2) | 0.00% (0/2) |
| KEYS | 100.00% (1/1) | 0.00% (0/1) | 0.00% (0/1) |
| GPA | 100.00% (1/1) | 0.00% (0/1) | 0.00% (0/1) |
| **Overall** | 81.32% (666/819) | 8.79% (72/819) | 9.89% (81/819) |

Table 56: Weighted Results per Type and Overall (Oracle for **WildChat**), Model: **claude-sonnet-4-20250514**

| Type | Weighted Redact | Weighted Abstract | Weighted Retain |
|---|---|---|---|
| NAME | 67.76% (103/152) | 18.42% (28/152) | 13.82% (21/152) |
| AFFILIATION | 73.43% (105/143) | 12.59% (18/143) | 13.99% (20/143) |
| GEOLOCATION | 67.09% (159/237) | 12.24% (29/237) | 20.68% (49/237) |
| USERNAME | 50.00% (1/2) | 50.00% (1/2) | 0.00% (0/2) |
| TIME | 65.10% (97/149) | 18.12% (27/149) | 16.78% (25/149) |
| AGE | 63.64% (14/22) | 13.64% (3/22) | 22.73% (5/22) |
| OCCUPATION | 72.97% (27/37) | 10.81% (4/37) | 16.22% (6/37) |
| QUANTITY | 66.67% (4/6) | 16.67% (1/6) | 16.67% (1/6) |
| ETHNICITY | 55.56% (5/9) | 22.22% (2/9) | 22.22% (2/9) |
| GENDER | 71.43% (5/7) | 14.29% (1/7) | 14.29% (1/7) |
| EMAIL | 100.00% (2/2) | 0.00% (0/2) | 0.00% (0/2) |
| URL | 87.50% (7/8) | 12.50% (1/8) | 0.00% (0/8) |
| HEALTH_INFORMATION | 66.67% (4/6) | 16.67% (1/6) | 16.67% (1/6) |
| RACE | 100.00% (1/1) | 0.00% (0/1) | 0.00% (0/1) |
| INCOME | 92.86% (13/14) | 0.00% (0/14) | 7.14% (1/14) |
| PRODUCT | 100.00% (1/1) | 0.00% (0/1) | 0.00% (0/1) |
| FINANCIAL_INFORMATION | 66.67% (4/6) | 16.67% (1/6) | 16.67% (1/6) |
| PHONE_NUMBER | 100.00% (2/2) | 0.00% (0/2) | 0.00% (0/2) |
| EDUCATIONAL_RECORD | 81.82% (9/11) | 9.09% (1/11) | 9.09% (1/11) |
| ID_NUMBER | 100.00% (2/2) | 0.00% (0/2) | 0.00% (0/2) |
| KEYS | 100.00% (1/1) | 0.00% (0/1) | 0.00% (0/1) |
| GPA | 100.00% (1/1) | 0.00% (0/1) | 0.00% (0/1) |
| **Overall** | 69.23% (567/819) | 14.41% (118/819) | 16.36% (134/819) |

Table 57: Weighted Results per Type and Overall (Oracle for **WildChat**), Model: **lgai/exaone-deep-32b**

| Type | Weighted Redact | Weighted Abstract | Weighted Retain |
|---|---|---|---|
| NAME | 83.55% (127/152) | 8.55% (13/152) | 7.89% (12/152) |
| AFFILIATION | 90.91% (130/143) | 5.59% (8/143) | 3.50% (5/143) |
| GEOLOCATION | 83.54% (198/237) | 7.17% (17/237) | 9.28% (22/237) |
| USERNAME | 100.00% (2/2) | 0.00% (0/2) | 0.00% (0/2) |
| TIME | 83.22% (124/149) | 10.07% (15/149) | 6.71% (10/149) |
| AGE | 77.27% (17/22) | 9.09% (2/22) | 13.64% (3/22) |
| OCCUPATION | 86.49% (32/37) | 2.70% (1/37) | 10.81% (4/37) |
| QUANTITY | 100.00% (6/6) | 0.00% (0/6) | 0.00% (0/6) |
| ETHNICITY | 66.67% (6/9) | 11.11% (1/9) | 22.22% (2/9) |
| GENDER | 71.43% (5/7) | 14.29% (1/7) | 14.29% (1/7) |
| EMAIL | 100.00% (2/2) | 0.00% (0/2) | 0.00% (0/2) |
| URL | 100.00% (8/8) | 0.00% (0/8) | 0.00% (0/8) |
| HEALTH_INFORMATION | 66.67% (4/6) | 33.33% (2/6) | 0.00% (0/6) |
| RACE | 100.00% (1/1) | 0.00% (0/1) | 0.00% (0/1) |
| INCOME | 100.00% (14/14) | 0.00% (0/14) | 0.00% (0/14) |
| PRODUCT | 100.00% (1/1) | 0.00% (0/1) | 0.00% (0/1) |
| FINANCIAL_INFORMATION | 66.67% (4/6) | 16.67% (1/6) | 16.67% (1/6) |
| PHONE_NUMBER | 100.00% (2/2) | 0.00% (0/2) | 0.00% (0/2) |
| EDUCATIONAL_RECORD | 100.00% (11/11) | 0.00% (0/11) | 0.00% (0/11) |
| ID_NUMBER | 100.00% (2/2) | 0.00% (0/2) | 0.00% (0/2) |
| KEYS | 100.00% (1/1) | 0.00% (0/1) | 0.00% (0/1) |
| GPA | 100.00% (1/1) | 0.00% (0/1) | 0.00% (0/1) |
| **Overall** | 85.23% (698/819) | 7.45% (61/819) | 7.33% (60/819) |

Table 58: Weighted Results per Type and Overall (Oracle for **WildChat**), Model: **mistralai/mistral-small-3.1-24b-instruct**

| Type | Weighted Redact | Weighted Abstract | Weighted Retain |
|---|---|---|---|
| NAME | 83.55% (127/152) | 5.26% (8/152) | 11.18% (17/152) |
| AFFILIATION | 88.11% (126/143) | 4.90% (7/143) | 6.99% (10/143) |
| GEOLOCATION | 78.06% (185/237) | 9.70% (23/237) | 12.24% (29/237) |
| USERNAME | 50.00% (1/2) | 0.00% (0/2) | 50.00% (1/2) |
| TIME | 75.17% (112/149) | 10.07% (15/149) | 14.77% (22/149) |
| AGE | 81.82% (18/22) | 0.00% (0/22) | 18.18% (4/22) |
| OCCUPATION | 59.46% (22/37) | 13.51% (5/37) | 27.03% (10/37) |
| QUANTITY | 100.00% (6/6) | 0.00% (0/6) | 0.00% (0/6) |
| ETHNICITY | 66.67% (6/9) | 0.00% (0/9) | 33.33% (3/9) |
| GENDER | 71.43% (5/7) | 14.29% (1/7) | 14.29% (1/7) |
| EMAIL | 100.00% (2/2) | 0.00% (0/2) | 0.00% (0/2) |
| URL | 100.00% (8/8) | 0.00% (0/8) | 0.00% (0/8) |
| HEALTH_INFORMATION | 83.33% (5/6) | 16.67% (1/6) | 0.00% (0/6) |
| RACE | 100.00% (1/1) | 0.00% (0/1) | 0.00% (0/1) |
| INCOME | 78.57% (11/14) | 21.43% (3/14) | 0.00% (0/14) |
| PRODUCT | 100.00% (1/1) | 0.00% (0/1) | 0.00% (0/1) |
| FINANCIAL_INFORMATION | 83.33% (5/6) | 0.00% (0/6) | 16.67% (1/6) |
| PHONE_NUMBER | 100.00% (2/2) | 0.00% (0/2) | 0.00% (0/2) |
| EDUCATIONAL_RECORD | 81.82% (9/11) | 18.18% (2/11) | 0.00% (0/11) |
| ID_NUMBER | 100.00% (2/2) | 0.00% (0/2) | 0.00% (0/2) |
| KEYS | 100.00% (1/1) | 0.00% (0/1) | 0.00% (0/1) |
| GPA | 100.00% (1/1) | 0.00% (0/1) | 0.00% (0/1) |
| **Overall** | 80.10% (656/819) | 7.94% (65/819) | 11.97% (98/819) |

Table 59: Weighted Results per Type and Overall (Oracle for **WildChat**), Model: **qwen/qwen2.5-7b-instruct**

| Type | Weighted Redact | Weighted Abstract | Weighted Retain |
|---|---|---|---|
| NAME | 36.18% (55/152) | 10.53% (16/152) | 53.29% (81/152) |
| AFFILIATION | 36.36% (52/143) | 16.08% (23/143) | 47.55% (68/143) |
| GEOLOCATION | 24.05% (57/237) | 21.10% (50/237) | 54.85% (130/237) |
| USERNAME | 0.00% (0/2) | 0.00% (0/2) | 100.00% (2/2) |
| TIME | 25.50% (38/149) | 13.42% (20/149) | 61.07% (91/149) |
| AGE | 9.09% (2/22) | 9.09% (2/22) | 81.82% (18/22) |
| OCCUPATION | 29.73% (11/37) | 8.11% (3/37) | 62.16% (23/37) |
| QUANTITY | 50.00% (3/6) | 50.00% (3/6) | 0.00% (0/6) |
| ETHNICITY | 11.11% (1/9) | 0.00% (0/9) | 88.89% (8/9) |
| GENDER | 42.86% (3/7) | 28.57% (2/7) | 28.57% (2/7) |
| EMAIL | 100.00% (2/2) | 0.00% (0/2) | 0.00% (0/2) |
| URL | 50.00% (4/8) | 25.00% (2/8) | 25.00% (2/8) |
| HEALTH_INFORMATION | 16.67% (1/6) | 0.00% (0/6) | 83.33% (5/6) |
| RACE | 0.00% (0/1) | 0.00% (0/1) | 100.00% (1/1) |
| INCOME | 7.14% (1/14) | 0.00% (0/14) | 92.86% (13/14) |
| PRODUCT | 0.00% (0/1) | 0.00% (0/1) | 100.00% (1/1) |
| FINANCIAL_INFORMATION | 33.33% (2/6) | 50.00% (3/6) | 16.67% (1/6) |
| PHONE_NUMBER | 50.00% (1/2) | 0.00% (0/2) | 50.00% (1/2) |
| EDUCATIONAL_RECORD | 18.18% (2/11) | 0.00% (0/11) | 81.82% (9/11) |
| ID_NUMBER | 0.00% (0/2) | 100.00% (2/2) | 0.00% (0/2) |
| KEYS | 100.00% (1/1) | 0.00% (0/1) | 0.00% (0/1) |
| GPA | 0.00% (0/1) | 0.00% (0/1) | 100.00% (1/1) |
| **Overall** | 28.82% (236/819) | 15.38% (126/819) | 55.80% (457/819) |

Table 60: Weighted Results per Type and Overall (Oracle for **WildChat**), Model: **qwen/qwen2.5-0.5b-instruct**

| Type | Weighted Redact | Weighted Abstract | Weighted Retain |
|---|---|---|---|
| NAME | 64.47% (98/152) | 34.21% (52/152) | 1.32% (2/152) |
| AFFILIATION | 44.76% (64/143) | 55.24% (79/143) | 0.00% (0/143) |
| GEOLOCATION | 59.07% (140/237) | 40.93% (97/237) | 0.00% (0/237) |
| USERNAME | 50.00% (1/2) | 50.00% (1/2) | 0.00% (0/2) |
| TIME | 40.94% (61/149) | 59.06% (88/149) | 0.00% (0/149) |
| AGE | 63.64% (14/22) | 36.36% (8/22) | 0.00% (0/22) |
| OCCUPATION | 45.95% (17/37) | 54.05% (20/37) | 0.00% (0/37) |
| QUANTITY | 50.00% (3/6) | 50.00% (3/6) | 0.00% (0/6) |
| ETHNICITY | 55.56% (5/9) | 44.44% (4/9) | 0.00% (0/9) |
| GENDER | 71.43% (5/7) | 28.57% (2/7) | 0.00% (0/7) |
| EMAIL | 100.00% (2/2) | 0.00% (0/2) | 0.00% (0/2) |
| URL | 87.50% (7/8) | 12.50% (1/8) | 0.00% (0/8) |
| HEALTH_INFORMATION | 83.33% (5/6) | 16.67% (1/6) | 0.00% (0/6) |
| RACE | 100.00% (1/1) | 0.00% (0/1) | 0.00% (0/1) |
| INCOME | 92.86% (13/14) | 7.14% (1/14) | 0.00% (0/14) |
| PRODUCT | 100.00% (1/1) | 0.00% (0/1) | 0.00% (0/1) |
| FINANCIAL_INFORMATION | 66.67% (4/6) | 33.33% (2/6) | 0.00% (0/6) |
| PHONE_NUMBER | 100.00% (2/2) | 0.00% (0/2) | 0.00% (0/2) |
| EDUCATIONAL_RECORD | 0.00% (0/11) | 100.00% (11/11) | 0.00% (0/11) |
| ID_NUMBER | 100.00% (2/2) | 0.00% (0/2) | 0.00% (0/2) |
| KEYS | 100.00% (1/1) | 0.00% (0/1) | 0.00% (0/1) |
| GPA | 0.00% (0/1) | 100.00% (1/1) | 0.00% (0/1) |
| **Overall** | 54.46% (446/819) | 45.30% (371/819) | 0.24% (2/819) |

Table 61: Weighted Results per Type and Overall (Prediction for **WildChat**), Model: **gpt-4.1-nano**

| Type | Weighted Redact | Weighted Abstract | Weighted Retain |
|---|---|---|---|
| NAME | 0.00% (0/152) | 100.00% (152/152) | 0.00% (0/152) |
| AFFILIATION | 0.00% (0/143) | 100.00% (143/143) | 0.00% (0/143) |
| GEOLOCATION | 0.00% (0/237) | 92.83% (220/237) | 7.17% (17/237) |
| USERNAME | 0.00% (0/2) | 100.00% (2/2) | 0.00% (0/2) |
| TIME | 0.00% (0/149) | 98.66% (147/149) | 1.34% (2/149) |
| AGE | 0.00% (0/22) | 95.45% (21/22) | 4.55% (1/22) |
| OCCUPATION | 0.00% (0/37) | 100.00% (37/37) | 0.00% (0/37) |
| QUANTITY | 0.00% (0/6) | 100.00% (6/6) | 0.00% (0/6) |
| ETHNICITY | 0.00% (0/9) | 100.00% (9/9) | 0.00% (0/9) |
| GENDER | 0.00% (0/7) | 100.00% (7/7) | 0.00% (0/7) |
| EMAIL | 0.00% (0/2) | 100.00% (2/2) | 0.00% (0/2) |
| URL | 0.00% (0/8) | 100.00% (8/8) | 0.00% (0/8) |
| HEALTH_INFORMATION | 0.00% (0/6) | 100.00% (6/6) | 0.00% (0/6) |
| RACE | 0.00% (0/1) | 100.00% (1/1) | 0.00% (0/1) |
| INCOME | 0.00% (0/14) | 100.00% (14/14) | 0.00% (0/14) |
| PRODUCT | 0.00% (0/1) | 100.00% (1/1) | 0.00% (0/1) |
| FINANCIAL_INFORMATION | 0.00% (0/6) | 100.00% (6/6) | 0.00% (0/6) |
| PHONE_NUMBER | 0.00% (0/2) | 100.00% (2/2) | 0.00% (0/2) |
| EDUCATIONAL_RECORD | 0.00% (0/11) | 100.00% (11/11) | 0.00% (0/11) |
| ID_NUMBER | 0.00% (0/2) | 100.00% (2/2) | 0.00% (0/2) |
| KEYS | 0.00% (0/1) | 100.00% (1/1) | 0.00% (0/1) |
| GPA | 0.00% (0/1) | 100.00% (1/1) | 0.00% (0/1) |
| **Overall** | 0.00% (0/819) | 97.56% (799/819) | 2.44% (20/819) |

Table 62: Weighted Results per Type and Overall (Prediction for **WildChat**), Model: **gpt-4.1**

| Type | Weighted Redact | Weighted Abstract | Weighted Retain |
|---|---|---|---|
| NAME | 16.45% (25/152) | 57.89% (88/152) | 25.66% (39/152) |
| AFFILIATION | 11.19% (16/143) | 59.44% (85/143) | 29.37% (42/143) |
| GEOLOCATION | 8.44% (20/237) | 57.38% (136/237) | 34.18% (81/237) |
| USERNAME | 50.00% (1/2) | 50.00% (1/2) | 0.00% (0/2) |
| TIME | 19.46% (29/149) | 68.46% (102/149) | 12.08% (18/149) |
| AGE | 13.64% (3/22) | 72.73% (16/22) | 13.64% (3/22) |
| OCCUPATION | 2.70% (1/37) | 75.68% (28/37) | 21.62% (8/37) |
| QUANTITY | 0.00% (0/6) | 16.67% (1/6) | 83.33% (5/6) |
| ETHNICITY | 0.00% (0/9) | 55.56% (5/9) | 44.44% (4/9) |
| GENDER | 28.57% (2/7) | 57.14% (4/7) | 14.29% (1/7) |
| EMAIL | 0.00% (0/2) | 0.00% (0/2) | 100.00% (2/2) |
| URL | 12.50% (1/8) | 50.00% (4/8) | 37.50% (3/8) |
| HEALTH_INFORMATION | 0.00% (0/6) | 100.00% (6/6) | 0.00% (0/6) |
| RACE | 0.00% (0/1) | 0.00% (0/1) | 100.00% (1/1) |
| INCOME | 14.29% (2/14) | 85.71% (12/14) | 0.00% (0/14) |
| PRODUCT | 0.00% (0/1) | 0.00% (0/1) | 100.00% (1/1) |
| FINANCIAL_INFORMATION | 0.00% (0/6) | 16.67% (1/6) | 83.33% (5/6) |
| PHONE_NUMBER | 0.00% (0/2) | 100.00% (2/2) | 0.00% (0/2) |
| EDUCATIONAL_RECORD | 0.00% (0/11) | 81.82% (9/11) | 18.18% (2/11) |
| ID_NUMBER | 100.00% (2/2) | 0.00% (0/2) | 0.00% (0/2) |
| KEYS | 0.00% (0/1) | 100.00% (1/1) | 0.00% (0/1) |
| GPA | 0.00% (0/1) | 100.00% (1/1) | 0.00% (0/1) |
| **Overall** | 12.45% (102/819) | 61.29% (502/819) | 26.25% (215/819) |

Table 63: Weighted Results per Type and Overall (Prediction for **WildChat**), Model: **gpt-5**

| Type | Weighted Redact | Weighted Abstract | Weighted Retain |
|---|---|---|---|
| NAME | 1.97% (3/152) | 61.84% (94/152) | 36.18% (55/152) |
| AFFILIATION | 0.00% (0/143) | 38.46% (55/143) | 61.54% (88/143) |
| GEOLOCATION | 0.42% (1/237) | 27.00% (64/237) | 72.57% (172/237) |
| USERNAME | 50.00% (1/2) | 50.00% (1/2) | 0.00% (0/2) |
| TIME | 3.36% (5/149) | 27.52% (41/149) | 69.13% (103/149) |
| AGE | 4.55% (1/22) | 45.45% (10/22) | 50.00% (11/22) |
| OCCUPATION | 0.00% (0/37) | 32.43% (12/37) | 67.57% (25/37) |
| QUANTITY | 0.00% (0/6) | 0.00% (0/6) | 100.00% (6/6) |
| ETHNICITY | 0.00% (0/9) | 11.11% (1/9) | 88.89% (8/9) |
| GENDER | 0.00% (0/7) | 0.00% (0/7) | 100.00% (7/7) |
| EMAIL | 0.00% (0/2) | 50.00% (1/2) | 50.00% (1/2) |
| URL | 12.50% (1/8) | 37.50% (3/8) | 50.00% (4/8) |
| HEALTH_INFORMATION | 0.00% (0/6) | 100.00% (6/6) | 0.00% (0/6) |
| RACE | 0.00% (0/1) | 0.00% (0/1) | 100.00% (1/1) |
| INCOME | 0.00% (0/14) | 28.57% (4/14) | 71.43% (10/14) |
| PRODUCT | 0.00% (0/1) | 100.00% (1/1) | 0.00% (0/1) |
| FINANCIAL_INFORMATION | 0.00% (0/6) | 66.67% (4/6) | 33.33% (2/6) |
| PHONE_NUMBER | 100.00% (2/2) | 0.00% (0/2) | 0.00% (0/2) |
| EDUCATIONAL_RECORD | 0.00% (0/11) | 81.82% (9/11) | 18.18% (2/11) |
| ID_NUMBER | 0.00% (0/2) | 0.00% (0/2) | 100.00% (2/2) |
| KEYS | 100.00% (1/1) | 0.00% (0/1) | 0.00% (0/1) |
| GPA | 0.00% (0/1) | 100.00% (1/1) | 0.00% (0/1) |
| **Overall** | 1.83% (15/819) | 37.48% (307/819) | 60.68% (497/819) |

Table 64: Weighted Results per Type and Overall (Prediction for **WildChat**), Model: **claude-3-7-sonnet-20250219**

| Type | Weighted Redact | Weighted Abstract | Weighted Retain |
|---|---|---|---|
| NAME | 0.66% (1/152) | 76.97% (117/152) | 22.37% (34/152) |
| AFFILIATION | 0.00% (0/143) | 60.84% (87/143) | 39.16% (56/143) |
| GEOLOCATION | 0.42% (1/237) | 34.60% (82/237) | 64.98% (154/237) |
| USERNAME | 50.00% (1/2) | 50.00% (1/2) | 0.00% (0/2) |
| TIME | 0.00% (0/149) | 57.05% (85/149) | 42.95% (64/149) |
| AGE | 0.00% (0/22) | 81.82% (18/22) | 18.18% (4/22) |
| OCCUPATION | 0.00% (0/37) | 62.16% (23/37) | 37.84% (14/37) |
| QUANTITY | 0.00% (0/6) | 100.00% (6/6) | 0.00% (0/6) |
| ETHNICITY | 0.00% (0/9) | 66.67% (6/9) | 33.33% (3/9) |
| GENDER | 0.00% (0/7) | 42.86% (3/7) | 57.14% (4/7) |
| EMAIL | 0.00% (0/2) | 100.00% (2/2) | 0.00% (0/2) |
| URL | 12.50% (1/8) | 75.00% (6/8) | 12.50% (1/8) |
| HEALTH_INFORMATION | 0.00% (0/6) | 100.00% (6/6) | 0.00% (0/6) |
| RACE | 0.00% (0/1) | 100.00% (1/1) | 0.00% (0/1) |
| INCOME | 0.00% (0/14) | 100.00% (14/14) | 0.00% (0/14) |
| PRODUCT | 0.00% (0/1) | 100.00% (1/1) | 0.00% (0/1) |
| FINANCIAL_INFORMATION | 0.00% (0/6) | 66.67% (4/6) | 33.33% (2/6) |
| PHONE_NUMBER | 0.00% (0/2) | 100.00% (2/2) | 0.00% (0/2) |
| EDUCATIONAL_RECORD | 0.00% (0/11) | 100.00% (11/11) | 0.00% (0/11) |
| ID_NUMBER | 0.00% (0/2) | 100.00% (2/2) | 0.00% (0/2) |
| KEYS | 0.00% (0/1) | 100.00% (1/1) | 0.00% (0/1) |
| GPA | 0.00% (0/1) | 100.00% (1/1) | 0.00% (0/1) |
| **Overall** | 0.49% (4/819) | 58.49% (479/819) | 41.03% (336/819) |

Table 65: Weighted Results per Type and Overall (Prediction for **WildChat**), Model: **claude-sonnet-4-20250514**

| Type | Weighted Redact | Weighted Abstract | Weighted Retain |
|---|---|---|---|
| NAME | 51.97% (79/152) | 47.37% (72/152) | 0.66% (1/152) |
| AFFILIATION | 34.27% (49/143) | 65.73% (94/143) | 0.00% (0/143) |
| GEOLOCATION | 35.86% (85/237) | 64.14% (152/237) | 0.00% (0/237) |
| USERNAME | 0.00% (0/2) | 100.00% (2/2) | 0.00% (0/2) |
| TIME | 40.94% (61/149) | 57.72% (86/149) | 1.34% (2/149) |
| AGE | 31.82% (7/22) | 63.64% (14/22) | 4.55% (1/22) |
| OCCUPATION | 13.51% (5/37) | 86.49% (32/37) | 0.00% (0/37) |
| QUANTITY | 0.00% (0/6) | 100.00% (6/6) | 0.00% (0/6) |
| ETHNICITY | 66.67% (6/9) | 33.33% (3/9) | 0.00% (0/9) |
| GENDER | 71.43% (5/7) | 28.57% (2/7) | 0.00% (0/7) |
| EMAIL | 100.00% (2/2) | 0.00% (0/2) | 0.00% (0/2) |
| URL | 75.00% (6/8) | 25.00% (2/8) | 0.00% (0/8) |
| HEALTH_INFORMATION | 16.67% (1/6) | 83.33% (5/6) | 0.00% (0/6) |
| RACE | 100.00% (1/1) | 0.00% (0/1) | 0.00% (0/1) |
| INCOME | 0.00% (0/14) | 100.00% (14/14) | 0.00% (0/14) |
| PRODUCT | 0.00% (0/1) | 100.00% (1/1) | 0.00% (0/1) |
| FINANCIAL_INFORMATION | 33.33% (2/6) | 66.67% (4/6) | 0.00% (0/6) |
| PHONE_NUMBER | 100.00% (2/2) | 0.00% (0/2) | 0.00% (0/2) |
| EDUCATIONAL_RECORD | 18.18% (2/11) | 81.82% (9/11) | 0.00% (0/11) |
| ID_NUMBER | 100.00% (2/2) | 0.00% (0/2) | 0.00% (0/2) |
| KEYS | 100.00% (1/1) | 0.00% (0/1) | 0.00% (0/1) |
| GPA | 0.00% (0/1) | 100.00% (1/1) | 0.00% (0/1) |
| **Overall** | 38.58% (316/819) | 60.93% (499/819) | 0.49% (4/819) |

Table 66: Weighted Results per Type and Overall (Prediction for **WildChat**), Model: **lgai/exaone-deep-32b**

| Type | Weighted Redact | Weighted Abstract | Weighted Retain |
|---|---|---|---|
| NAME | 1.97% (3/152) | 97.37% (148/152) | 0.66% (1/152) |
| AFFILIATION | 0.00% (0/143) | 98.60% (141/143) | 1.40% (2/143) |
| GEOLOCATION | 0.00% (0/237) | 99.58% (236/237) | 0.42% (1/237) |
| USERNAME | 50.00% (1/2) | 50.00% (1/2) | 0.00% (0/2) |
| TIME | 0.00% (0/149) | 97.99% (146/149) | 2.01% (3/149) |
| AGE | 4.55% (1/22) | 95.45% (21/22) | 0.00% (0/22) |
| OCCUPATION | 0.00% (0/37) | 94.59% (35/37) | 5.41% (2/37) |
| QUANTITY | 0.00% (0/6) | 100.00% (6/6) | 0.00% (0/6) |
| ETHNICITY | 0.00% (0/9) | 100.00% (9/9) | 0.00% (0/9) |
| GENDER | 0.00% (0/7) | 100.00% (7/7) | 0.00% (0/7) |
| EMAIL | 0.00% (0/2) | 50.00% (1/2) | 50.00% (1/2) |
| URL | 12.50% (1/8) | 87.50% (7/8) | 0.00% (0/8) |
| HEALTH_INFORMATION | 0.00% (0/6) | 100.00% (6/6) | 0.00% (0/6) |
| RACE | 0.00% (0/1) | 100.00% (1/1) | 0.00% (0/1) |
| INCOME | 0.00% (0/14) | 100.00% (14/14) | 0.00% (0/14) |
| PRODUCT | 0.00% (0/1) | 100.00% (1/1) | 0.00% (0/1) |
| FINANCIAL_INFORMATION | 0.00% (0/6) | 100.00% (6/6) | 0.00% (0/6) |
| PHONE_NUMBER | 0.00% (0/2) | 100.00% (2/2) | 0.00% (0/2) |
| EDUCATIONAL_RECORD | 0.00% (0/11) | 27.27% (3/11) | 72.73% (8/11) |
| ID_NUMBER | 0.00% (0/2) | 100.00% (2/2) | 0.00% (0/2) |
| KEYS | 0.00% (0/1) | 100.00% (1/1) | 0.00% (0/1) |
| GPA | 0.00% (0/1) | 100.00% (1/1) | 0.00% (0/1) |
| **Overall** | 0.73% (6/819) | 97.07% (795/819) | 2.20% (18/819) |

Table 67: Weighted Results per Type and Overall (Prediction for **WildChat**), Model: **mistralai/mistral-small-3.1-24b-instruct**

| Type | Weighted Redact | Weighted Abstract | Weighted Retain |
|---|---|---|---|
| NAME | 3.95% (6/152) | 96.05% (146/152) | 0.00% (0/152) |
| AFFILIATION | 0.00% (0/143) | 97.90% (140/143) | 2.10% (3/143) |
| GEOLOCATION | 1.27% (3/237) | 98.73% (234/237) | 0.00% (0/237) |
| USERNAME | 0.00% (0/2) | 100.00% (2/2) | 0.00% (0/2) |
| TIME | 4.03% (6/149) | 95.30% (142/149) | 0.67% (1/149) |
| AGE | 0.00% (0/22) | 100.00% (22/22) | 0.00% (0/22) |
| OCCUPATION | 0.00% (0/37) | 100.00% (37/37) | 0.00% (0/37) |
| QUANTITY | 0.00% (0/6) | 100.00% (6/6) | 0.00% (0/6) |
| ETHNICITY | 0.00% (0/9) | 100.00% (9/9) | 0.00% (0/9) |
| GENDER | 0.00% (0/7) | 100.00% (7/7) | 0.00% (0/7) |
| EMAIL | 0.00% (0/2) | 100.00% (2/2) | 0.00% (0/2) |
| URL | 0.00% (0/8) | 100.00% (8/8) | 0.00% (0/8) |
| HEALTH_INFORMATION | 0.00% (0/6) | 100.00% (6/6) | 0.00% (0/6) |
| RACE | 0.00% (0/1) | 100.00% (1/1) | 0.00% (0/1) |
| INCOME | 0.00% (0/14) | 100.00% (14/14) | 0.00% (0/14) |
| PRODUCT | 0.00% (0/1) | 100.00% (1/1) | 0.00% (0/1) |
| FINANCIAL_INFORMATION | 0.00% (0/6) | 100.00% (6/6) | 0.00% (0/6) |
| PHONE_NUMBER | 0.00% (0/2) | 100.00% (2/2) | 0.00% (0/2) |
| EDUCATIONAL_RECORD | 0.00% (0/11) | 100.00% (11/11) | 0.00% (0/11) |
| ID_NUMBER | 0.00% (0/2) | 100.00% (2/2) | 0.00% (0/2) |
| KEYS | 100.00% (1/1) | 0.00% (0/1) | 0.00% (0/1) |
| GPA | 0.00% (0/1) | 100.00% (1/1) | 0.00% (0/1) |
| **Overall** | 1.95% (16/819) | 97.56% (799/819) | 0.49% (4/819) |

Table 68: Weighted Results per Type and Overall (Prediction for **WildChat**), Model: **qwen/qwen2.5-7b-instruct**