# OpenReview forum: "Operationalizing Data Minimization for Privacy-Preserving LLM Prompting"
_ICLR.cc/2026/Conference — ICLR 2026 Poster_

### Official Review · Reviewer_NzwN · 2025-10-25

**Soundness:** 4
**Presentation:** 4
**Contribution:** 3
**Rating:** 8
**Confidence:** 4

**Summary:**

This paper presents a framework to formally define and operationalize data minimization, in the context of users sharing personal information to LLMs. For a given user prompt and response model, they quantify the least privacy-revealing disclosure that maintains utility, and propose a priority-queue tree search to locate this. They evaluated this on a diverse set of four datasets and nine LLMs, and show that for the same user prompts, larger frontier LLMs can have stronger data minimization while maintaining task quality. They also show that LLMs are poor predictors of data minimization and have a bias towards oversharing.

**Strengths:**

- This paper formally operationalizes data minimization in the context of privacy-preserving prompts for LLMs and presents an algorithm for doing this, which is different from existing approaches that mainly focus on detecting personal information and redacting or abstracting it.
- The authors ran a systematic evaluation on a good range of datasets and LLMs and were able to show how more powerful frontier models can be better for data minimization.

**Weaknesses:**

- It is not clear how to directly apply these findings to real applications, since the data minimization algorithm involves querying and checking the utility for multiple variants of the original prompt that will reveal the personal information, and might also take significant cost and latency.
- It would be good to include more details about the PII annotation, such as how the annotators were chosen, and more information about the amount of consensus. For instance, in table 1, how many examples are there with human consensus at least 0.8?

**Questions:**

- How could the results be applied to real applications?
- Could more details about the PII annotation be provided?

---

> ### Author Response · Authors · 2025-11-21
>
> Thank you very much for the clear review. We especially appreciate that you found both the formulation and the systematic evaluation compelling. Your questions helped us clarify important aspects of how the framework can be applied and how the human annotations were used, and we are grateful for the opportunity to strengthen our explanations.
>
>
> **Weakness 1 + Question 1**
>
> Thank you for raising this important question about real-world applicability. The full data-minimization algorithm in the paper, *oracle*, is *not* the procedure that a deployed system would run. Its purpose is analytical: it computes the upper bound of how much information can be safely removed for each model. Because it is an offline process, it can afford to query multiple prompt variants in order to fully map out the space of utility-preserving transformations.
>
> In an actual application, users would not interact with this search at all. A deployed system would instead rely on a *single-pass predictor* that produces **retain / abstract / redact** decisions in one step before the prompt is sent to an external API model. This predictor can be trained or distilled from the oracle outputs, and therefore inherits the oracle’s definition of “safe minimization” without incurring any additional latency or revealing any intermediate prompt variants to the cloud model.
>
> To understand whether prompts minimized by a smaller local predictor would still work well for a larger cloud LLM, we added a new experiment during the rebuttal period. We compared each model’s oracle to the GPT-5 oracle and measured their overlap. For **redaction**, the overlap is consistently high (around *75–85%* across datasets), meaning that different models largely agree on which spans must be removed to preserve utility (more details can be found in our answer to reviewer nGjY's Question 2). Although abstraction overlap is lower, abstraction accounts for a much smaller share of the minimization decisions. These findings indicate that minimized prompts produced by a smaller on-device predictor can transfer effectively to the larger cloud model that ultimately performs the task.
>
> In summary, the oracle is an offline mechanism used to define the upper bound of safe minimization, while the real-world deployment relies on a lightweight, single-pass predictor. The new overlap experiment shows that the resulting minimized prompts remain stable across models, supporting their use in practical privacy-preserving prompting pipelines.
>
>
> ---
>
> **Weakness 2 + Question 2**
>
> Thank you for the thoughtful question and for carefully examining the annotation setup. We are happy to explain this part.
>
> All PII spans are detected automatically by GPT-4o. We use GPT-4o because prior work — and our own pilot checks — show that strong LLMs are reliable for span detection in this setting. The human annotators are *not* asked to identify PII. Instead, they evaluate **privacy preferences** between two masked versions of the same message. That is, they compare how two masking choices affect the perceived level of privacy, rather than labeling which spans contain PII.
>
> Because privacy preference is subjective, we report consensus levels to indicate how strongly annotators agree on each comparison. The agreement numbers in Table 1 are therefore conditioned on *privacy-preference* consensus, not PII-labeling consensus.
>
> Regarding your specific question:
> - among the 150 A/B comparison pairs, **121** reach consensus ≥ 0.6, and
> - **73** reach consensus ≥ 0.8.
>
> We will include these exact counts in the revised version for clarity.
>
> We appreciate the reviewer highlighting this point, and we will make the distinction between PII detection and privacy-preference evaluation more explicit in our revised version.

---

> > ### Comment · Reviewer_NzwN · 2025-11-27
> >
> > Thank you for the detailed responses and clarifications. I will maintain my positive score.

---

> > > ### Author Response · Authors · 2025-11-27
> > >
> > > Thank you for your feedback. We sincerely appreciate your evaluation and support.

---

### Official Review · Reviewer_Djhr · 2025-10-29

**Soundness:** 3
**Presentation:** 3
**Contribution:** 3
**Rating:** 6
**Confidence:** 4

**Summary:**

The paper proposes a framework for reducing privacy leakage in LLM prompting. It first identifies the sensitive spans in the prompt and introduces a priority-queue tree search algorithm that explores combinations of RETAIN / ABSTRACT / REDACT operations on the spans.

**Strengths:**

1.	The paper addresses a important problem as how to minimize the privacy leakage of LLM prompt.
The paper addresses an important and timely problem — practical data minimization for LLM usage — with clear motivation.
2.	The paper includes comprehensive experimental evaluation with  multiple datasets, models, and attack evaluations.
3.	Although without strict privacy leakage guarantee, the proposed ranking-based evaluation method offers an insightful and practical perspective for quantifying privacy leakage.

**Weaknesses:**

1. The evaluation of privacy relies on pairwise comparison, lacking of theoretical guarantees.
2. The paper evaluates utility by comparing the generated output with the target response, treating it as a binary classification problem. This oversimplification fails to reflect the continuous performance characteristics of modern LLMs.
3. The practicability of the proposed method remains questionable, as it assumes there is a local deployed LLM while the aim of the paper is to protect privacy of  prompt  when using LLM via API.

**Questions:**

1. Could you please present how sensitive are the utility threshold gamma? From my point of view, a little gamma may lead to significant performance degradation in LLMs.
2. I am curious about  whether the “utility predictor” can be generalized across tasks or must be retrained for each task?

3.During the search process, are all intermediate prompts exposed to the main LLM? If so, how is privacy leakage mitigated? If using different LLMs, assuming a smaller LlM local deployed, can the obtained privacy-preserved still transfere effectively to the larger main LLM?

---

> ### Author Response · Authors · 2025-11-21
>
> Thank you very much for the thoughtful review. We really appreciate that you highlighted both the motivation and the practicality of our problem setting, and we are glad that the breadth of experiments and the overall framing of semantic data minimization came through. Your comments, especially on evaluation methodology and deployability, helped us articulate the scope and limitations of the work more precisely.
>
> **Weakness 1**
> Thank you for raising this point. We agree that our pairwise privacy comparison does not provide a formal theoretical guarantee. This reflects a broader challenge in evaluating *semantic*, utility-preserving privacy: unlike syntactic notions of privacy, there is currently no analytic metric that can determine whether one prompt semantically reveals “less sensitive information” while still allowing the LLM to complete the task.
>
> It may be helpful to distinguish this setting from differential privacy. Differential privacy offers strong theoretical guarantees, but it is designed for *training* and for certain types of inference mechanisms. Our work focuses on the *prompt layer*, before any interaction with the model occurs, and the privacy notion here is inherently semantic: we aim to remove or abstract sensitive meanings in a way that preserves downstream utility. This type of “semantic privacy” does not yet have a closed-form theoretical formulation, which is why preference-based comparison is widely used in related areas such as importance estimation, safety judgments, attribution, and RLHF.
>
> Within this limitation, we ground our comparator in human preferences. It is trained on human-labeled comparisons and validated against human consensus to ensure that its behavior reflects human views of what counts as more or less privacy-preserving. While this is not a mathematical guarantee, it provides an empirically stable and human-aligned notion of correctness for semantic evaluation. Moreover, our approach is complementary to rigorous methods such as differential privacy — nothing in our pipeline prevents later stages from applying DP mechanisms on top of the semantically minimized prompt.
>
> **Weakness 3**
> Thank you for bringing this up. We agree that the full oracle search is not meant to run on-device and would be impractical in a deployment setting. In our framework, the oracle is only used to establish the *upper bound* of achievable data minimization, so that we can quantify oversharing and evaluate how well different models approximate this bound.
>
> The component intended for real-world use is the predictor model described in Section 5.4. This predictor produces {retain, abstract, redact} decisions in a single pass and can be packaged as a small local model. Such a model can be directly fine-tuned or distilled from the oracle outputs, allowing users to perform data minimization locally before sending any prompt to a cloud LLM API. This aligns with how practical sanitization workflows are typically deployed.
>
> In summary, the oracle defines the target behavior, while the predictor represents the path toward practical deployment. Our current single-pass predictors are commercial models and therefore not yet suitable for on-device use, but the oracle gives us the ground-truth supervision needed to eventually train or distill a lightweight local predictor. This remains an important direction for future work.

---

> ### Author Response · Authors · 2025-11-21
>
> **Weakness 2 + Question 1 (utility evaluation & gamma sensitivity)**
>
> Thank you for raising this point. We agree that LLM utility can be continuous, but in our framework utility is not used as a scoring metric. Instead, it functions as a *hard constraint*: “does this transformation still preserve the task outcome?” Under this constrained-optimization view, a binary pass/fail check is the standard formulation and has been commonly used in similar constrained-LLM settings.
>
> To directly address your concern, we conducted an additional experiment that evaluates the sensitivity of the utility threshold γ. In the main paper, our utility check is binary because the evaluator returns a categorical judgment, and because strict preservation is the most natural formulation for open-ended datasets (ShareGPT, WildChat). Closed-ended datasets already have fixed gold answers, so γ does not apply there.
>
> For the new study, we modified the evaluator so that it returns a **1–10 utility score**, with the original response defined as “10.” We then re-ran our full pipeline on ShareGPT and WildChat using γ ∈ {0, 1, 2}, meaning the candidate response must score ≥10, ≥9, or ≥8 to pass. All other components follow the same pipeline as in Section X of the paper.
>
> After obtaining the minimized prompts for each γ, we ran a **90-participant Prolific study** to evaluate human-perceived utility differences. For each of 90 sampled prompts, participants compared the original response against a response from one of the γ-pipelines (randomly selected, without duplication), choosing whether A, B, or SAME quality.
>
> The results were:
>
> - γ = 0: SAME = 36.8%, original = 32.6%, masked = 30.5%
> - γ = 1: SAME = 30.9%, original = 42.5%, masked = 26.6%
> - γ = 2: SAME = 28.1%, original = 44.1%, masked = 27.8%
>
> We see a clear trend: as γ increases (i.e., the utility requirement is relaxed), the SAME rate consistently drops and preference for the original response increases. Even small relaxations therefore lead to *human-perceptible* quality degradation.
>
> This directly supports our design choice in the paper: a strict binary predicate (utility must remain unchanged) is the most reliable and conservative formulation for utility-preserving minimization. Although LLM utility is continuous in general, our user study confirms that even small γ leads to noticeable drops in perceived quality.
>
> We appreciate the reviewer for pointing out this issue; it helped us strengthen the empirical justification for the binary utility predicate used in our method.

---

> ### Author Response · Authors · 2025-11-21
>
> **Question 2**
>
> Thank you for the question. In our framework, the *“utility predictor”* refers to the LLM-as-a-judge that checks whether a transformed prompt still preserves the quality of the response. For the open-ended datasets we study (*ShareGPT* and *WildChat*), which resemble real-world conversational queries, this judge follows a general rubric appropriate for everyday tasks.
>
> To confirm that it actually matches human perceptions of utility, we conducted a new human study during the rebuttal period. We compared GPT-4o’s PASS/FAIL decisions against judgments from 75 Prolific participants (more details are provided in our response to reviewer oafx’s Question 2). This directly validates that the judge aligns with human preferences in the exact scenarios where it is used.
>
> Our paper evaluates both open-ended and closed-ended settings because they reflect two distinct notions of utility: open-ended tasks emphasize meaning and helpfulness, while closed-ended tasks such as *MedQA* or *CaseHold* require fixed correctness. We appreciate the reviewer pointing out this distinction; in domain-specific or high-stakes settings, the utility judge can indeed be adapted or fine-tuned to match domain requirements.
>
> In summary, the judge we use is validated for open-ended tasks through the new human study, and closed-ended tasks are handled separately with fixed-answer criteria. This separation is intentional and reflects the two utility regimes our work aims to cover.
>
> ---
>
> **Question 3**
>
> Thank you for raising this question. Computing the oracle does require querying the response model on intermediate, partially sanitized prompts, because our goal is to establish the *upper bound* of data minimization for each model. This is the first time such an upper bound has been computed, and doing so necessarily involves interacting with non-final variants of the prompt. In deployment, however, users would never run this full search; the oracle exists so that a predictor can eventually replace the entire iterative process.
>
> To ensure that exposing intermediate prompts does not introduce privacy leakage, we added a self-attack experiment in the rebuttal period. Even when GPT-5 is asked to recover the original PII from its own minimized inputs, recovery remains near 0–3% across datasets. This suggests that the sanitized intermediate prompts simply do not contain enough signal for a model, or even the response model itself to infer the removed PII.
>
> Regarding transferability to a larger main LLM, we added a second experiment designed specifically to address this concern (more details can be found in our answer to reviewer nGjY's Question 2). We computed the overlap between each model’s oracle and the GPT-5 oracle. For *redaction*, overlap is consistently high (typically 75–85% across datasets), showing strong agreement on which spans must be removed. Although *abstraction* overlap is lower, abstraction accounts for a much smaller share of operations. Overall, this indicates that the core privacy-preserving operations are largely stable across models.
>
> In short, the oracle provides the upper bound, the predictor will eventually remove the need for intermediate prompting, and our new overlap experiment shows that minimized prompts produced by a smaller local model transfer well to larger cloud LLMs. This helps supporting the deployment scenario raised by the reviewer.

---

### Official Review · Reviewer_nGjY · 2025-10-31

**Soundness:** 3
**Presentation:** 3
**Contribution:** 2
**Rating:** 4
**Confidence:** 3

**Summary:**

The work addresses the problem of "oversharing" of personal data in LLM prompts. The authors study the data minimization problem as a constrained optimization problem where they want to maximize privacy with a strict constraint on utility degradation. The work aims to find minimal prompts for each model that does not degrade utility. The paper proposes a "Freeze-then-Search" algorithm to find this optimal prompt. The search space consists of span-level transformations: {RETAIN, ABSTRACT, REDACT} where RETAIN is the least private and REDACT is the most private. To obtain an full ordering on the space, the authors utility a privacy comparator model that is a fine-tuned Qwen2.5-7B-Instruct model.  The search is conduced using a priority-queue tree search and terminates once a prompt is found that satisfies the utility constraint. To measure that utility constraint, the authors use a judge model (GPT-4o) to measure whether the answer from the minimized prompt is compared to a reference response.

The work finds these minimized prompts for 9 LLMs (from small models like Qwen-0.5B to large ones like GPT-5) on 4 datasets. The authors find that large models are tolerant of far more minimization than small models.

The work also tries to understand if LLMs are good predictors of these minimized prompts and find that all LLMs are poor predictors of this. They also find that the predictors exhibit a strong bias towards ABSTRACT rather than using REDACT as in the minimized prompts.

**Strengths:**

- Paper does a good job of formulating the data minimization problem as a constrained optimization problem including defining a well ordered search space.
- The generated minimized prompts seem useful to make progress on data minimization predictors.
- The insight that models are poor predictors of predicting the minimal prompt required is interesting.

**Weaknesses:**

- The minimized prompts are model specific.
-

**Questions:**

- Is it the case that large models require less information is probably because they are able to infer the missing pieces of information better?
- The "abstract" bias seems interesting. Could it be related to how the model is instructed to generate its minimal prompt?

---

> ### Author Response · Authors · 2025-11-21
>
> Thank you for the thoughtful review. We really appreciate your clear summary of the paper and are glad that the formulation of the problem, the search framework, and the analysis were helpful. Your comments on model-specificity and abstraction bias were especially valuable and motivated additional experiments that we added during this rebuttal period.

---

> ### Author Response · Authors · 2025-11-21
>
> **Weakness**
>
> We agree that different models can yield different minimized prompts. However, this is an inherent property of data minimization rather than a limitation of our method: the minimal information needed to preserve utility depends on the target model’s inference ability. One aim of our oracle is to *make this variation explicit*.
>
> To address your comment more directly, we added a new experiment comparing each model’s oracle to the GPT-5 oracle and computing their overlap. For each decision type (**REDACT**, **ABSTRACT**), we compute a Jaccard overlap over PII spans selected by both vs. either oracle. This captures how stable minimization decisions are across models.
>
> For **redaction**, overlap is consistently high across nearly all models and datasets (around or above **80%**), indicating that most removed spans are agreed upon broadly across models. For **abstraction**, overlap is lower, but abstraction accounts for a much smaller portion of the decisions, so the impact is limited. Most of the “core” privacy removals are shared across models, and model-specific differences appear mostly in the optional abstraction region.
>
> Overall, while minimal prompts are inherently model-dependent, the new experiment shows that the *core* redactions generalize well across models. The oracle reveals how minimization tolerance varies by model, and this variation itself is an important insight. Improving cross-model transferability is a promising future direction, but the current results already show strong consistency where it matters most.
>
> ### Redaction Overlap (vs. GPT-5 Oracle)
> | dataset | model | overlap |
> |------------------|-------------------------|----------|
> | ShareGPT | gpt-4.1-nano | 0.802493 |
> | ShareGPT | gpt-4.1 | 0.845057 |
> | ShareGPT | claude-3-7-sonnet-20250219 | 0.792776 |
> | ShareGPT | claude-sonnet-4-20250514 | 0.754572 |
> | ShareGPT | lgai_exaone-deep-32b | 0.583899 |
> | ShareGPT | mistralai_mistral-small-3.1-24b-instruct | 0.743466 |
> | ShareGPT | qwen_qwen-2.5-7b-instruct | 0.686103 |
> | ShareGPT | local_qwen2.5-0.5b-instruct | 0.145875 |
> | wildchat | gpt-4.1-nano | 0.849807 |
> | wildchat | gpt-4.1 | 0.860465 |
> | wildchat | claude-3-7-sonnet-20250219 | 0.797419 |
> | wildchat | claude-sonnet-4-20250514 | 0.800771 |
> | wildchat | lgai_exaone-deep-32b | 0.701961 |
> | wildchat | mistralai_mistral-small-3.1-24b-instruct | 0.853816 |
> | wildchat | qwen_qwen-2.5-7b-instruct | 0.794839 |
> | wildchat | local_qwen2.5-0.5b-instruct | 0.298128 |
> | medQA | gpt-4.1-nano | 0.875905 |
> | medQA | gpt-4.1 | 0.950464 |
> | medQA | claude-3-7-sonnet-20250219 | 0.708768 |
> | medQA | claude-sonnet-4-20250514 | 0.952183 |
> | medQA | lgai_exaone-deep-32b | 0.762055 |
> | medQA | mistralai_mistral-small-3.1-24b-instruct | 0.954451 |
> | medQA | qwen_qwen-2.5-7b-instruct | 0.889583 |
> | medQA | local_qwen2.5-0.5b-instruct | 0.292683 |
> | casehold | gpt-4.1-nano | 0.945274 |
> | casehold | gpt-4.1 | 0.985075 |
> | casehold | claude-3-7-sonnet-20250219 | 0.987531 |
> | casehold | claude-sonnet-4-20250514 | 0.987562 |
> | casehold | lgai_exaone-deep-32b | 0.736181 |
> | casehold | mistralai_mistral-small-3.1-24b-instruct | 0.937811 |
> | casehold | qwen_qwen-2.5-7b-instruct | 0.957711 |
> | casehold | local_qwen2.5-0.5b-instruct | 0.395522 |
>
> ### Abstraction Overlap (vs. GPT-5 Oracle)
> | dataset | model | overlap |
> |------------------|-------------------------|----------|
> | ShareGPT | gpt-4.1-nano | 0.203704 |
> | ShareGPT | gpt-4.1 | 0.256039 |
> | ShareGPT | claude-3-7-sonnet-20250219 | 0.182648 |
> | ShareGPT | claude-sonnet-4-20250514 | 0.178261 |
> | ShareGPT | lgai_exaone-deep-32b | 0.090625 |
> | ShareGPT | mistralai_mistral-small-3.1-24b-instruct | 0.189189 |
> | ShareGPT | qwen_qwen-2.5-7b-instruct | 0.152941 |
> | ShareGPT | local_qwen2.5-0.5b-instruct | 0.014354 |
> | wildchat | gpt-4.1-nano | 0.141414 |
> | wildchat | gpt-4.1 | 0.132653 |
> | wildchat | claude-3-7-sonnet-20250219 | 0.114035 |
> | wildchat | claude-sonnet-4-20250514 | 0.085470 |
> | wildchat | lgai_exaone-deep-32b | 0.108974 |
> | wildchat | mistralai_mistral-small-3.1-24b-instruct | 0.126214 |
> | wildchat | qwen_qwen-2.5-7b-instruct | 0.090909 |
> | wildchat | local_qwen2.5-0.5b-instruct | 0.052326 |
> | medQA | gpt-4.1-nano | 0.023810 |
> | medQA | gpt-4.1 | 0.093750 |
> | medQA | claude-3-7-sonnet-20250219 | 0.060403 |
> | medQA | claude-sonnet-4-20250514 | 0.175000 |
> | medQA | lgai_exaone-deep-32b | 0.031008 |
> | medQA | mistralai_mistral-small-3.1-24b-instruct | 0.111111 |
> | medQA | qwen_qwen-2.5-7b-instruct | 0.090909 |
> | medQA | local_qwen2.5-0.5b-instruct | 0.026316 |
> | casehold | gpt-4.1-nano | 0.000000 |
> | casehold | gpt-4.1 | 0.000000 |
> | casehold | claude-3-7-sonnet-20250219 | 0.200000 |
> | casehold | claude-sonnet-4-20250514 | 0.000000 |
> | casehold | lgai_exaone-deep-32b | 0.000000 |
> | casehold | mistralai_mistral-small-3.1-24b-instruct | 0.000000 |
> | casehold | qwen_qwen-2.5-7b-instruct | 0.000000 |
> | casehold | local_qwen2.5-0.5b-instruct | 0.000000 |

---

> ### Author Response · Authors · 2025-11-21
>
> **Question 1**
> Thank you for raising this insightful question. To directly test whether larger models “need less information” because they can better infer the missing private details, we added a new privacy-audit experiment in the rebuttal period. In addition to the two attacker models in Section 5.3, we now let **GPT-5 attack the minimized prompts generated for GPT-5 itself**. This is the strictest possible test because the response model and the attacker are identical.
>
> To clarify the evaluation (which is also used in Sec 5.3 ):
> 1) *Hit@1 / Hit@3* measure whether the attacker can guess (top guess/3 guesses) the exact original PII value from the masked prompt.
> 2) *Masked Hit@1 = 0%* means the model never recovers the correct PII.
> 3) The “masked prompt” refers to the minimized prompt after abstraction/redaction.
>
> On WildChat, GPT-5’s recovery of masked PII drops to **0% for both Hit@1 and Hit@3**, despite being around 90–100% on the original (unmasked) prompts. In other words, once the span is masked, there is no semantic signal left for the model to infer back the true value.
>
> We see the same pattern across all PII categories: masked Hit@1 remains **0–3%**, indicating that GPT-5 cannot fill in the missing details. This rules out the hypothesis that large models accept more minimization because they “guess” the missing pieces.
>
> For the span-wise evaluation (action-level analysis), the numbers tell the same story. Here:
> - **correct_rate** = proportion of masked spans where GPT-5’s best guess matches the original PII
> - **unknown_rate** = proportion where GPT-5 answers “I don’t know” or equivalent
> - **mean_conf** = model’s confidence in its guesses
>
> | action | N | correct_rate | correct_lo | correct_hi | unknown_rate | unknown_lo | unknown_hi | mean_conf |
> |------------|-------|--------------|--------------|--------------|--------------|--------------|--------------|--------------|
> | abstract | 55 | 0.018182 | 0.000000 | 0.054545 | 0.472727 | 0.345455 | 0.600000 | 0.524545 |
> | redact | 735 | 0.040816 | 0.027211 | 0.055782 | 0.922449 | 0.902041 | 0.941497 | 0.066558 |
>
>
> We observe consistently low correct-recovery rates across all datasets, mostly below 13%, and high unknown rates that typically range from 30% to nearly 100%. This confirms that GPT-5 is *not able to reconstruct the removed private information*, even when directly prompted to do so.
>
>
>
> Putting everything together, the results show that larger models tolerate more minimization because they are more robust in solving tasks with less context, *not* because they are better at inferring or hallucinating the hidden PII.
>
> ---
>
> **Question 2**
> Thanks for pointing this out. We also thought the abstraction bias was interesting, and your question made us look into it more carefully. Our current understanding is that there are a few reasons behind it.
>
> First, models have a **formatting prior**: they are trained on natural-language text, not placeholder patterns. So abstraction feels more “natural” than inserting something like `[NAME3]`.
>
> Second, instruction-tuned models tend to preserve **coherence and readability**. Abstracting keeps the sentence flowing; redaction breaks the structure, so models instinctively lean toward abstraction.
>
> Third, we ran an **additional experiment** to check whether our own prompt wording caused the bias. In Appendix F.2, the system prompt contains the line:
> “*prefer stronger only when quality/correctness is unchanged*.”
>
> We wondered whether this discouraged REDACT. So we tested two ablations:
> 1) **order_only** — remove the “prefer stronger…” clause
> 2) **no order/notion** — remove the entire minimization-order line
>
> We ran *GPT-5*, *Mistral-small-3.1-24B*, and *Qwen2.5-7B* on ShareGPT and MedQA under all three prompt variants. The results were consistent: removing the clause does **not** eliminate the abstraction bias, and even removing the entire sentence still leads models to strongly prefer ABSTRACT. Redaction increases slightly, but the overall pattern remains. The result table will be shown in the next comment.
>
> So overall, the abstraction bias does not come from our prompt. It appears to be a **built-in tendency of instruction-tuned LLMs**: when asked to “maintain quality,” they naturally gravitate toward abstraction because it preserves fluency. Our search-based oracle, in contrast, has an explicit utility check and therefore ends up much more **REDACT-heavy**, without relying on these internal heuristics.

---

> ### Author Response · Authors · 2025-11-21
>
> | dataset   | model                                      | prompt type        | total # of PII | redact          | abstract          |
> |------------------|--------------------------------|----------------|-------------|----------------|----------------|
> | ShareGPT  | gpt-5                                       | order+notion        | 1190           | 164 (13.8%)      | 663 (55.7%)        |
> | ShareGPT  | gpt-5                                       | order_only          | 1190           | 225 (18.9%)      | 656 (55.1%)        |
> | ShareGPT  | gpt-5                                       | no order/notion     | 1190           | 153 (12.9%)      | 728 (61.2%)        |
> | ShareGPT  | mistralai_mistral-small-3.1-24b-instruct    | order+notion        | 1190           | 6 (0.5%)         | 1155 (97.1%)       |
> | ShareGPT  | mistralai_mistral-small-3.1-24b-instruct    | order_only          | 1190           | 20 (1.7%)        | 1070 (89.9%)       |
> | ShareGPT  | mistralai_mistral-small-3.1-24b-instruct    | no order/notion     | 1190           | 20 (1.7%)        | 1027 (86.3%)       |
> | ShareGPT  | qwen_qwen-2.5-7b-instruct                   | order+notion        | 1190           | 16 (1.3%)        | 1174 (98.7%)       |
> | ShareGPT  | qwen_qwen-2.5-7b-instruct                   | order_only          | 1190           | 38 (3.2%)        | 1116 (93.8%)       |
> | ShareGPT  | qwen_qwen-2.5-7b-instruct                   | no order/notion     | 1190           | 139 (11.7%)      | 1000 (84.0%)       |
> | medQA     | gpt-5                                       | order+notion        | 976            | 533 (54.6%)      | 405 (41.5%)        |
> | medQA     | gpt-5                                       | order_only          | 976            | 808 (82.8%)      | 140 (14.3%)        |
> | medQA     | gpt-5                                       | no order/notion     | 976            | 688 (70.5%)      | 258 (26.4%)        |
> | medQA     | mistralai_mistral-small-3.1-24b-instruct    | order+notion        | 976            | 0 (0.0%)         | 912 (93.4%)        |
> | medQA     | mistralai_mistral-small-3.1-24b-instruct    | order_only          | 976            | 2 (0.2%)         | 609 (62.4%)        |
> | medQA     | mistralai_mistral-small-3.1-24b-instruct    | no order/notion     | 976            | 2 (0.2%)         | 531 (54.4%)        |
> | medQA     | qwen_qwen-2.5-7b-instruct                   | order+notion        | 976            | 0 (0.0%)         | 973 (99.7%)        |
> | medQA     | qwen_qwen-2.5-7b-instruct                   | order_only          | 976            | 2 (0.2%)         | 940 (96.3%)        |
> | medQA     | qwen_qwen-2.5-7b-instruct                   | no order/notion     | 976            | 19 (1.9%)        | 928 (95.1%)        |
>
> *Retained spans are omitted for brevity.*
>
> From the table, the abstraction bias clearly does *not* come from our prompt wording. Even after removing the preference clause (*order_only*) or removing the entire minimization-order instruction (*no order/notion*), all models still choose **ABSTRACT** at very high rates. For example: GPT-5 stays around *55–61%*, Mistral-24B remains *86–97%*, and Qwen2.5-7B stays above *84%* across all variants. If the bias came from our prompt, these numbers would shift dramatically toward **REDACT**, but they do not.
>
> The small changes in redaction confirm that the bias is structural to instruction-tuned LLMs rather than induced by our prompt.

---

> > ### Comment · Reviewer_nGjY · 2025-11-25
> >
> > Thanks for the additional experiments and answering my questions. Some of the analysis might be a nice addition to the paper. Based on this, I have increased my rating from 4 to 6.

---

> > > ### Author Response · Authors · 2025-11-26
> > >
> > > Thank you very much for the update and for raising your rating! We’re glad the additional analysis was helpful, and we’ll incorporate it into the final version. Thanks again for your thoughtful review.

---

### Official Review · Reviewer_ofax · 2025-11-01

**Soundness:** 3
**Presentation:** 3
**Contribution:** 2
**Rating:** 4
**Confidence:** 4

**Summary:**

This paper introduces a framework to define and operationalize the principle of data minimization (PII redaction) for LLM prompting. The authors formulate data minimization as an optimization problem: finding the most privacy-preserving version of a user prompt (by applying a series of actions to sensitive spans) that still maintains a minimum level of task utility for a given response LLM.

**Strengths:**

The paper is well-written, with motivated and formalized problem settings. The introduction and related work sections do a good job of comparing the data minimization formulation within existing privacy-preserving techniques like DP-training or simple sanitization.

**Weaknesses:**

1. The most significant weakness is the lack of comparison against simple, non-LLM baselines. The paper compares its computationally expensive oracle to single-pass LLM predictors. However, it fails to include a much simpler and more practical baseline: using a standard PII identification or NER tool to identify and redact PIIs.
2.  The methodology relies on an LLM-as-a-Judge (GPT-4o) to evaluate utility for open-ended tasks. While the authors commendably validate their privacy comparator LLM against human consensus, the utility judge is not validated at all. The authors provide no data on the judge's accuracy, its agreement with human assessments of utility, or the robustness of its rubric.

**Questions:**

1. What PII is commonly present in MedQA, a curated medical expert QA dataset? How representative is this of PII that users would actually share, as opposed to PII that is simply embedded in the original exam questions?
2. Could using test-time compute (Gemini Pro Thinking or Claude Extended Thinking) help close the gap in PII removal between single-pass and expensive search-based methods?

---

> ### Author Response · Authors · 2025-11-21
>
> Thank you very much for the thoughtful and constructive review. We appreciate the time you spent engaging with the paper, and we are glad that you found the problem formulation and presentation to be clear and well-motivated. Several of your comments, especially on non-LLM baselines and validation of the utility judge, directly motivated new experiments during the rebuttal period, which strengthened the paper.
>
> **Weakness 1.**
> Thank you for highlighting this. Following your suggestion, we added a non-LLM baseline using a pure NER system. Since NER models cannot generate abstractions or reason about how much masking preserves utility, we adopted the strongest possible version of this baseline by redacting *all (100%)* detected PII spans. We applied this to ShareGPT and evaluated utility with GPT-4o. We tested the NER-only baseline on the same set of response models used in our oracle experiments, covering a frontier model (gpt-5), a strong frontier-tier closed model (Claude-3.7-Sonnet-20250219), a mid-sized open-weight model (Qwen2.5-7B-Instruct), and a small model (gpt-4.1-nano). The **NER-only** baseline fails utility at a *high* rate: for example, Claude-3.7-Sonnet-20250219 fails 75/176 cases *(42%)*, Qwen2.5-7B-Instruct fails 104/176 *(59%)*, gpt-4.1-nano fails 74/176 *(42%)*, and even gpt-5 fails 42/176 *(24%)*. In contrast, the oracle for each of these models always finds a utility-preserving transformation (*100%* pass rate) while still achieving a *substantial* level of masking. For example, for Claude-3.7-Sonnet-20250219 the oracle redacts 77.5% of spans on average (Table 2), compared to the NER baseline’s full redaction of 100%, yet still fails *42%* of cases. This demonstrates that pure NER cannot identify the maximal safe masking boundary or decide when abstraction is necessary, whereas the oracle search precisely finds the strongest masking that still preserves utility. We appreciate the reviewer for motivating this comparison.
>
>
> **Weakness 2.**
> Thank you for raising this concern. Prior work frequently employs LLM-as-a-Judge to assess relevance or utility, but the reviewer is right that this does not automatically justify using an LLM judge for our utility predicate. In response, we conducted a new human evaluation to directly measure how well GPT-4o aligns with human judgment as a utility judge. We recruited 75 English-speaking, US-based participants on Prolific. We sampled utility-evaluation pairs from our pipeline logs, where each question contains the *original user message* and a pair comparing the original response against the response to a masked version of the prompt.
>
> Pairs were randomly selected across all stages of our search process, covering both cases GPT-4o labeled PASS and FAIL. We selected 75 GPT-4o-PASS and 75 GPT-4o-FAIL cases, shuffled them, and distributed them across 15 surveys (10 items + 1 attention check each), ensuring each survey was completed by 5 different participants. For each question, humans chose **ACCEPT** (utility preserved) or **REJECT** (utility degraded). We then computed agreement conditional on human consensus.
>
> We analyze model–annotator agreement using a row-normalized confusion matrix, reporting *P(model decision | human consensus)*. This measures how consistently the model follows human majority judgment for both ACCEPT and REJECT cases.
>
>
> The results are summarized below:
>
> #### Consensus ≥ 0.6 (150 questions)
>
> | Human choice     | LLM=FAIL | LLM=PASS |
> |------------------|------------------------|----------|
> | ACCEPT (93 qs)   | 0.3118 (29 / 93)       | 0.6882 (64 / 93) |
> | REJECT (57 qs)   | 0.8070 (46 / 57)       | 0.1930 (11 / 57) |
>
>
> #### Consensus ≥ 0.8 (90 questions)
>
> | Human choice     | LLM=FAIL | LLM=PASS |
> |------------------|----------|----------|
> | ACCEPT (62 qs)   | 0.1613 (10 / 62) | 0.8387 (52 / 62) |
> | REJECT (28 qs)   | 0.9286 (26 / 28) | 0.0714 (2 / 28) |
>
> #### Consensus = 1.0 (36 questions)
>
> | Human choice     | LLM=FAIL | LLM=PASS |
> |------------------|----------|----------|
> | ACCEPT (32 qs)   | 0.0625 (2 / 32) | 0.9375 (30 / 32) |
> | REJECT (4 qs)    | 1.0000 (4 / 4)  | 0.0000 (0 / 4) |
>
>
> As human consensus increases, alignment with GPT-4o rises from 0.688 to 0.839 to 0.938, confirming that GPT-4o behaves reliably in high-consensus regimes. This pattern mirrors our privacy comparator results (Section 4.2), suggesting that both evaluators exhibit stable, human-aligned behavior where human agreement is strongest.

---

> ### Author Response · Authors · 2025-11-21
>
> continued ...
>
> **Question 1**
> Thank you for the question. In MedQA, the PII categories that appear are mainly those naturally involved in clinical case descriptions. Based on our extraction, the largest category is **HEALTH_INFORMATION**, followed by **AGE**, **GENDER**, **TIME**, **GEOLOCATION**, and **OCCUPATION**. These categories correspond to the kinds of contextual details commonly used in medical vignettes.
>
> Although MedQA does not contain some PII types that frequently appear in everyday conversations, such as personal names, this is expected because curated exam-style datasets intentionally remove overt identifiers. In contrast, in real-world interactions such as those in ShareGPT and WildChat, users often disclose health concerns together with attributes like age, time, and gender. *These are precisely the same categories that dominate MedQA.*
>
> For this reason, the PII in MedQA is narrower in variety but still representative of the information users naturally share when seeking medical or health-related advice. We use MedQA because it provides a domain with clear, fixed-answer utility, which allows us to evaluate data minimization in a setting that requires professional knowledge while still involving realistic health-related PII.
>
> **Question 2**
> In our setting, one of our single-pass predictions already uses a frontier reasoning model: **GPT-5**, which includes a built-in chain-of-thought–style reasoning mode, and we used it with its default *reasoning effort = medium*, as described in Section 5.2.
>
> GPT-5 is therefore directly comparable to models such as Gemini Pro Thinking or Claude’s Extended Thinking mode in terms of its ability to use test-time computation to reason through problems.
>
> And even with GPT-5’s reasoning ability, it still struggles in the prediction setting, just like the other single-pass models, and tends to predict **overshare** data minimization results. It also shows the same **abstraction-first bias** we report in Fig. 3 and Section 6.2.
>
> This is consistent with our framing: the prediction task requires anticipating utility under transformations, which is fundamentally different from normal inference-time reasoning.
>
> Once again, we appreciate that the reviewer identified these two nuanced points. Your comments were very helpful in strengthening the paper and clarifying the distinctions between reasoning capability and utility-preserving prediction.

---

> ### Author Response · Authors · 2025-11-25
>
> **Additional Response To Question 1**
>
> We added an additional analysis on top of our previous comment to examine this question more directly. We returned to the original ShareGPT and WildChat datasets and extracted all first turn user messages that are clearly related to health or medical topics. We applied the same pre filtering criteria as in our main experiments. The messages must be in English, they must contain a clear intent, and they must include at least three PII spans as independently detected by both the NER based and the LLM based detectors. This process produced 55 health related messages from WildChat and 33 from ShareGPT.
>
> We combined these *88* real world health related queries and compared their normalized PII distribution with that of the *108* MedQA questions used in our main study. In MedQA, the dominant category is HEALTH_INFORMATION, followed by AGE, GENDER, TIME, GEOLOCATION, and several other low-frequency categories. The real-world prompts from ShareGPT and WildChat show a similar pattern across these clinical categories, but they contain more frequent non-clinical identifiers (e.g., NAME, AFFILIATION) than MedQA, where such identifiers are probably rare due to curation.
>
> To quantify the similarity between the two distributions, we computed KL divergence and Jensen Shannon divergence. KL(medQA || ShareGPT plus WildChat) equals 0.34, and the JSD is 0.10, indicating that the two distributions are highly similar. The remaining differences arise from the fact that real users add non clinical identifiers such as names and affiliations, while MedQA intentionally omits these.
>
> These results show that MedQA contains a narrower but still representative subset of the PII that users naturally disclose when seeking medical or health related advice. The central categories that carry clinical utility, including HEALTH_INFORMATION, AGE, GENDER, TIME, and GEOLOCATION, appear consistently across both sources. This *supports* our use of MedQA as a structured benchmark for evaluating data minimization in medical contexts.

---

### Author Response · Authors · 2025-12-02

Thanks to all reviewers for the thoughtful and engaged discussion. We really appreciate how the feedback pushed us to strengthen our paper, and several of the main comments directly inspired new experiments during this rebuttal phase.

We added multiple new analyses: a non-LLM NER baseline, human validation of the utility judge, cross-model overlap for minimization consistency, self-attack privacy-recovery and γ (utility threshold) sensitivity studies, and a deeper look into models’ abstraction bias. These additions not only made the framework and results more solid, but also further confirmed that our original findings on minimization robustness and cross-model consistency are well-supported.

We are grateful for the constructive exchange and glad that the new experiments helped clarify both the practicality and robustness of our approach. *We have updated the PDF and annotated the changes.*

We also thank the reviewers who followed up after the rebuttal, including one who maintained a positive score of 8 and another who raised their rating from 4 to 6 (final scores **8, 6, 6, 4** before reverting back to the original score).

---

### Meta-Review · Area_Chair_82Cz · 2026-01-09

**Summary:**

This paper formalises data minimisation as a constrained optimisation problem. Authors provide the first oracle-based upper bounds for privacy-preserving prompt transformation. The priority-queue tree search algorithm systematically explores the space of prompt transformations ordered by privacy sensitivity.

The authors addressed major concerns with new experiments. For the camera-ready version, authors should incorporate all new experiments from the rebuttal (NER baseline, human validation, cross-model overlap, prompt ablations).

**Reviewer Concerns:**

### 1. Model-specificity and transferability [nGjY, Djhr, NzwN]

[Reviewers]
- Minimised prompts are model-specific (nGjY).
- How can results be applied to real applications given the requirement for intermediate prompt exposure? (Djhr, NzwN)
- Can minimised prompts from a smaller local model transfer to larger cloud LLMs? (Djhr)

[Authors] Added cross-model overlap experiments comparing each model's oracle against the GPT-5 oracle. For redaction, overlap is consistently high (75-85% across datasets) - core privacy removals are stable across models. Abstraction overlap is lower but accounts for a smaller portion of decisions. For deployment, the oracle is an offline mechanism to establish upper bounds. Real-world use would rely on a single-pass predictor that can be distilled from oracle outputs and run locally before sending prompts to cloud APIs. Self-attack experiments show that even GPT-5 attacking its own minimised prompts achieves near 0% PII recovery.

[Follow-up] nGjY raised rating from 4 to 6.

[AC] Core minimisation decisions seem to generalise across models.

### 2. Practical deployment concerns [Djhr, NzwN]

[Reviewers]
- The method assumes a locally deployed LLM while aiming to protect privacy when using LLMs via API (Djhr).
- It is unclear how findings can be directly applied to real applications (NzwN).
- Are all intermediate prompts exposed to the main LLM during search? (Djhr)

[Authors] The oracle search is not intended for on-device deployment. It serves to establish the upper bound of achievable data minimisation. The component for real-world use is the single-pass predictor that can be packaged as a small local model, trained or distilled from oracle outputs. Regarding intermediate prompt exposure, the self-attack experiment shows that GPT-5 cannot recover original PII from minimised prompts.

[Follow-up] NzwN maintained positive rating after clarifications.

[AC] Reasonable.

### 3. Representativeness of PII [ofax, nGjY]

[Reviewers] What PII is present in MedQA? How representative is this of real user disclosures? (ofax) Could the abstraction bias come from how models are instructed? (nGjY)

[Authors] For MedQA, dominant PII categories are HEALTH_INFORMATION, AGE, GENDER, TIME, and GEOLOCATION. A distributional comparison with health-related queries from ShareGPT and WildChat shows similarity. MedQA contains a narrower but representative subset of health-related PII. For abstraction bias, prompt ablation experiments tested three variants. All models maintain strong abstraction preference across all variants. This confirms the bias is structural to instruction-tuned LLMs rather than induced by the prompt.

[AC] Concern is addressed.


### 4. Validation of evaluation methodology [ofax, Djhr]

[Reviewers] The utility judge (GPT-4o) is not validated for alignment with human assessments. Binary utility classification oversimplifies the continuous performance characteristics of LLMs. Privacy evaluation relies on pairwise comparison without theoretical guarantees.

[Authors] Conducted a human study with 75 Prolific participants comparing GPT-4o PASS/FAIL decisions against human judgments. The decisions are aligned rather well between humans and LLM. For binary utility, the authors argue this is the standard formulation for constrained optimisation. An additional sensitivity study on the utility threshold with 90 participants shows that even small relaxations (1 or 2) lead to human-perceptible quality degradation. On theoretical privacy guarantees, the authors acknowledge this is a broader challenge for semantic privacy. Unlike differential privacy, their approach targets the prompt layer before model interaction.

[AC] Authors response resolves concern to some degree.

### 5. Lack of comparison with non-LLM baselines [ofax]

[Reviewers] The paper compares its expensive oracle search against single-pass LLM predictors. It omits a simpler and more practical baseline using standard PII identification or NER tools to identify and redact PIIs.

[Authors] Added a NER-only baseline that redacts 100% of detected PII spans on ShareGPT. Results show high utility failure rates. In contrast, the oracle achieves 100% pass rate while still redacting a substantial portion (77.5% for Claude). This demonstrates that pure NER cannot identify the maximal safe masking boundary.

[AC] Authors response makes sense.

**Reviewer Scores:**

ofax: 4 > 6 (non-LLM baseline added)
nGjY: 4 > 6 (explicitly raised rating)
Djhr: 6 > 6 (concerns addressed)
NzwN: 8 > 8 (explicitly maintained rating)

---

### Decision · Program_Chairs · 2026-01-26

Accept (Poster)